

# A Novel Transformation of the Ice Sheet Stokes Equations and Some of its Properties and Applications

John K. Dukowicz

Guest Scientist, Group T-3, Los Alamos National Laboratory,

Los Alamos, New Mexico, 87545, USA

*Correspondence to:* John K. Dukowicz (jn.dk@outlook.com)

**Abstract.**    A full-Stokes model provides the most accurate but also the most expensive representation of ice sheet dynamics. The Blatter-Pattyn model is a widely used less expensive approximation that is valid for ice sheets characterized by a small aspect ratio. Here we introduce a novel transformation of the Stokes equations into a form that closely resembles the Blatter-Pattyn equations. The transformed exact Stokes equations only differ from the approximate Blatter-Pattyn equations by a few additional terms, while their variational formulations differ only by the presence of a single term in each horizontal direction (one term in 2D and two terms in 3D). Specifically, the variational formulations differ only by the absence (or the neglect) of the vertical velocity in the second invariant of the strain rate tensor in the Blatter-Pattyn model when compared to the Stokes case. Here we make use of the new transformation in two different ways. First, we consider incorporating the transformed equations into a code that can be very easily converted from a Stokes to a Blatter-Pattyn model, and vice-versa, simply by switching these terms on or off. This may be generalized so that the Stokes model is switched on adaptively only where the Blatter-Pattyn model loses accuracy, hopefully retaining most of the accuracy of the Stokes model but at a lower cost. Second, the key role played by the vertical velocity in converting the transformed Stokes model into the Blatter-Pattyn model motivates new approximations that improve on the Blatter-Pattyn model, heretofore the best approximate ice sheet model. These applications require the use of a grid that enables the discrete continuity equation to be invertible for the vertical velocity in terms of the horizontal velocity components. Examples of such grids, such as the first order P1-E0 grid and the second order P2-E1 grid are given in both 2D and 3D. It should be noted, however, that the transformed Stokes model has the same type of gravity forcing as the Blatter-Pattyn model, i.e., determined by the slope of the ice sheet upper surface, thereby forgoing some of the grid-generality of the traditional formulation of the Stokes model. This is not a serious disadvantage, however, since in practice it has not impaired the widespread use of the Blatter-Pattyn model.



## 1 Introduction

Concern and uncertainty about the magnitude of sea level rise due to melting of the
Greenland and Antarctic ice sheets have led to increased interest in improved ice sheet
and glacier modeling. The gold standard is a full-Stokes model (i.e., a model that solves
the nonlinear, non-Newtonian Stokes system of equations for incompressible ice sheet
dynamics) because it is applicable to all geometries and flow regimes. However, the
Stokes model is computationally demanding and expensive to solve. It is a nonlinear,
three-dimensional model involving four variables, namely, the three velocity components
and pressure. In addition, pressure is a Lagrange multiplier enforcing incompressibility
and this creates a more difficult indefinite "saddle point" problem. As a result, full-
Stokes models exist but are not commonly used in practice (examples are FELIX-S, Leng
et al., 2012; Elmer/Ice, Gagliardini et al., 2013).

Because of these difficulties with the Stokes model, there is much interest in
simpler and cheaper approximate models. There is a hierarchy of very simple models
such as the shallow ice (SIA) and shallow-shelf (SSA) models, and there are also various
higher-order approximations. These culminate in the Blatter-Pattyn (BP) approximation
(Blatter, 1995; Pattyn, 2003), which is currently used in production code packages such
as ISSM (Larour et al., 2012), MALI (Hoffman et al., 2018; Tezaur et al., 2015) and
CISM (Lipscomb et al., 2019). This approximation is based on the assumption of a small
ice sheet aspect ratio, i.e., $\varepsilon = H/L \ll 1$, where $H, L$ are the vertical and horizontal
length scales, and consequently it eliminates certain stress terms and implicitly assumes
small basal slopes. Both the Stokes and Blatter-Pattyn models are described in detail in
Dukowicz et al. (2010), hereafter referred to as DPL (2010). Although the Blatter-Pattyn
model is reasonably accurate for large-scale motions, accuracy deteriorates for small
horizontal scales, less than about five ice thicknesses in the ISMIP–HOM model
intercomparison (Pattyn et al., 2008; Perego et al., 2012), or below a 1 km resolution as
found in a detailed comparison with full Stokes calculations (Rückamp et al, 2022). This
can become particularly important for calculations involving details near the grounding
line where the full accuracy of the Stokes model is needed (Nowicki and Wingham,
2008). Attempts to address the problem while avoiding the use of full Stokes solvers
include variable grid resolution coupled with a Blatter-Pattyn solver (Hoffman et al.,
2018) and variable model complexity, where a Stokes solver is embedded locally in a



lower order model (Seroussi et al., 2012). Better approximations, more accurate than
Blatter-Pattyn but cheaper than Stokes, are not currently available.

The present paper introduces two innovations that may begin to address some of

these issues. The first is a novel transformation of the Stokes model, described in §3,
which puts it into a form closely resembling the Blatter-Pattyn model and differing only
by the presence of a few extra terms. This allows a code to be switched over from Stokes
to Blatter-Pattyn, and vice-versa, globally or locally, by the use of a single parameter that
turns off these extra terms. As a result, variable model complexity can be very simply
implemented, as described in §6.1. The second innovation is the introduction of new
finite element grids that decouple the discrete continuity equation and allow it to be
solved for the vertical velocity in terms of the horizontal velocity components. Several
elements that may be used to construct such grids are described in Appendix C in both
2D and 3D, primarily the first order P1-E0 and second order P2-E1 elements (these two
elements are so-named because they employ edge-based pressures). Within the
framework of the transformed Stokes model these grids facilitate new approximations
that improve on the Blatter-Pattyn approximation so that it is no longer strictly limited to
a small ice sheet aspect ratio. We describe two such approximations in §6.2. There is
another very significant benefit. A conventional ice sheet Stokes model discretized on
such a grid is numerically equivalent to an inherently stable positive-definite
minimization (i.e., optimization) problem, as demonstrated in Appendix D. This is in
contrast to the ubiquitous Stokes finite element practice of needing to use elements that
satisfy the "inf-sup" or "LBB" condition for stability (see Elman et al., 2014, and the
brief discussion in §4.3.1).

**2 The Standard Formulation of the Stokes Ice Sheet Model**
**2.1 The Assumed Ice Sheet Configuration**

An ice sheet may be divided into two parts, a part in contact with the bed and a floating
ice shelf located beyond the grounding line. The Stokes ice sheet model is capable of
describing the flow of an arbitrarily shaped ice sheet, including a floating ice shelf as
illustrated in Fig. 1, given appropriate boundary conditions (e.g., Cheng et al., 2020).
One limitation of the methods proposed here, in common with the Blatter-Pattyn model,
will be that upper and basal surfaces must able to be connected by a vertical line of sight,



as is the case in Fig. 1. Here, for simplicity, we will only consider a fully grounded ice
sheet with periodic lateral boundary conditions, i.e., no ice shelf.

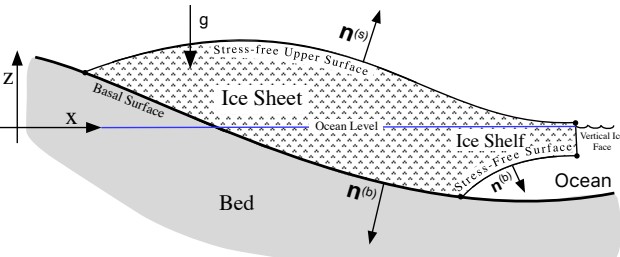

**Figure 1** A simplified illustration of the admissible ice sheet configuration.


Referring to Fig. 1, the entire surface of the ice sheet is denoted by $S$. An upper

surface, labeled $S_S$ and specified by $\varsigma_s(x,y,z) = z - z_s(x,y) = 0$, is exposed to the
atmosphere and thus experiences stress-free boundary conditions. The bottom or basal
surface, denoted by $S_B$ and specified by $\varsigma_b(x,y,z) = z - z_b(x,y) = 0$, is in contact with
the bed. The basal surface may be subdivided into two sections, $S_B = S_{B1} + S_{B2}$, where
$S_{B1}$, specified by $z = z_{b1}(x,y)$, is the part where ice is frozen to the bed (a no-slip
boundary condition), and $S_{B2}$, specified by $z = z_{b2}(x,y)$, is where frictional sliding
occurs. We assume Cartesian coordinates such that $x_i = (x,y,z)$ are position coordinates
with $z = 0$ at the ocean surface, and the index $i \in \{x,y,z\}$ represents the three Cartesian
indices. Later we shall have occasion to introduce the restricted index $(i) \in \{x,y\}$ to
represent just the two horizontal indices. The associated unit normal vectors are $n_i^{(s)}$,
$n_i^{(b1)}$, $n_i^{(b2)}$ at the stress-free and basal surfaces, respectively. For the particular geometry
illustrated in Fig. 1 we see that $n_z^{(s)} > 0$ and $n_z^{(b1)}, n_z^{(b2)} < 0$. Unit normal vectors
appropriate for the ice sheet configuration of Fig. 1 are given by





$$n_i^{(s)} = \left(n_x^{(s)}, n_y^{(s)}, n_z^{(s)}\right) = \frac{\partial \varsigma_s\left(x,y,z\right)\big/\partial x_i}{\left|\partial \varsigma_s\left(x,y,z\right)\big/\partial x_i\right|} = \frac{\left(-\partial z_s\big/\partial x, -\partial z_s\big/\partial y, 1\right)}{\sqrt{1+\left(\partial z_s\big/\partial x\right)^2+\left(\partial z_s\big/\partial y\right)^2}},$$

$$n_i^{(b)} = \left(n_x^{(b)}, n_y^{(b)}, n_z^{(b)}\right) = -\frac{\partial \varsigma_b\left(x,y,z\right)\big/\partial x_i}{\left|\partial \varsigma_b\left(x,y,z\right)\big/\partial x_i\right|} = \frac{\left(\partial z_b\big/\partial x, \partial z_b\big/\partial y, -1\right)}{\sqrt{1+\left(\partial z_b\big/\partial x\right)^2+\left(\partial z_b\big/\partial y\right)^2}}.$$

(1)



**2.2 The Stokes Equations**

The Stokes model is given by a system of nonlinear partial differential equations and
associated boundary conditions (Greve and Blatter, 2009; DPL, 2010). In a Cartesian
coordinate system the Stokes equations, the three momentum equations and the
continuity equation, for the three velocity components $u_i = \left(u,v,w\right)$ and the pressure $P$
are given by
$$\frac{\partial \tau_{ij}}{\partial x_j} - \frac{\partial P}{\partial x_i} + \rho g_i = 0 \, ,$$
(2)

$$\frac{\partial u_i}{\partial x_i} = 0 \, ,$$
(3)

where $\rho$ is the density, and $g_i$ is the acceleration due to gravity vector, arbitrarily
oriented in general but here taken to be oriented in the negative z-direction,
$g_i = \left(0,0,-g\right)$. Repeated indices imply summation (the Einstein notation). The
deviatoric stress tensor $\tau_{ij}$ is given by
$$\tau_{ij} = 2\mu_n \, \dot{\varepsilon}_{ij} \, ,$$
(4)

where $\mu_n$ is a nonlinear ice viscosity defined by
$$\mu_n = \eta_0 \left(\dot{\varepsilon}^2\right)^{(1-n)/2n} \, ,$$
(5)

and $\dot{\varepsilon}^2 = \dot{\varepsilon}_{ij}\dot{\varepsilon}_{ij}\big/2$ is the second invariant of the strain rate tensor $\dot{\varepsilon}_{ij}$. The strain rate
tensor is given by
$$\dot{\varepsilon}_{ij} = \frac{1}{2}\left(\frac{\partial u_i}{\partial x_j} + \frac{\partial u_j}{\partial x_i}\right),$$
(6)

and therefore the second invariant may be written out as



$$\dot{\varepsilon}^2 = \frac{1}{2}\left[\left(\frac{\partial u}{\partial x}\right)^2 + \left(\frac{\partial v}{\partial y}\right)^2 + \left(\frac{\partial w}{\partial z}\right)^2\right] + \frac{1}{4}\left[\left(\frac{\partial u}{\partial y} + \frac{\partial v}{\partial x}\right)^2 + \left(\frac{\partial u}{\partial z} + \frac{\partial w}{\partial x}\right)^2 + \left(\frac{\partial v}{\partial z} + \frac{\partial w}{\partial y}\right)^2\right]. \quad (7)$$
Note that the second invariant is positive-definite, i.e., $\dot{\varepsilon}^2 \geq 0$. As usual, ice is assumed
to obey Glen's flow law, where $n$ is the Glen's law exponent ($n = 1$ for a linear
Newtonian fluid, and typically $n = 3$ in ice sheet modeling, resulting in a nonlinear non-
Newtonian fluid). The coefficient $\eta_0$ is defined by $\eta_0 = A^{-1/n}/2$, where $A$ is an ice flow
factor, here taken to be a constant but in general depending on temperature and other
variables (see Schoof and Hewitt, 2013). The three-dimensional Stokes system (2), (3)
requires a set of boundary conditions at every bounding surface, each set being composed
of three components. Aside from the periodic lateral boundary conditions used in our test
problems, the relevant boundary conditions are as follows
(1) Stress-free boundary conditions on surfaces $S_S$ not in contact with the bed, such
as the upper surface $S_S$:
$$\tau_{ij}n_j^{(s)} - Pn_i^{(s)} = 0. \quad (8)$$
The basal boundary conditions are given by
(2) No-slip or frozen to the bed conditions on surface segment $S_{B1}$:
$$u_i = 0 \quad (9)$$
(3) Frictional tangential sliding conditions on surface segment $S_{B2}$:
Frictional conditions are more complicated and are discussed in detail in Appendix A. In
summary, these conditions are composed of two parts,
(3a) A single condition enforcing tangential flow at the basal surface:
$$u_i n_i^{(b2)} = 0. \quad (10)$$
(3b) Two conditions specifying the horizontal components of the tangential
frictional stress force vector. From Appendix A, the simplest representation of these two
conditions is
$$n_z^{(b2)}\left(\tau_{(i)j}n_j^{(b2)} + f_{(i)}\right) - n_{(i)}^{(b2)}\left(\tau_{zj}n_j^{(b2)} + f_z\right) = 0, \quad (11)$$
where $(i) \in \{x, y\}$ is the notation previously introduced for restricted (horizontal) indices,
and $f_i$ is a specified frictional sliding force vector, tangential to the bed $\left(n_i^{(b2)}f_i = 0\right)$.



This is potentially a complicated function of position and velocity (e.g., Schoof, 2010),
however, here we assume only simple linear frictional sliding,
$$f_i = \beta(x)\, u_i,\qquad(12)$$

where $\beta(x) > 0$ is a position-dependent drag law coefficient. For simplicity we assume
there is no melting or refreezing at the bed resulting in vertical inflows or outflows. If
needed, these can be easily added (Dukowicz et al., 2010; Heinlein et al., 2022).

**2.3 The Stokes Variational Principle**

A variational principle, if available, is usually the most compact way of representing a
particular problem. The Stokes model possesses a variational principle that is
particularly useful for discretization purposes and for the specification of boundary
conditions (see DPL, 2010, for a fuller description of the variational principle applied to
ice sheet modeling). There are a number of significant advantages. For example, all
boundary conditions are conveniently incorporated in the variational formulation, all
terms in the variational functional, including boundary condition terms, contain lower
order derivatives than in the momentum equations, and the solution of the discrete
problem automatically involves a symmetric matrix. In discretizing the momentum
equations, stress terms at boundaries involve derivatives that require information from
across boundaries. This problem does not arise in the variational formulation since all
terms are evaluated in the interior. Finally, stress-free boundary conditions, as at the
upper surface for example, need not be specified at all since they are automatically
incorporated in the functional as natural boundary conditions. In discrete applications,
the variational method presented here is closely related to the Galerkin finite element
method, a subset of the weak formulation method in which the test and trial functions are
the same (see Schoof, 2010, in connection with the Blatter-Pattyn model).

The variational functional for the standard Stokes model may be written in two
alternative forms:
(1) Basal boundary conditions imposed using Lagrange multipliers:
$$\mathcal{A}[u_i, P, \lambda_i, \Lambda] = \int_V dV\left[\frac{4n}{n+1}\eta_0\left(\dot{\varepsilon}^2\right)^{(1+n)/2n} - P\frac{\partial u_i}{\partial x_i} + \rho g w\right]$$
$$+ \int_{S_{B1}} dS\,\lambda_i u_i + \int_{S_{B2}} dS\left[\Lambda u_i n_i^{(b2)} + \frac{1}{2}\beta(x)\, u_i u_i\right],\qquad(13)$$



where $\lambda_i$ and $\Lambda$ are Lagrange multipliers used to enforce the no-slip condition and
frictional tangential sliding, respectively.  As in DPL (2010), arguments enclosed in
square brackets, here $u_i, P, \lambda_i, \Lambda$, indicate those variables that are used in the variation of
the functional.

(2) Basal boundary conditions imposed by direct substitution:

In this case, the two conditions (9), (10) are used directly in the functional to specify all
three velocity components $u_i$ in the first case, and the vertical velocity $w$ in terms of the
horizontal velocity components in the second case, along the entire basal boundary in
both the volume and surface integrals in (13).  In particular, (10) is used in the following
form,

$$w = -\frac{u_{(i)} n_{(i)}^{(b2)}}{n_z^{(b2)}} = u_{(i)} \frac{\partial z_b}{\partial x_{(i)}}, \tag{14}$$

to replace $w$ in terms of the horizontal velocity components $u_{(i)}$ on the basal boundary
segment $S_{B2}$.  Here we use $z_b$ as a shorthand notation for $z_b(x, y)$.  The variational
functional in this case becomes

$$\mathcal{A}[u_i, P] = \int_V dV \left[ \frac{4n}{n+1} \eta_0 \left( \dot{\varepsilon}^2 \right)^{(1+n)/2n} - P \frac{\partial u_i}{\partial x_i} + \rho g w \right]$$
$$+ \frac{1}{2} \int_{S_{B2}} dS\, \beta(x) \left( u_{(i)} u_{(i)} + \left( u_{(i)} n_{(i)}^{(b2)} / n_z^{(b2)} \right)^2 \right). \tag{15}$$

Note that (14) has been explicitly used to replace $w$ in the basal boundary component of
the functional (15) but, importantly, it must also be used in the volume integral part of
(15) to replace all values of $w$ that lie on the basal boundary segment $S_{B2}$.

As described in DPL (2010), a variational procedure, i.e., taking the variation

with respect to the independent functions $u_i, P, \lambda_i, \Lambda$ in (13), and $u_i, P$ in (15), yields the
full set of Euler-Lagrange equations and boundary conditions that specify the standard
Stokes model, equivalent to (2)-(11).  In the case of (13), the system determines all the
discrete variables specified on the mesh: the velocity components and the pressure, $u_i, P$,
together with the Lagrange multipliers, $\lambda_i, \Lambda$.  In the case of (15), the system only
determines the unspecified velocity variables $u_i$ and the pressure $P$.  The specified





values of velocity are then obtainable a posteriori from (9) or (14). As a result, system
(15) is smaller and simpler and is therefore the one predominantly used in this paper.

**3. A Transformation of the Stokes Model**
**3.1 Origin of the Transformation**

The transformation is motivated by the Blatter-Pattyn approximation. Consider the
vertical component of the momentum equation and the corresponding stress-free upper
surface boundary condition in the Blatter-Pattyn approximation (from DPL, 2010, for
example), which are given by

$$\frac{\partial}{\partial z}\left(2\mu_n \frac{\partial w}{\partial z}\right) - \frac{\partial P}{\partial z} - \rho g = 0,$$

$$\left(2\mu_n \frac{\partial w}{\partial z} - P\right)n_z^{(s)} = 0 \quad \text{at} \quad z = z_s(x,y).$$

(16)

These equations may be rewritten in the form

$$\frac{\partial}{\partial z}\left(P - 2\mu_n \frac{\partial w}{\partial z} + \rho g\left(z - z_s(x,y)\right)\right) = 0,$$

$$\left(P - 2\mu_n \frac{\partial w}{\partial z} + \rho g\left(z - z_s(x,y)\right)\right)n_z^{(s)} = 0 \quad \text{at} \quad z = z_s(x,y).$$

(17)

This suggests the introduction of a new variable $\tilde{P}$, to be called the transformed pressure,

$$\tilde{P} = P - 2\mu_n \frac{\partial w}{\partial z} + \rho g\left(z - z_s(x,y)\right),$$

(18)

which simplifies system (17) as follows

$$\frac{\partial \tilde{P}}{\partial z} = 0,$$

$$\tilde{P} n_z^{(s)} = 0 \quad \text{at} \quad z = z_s(x,y).$$

(19)

This is a complete one-dimensional partial differential system, that, when integrated from
the top surface down yields

$$\tilde{P} = 0 \,.$$

(20)

Thus, the transformed pressure vanishes in the Blatter-Pattyn case. The definition (18)
forms the basis of the present transformation but we also use the continuity equation to
eliminate $\partial w/\partial z$ as is done in the Blatter-Pattyn approximation (see DPL, 2010).



Therefore, the transformation consists of eliminating $P$ and $\partial w/\partial z$ in the Stokes system
(2), (4)-(11) (i.e., everywhere except in the continuity equation (3) itself) by means of

$$P = \tilde{P} - 2\mu_n\left(\frac{\partial u}{\partial x} + \frac{\partial v}{\partial y}\right) + \rho g\left(z_s - z\right), \tag{21}$$


$$\frac{\partial w}{\partial z} = -\left(\frac{\partial u}{\partial x} + \frac{\partial v}{\partial y}\right), \tag{22}$$

where $z_s$ is a shorthand notation for $z_s\left(x,y\right)$.

In the standard Stokes system the pressure P is primarily a Lagrange multiplier

enforcing incompressibility but with a very large hydrostatic component. The
transformation eliminates the hydrostatic pressure from $\tilde{P}$, as illustrated in Fig. 2 where
the two pressures, plotted along grid lines, from Exp. B in the ISMIP–HOM model
intercomparison (Pattyn et al., 2008) at L = 10 km are compared. The standard Stokes
pressure $P$ is some three orders of magnitude larger than the transformed pressure $\tilde{P}$.

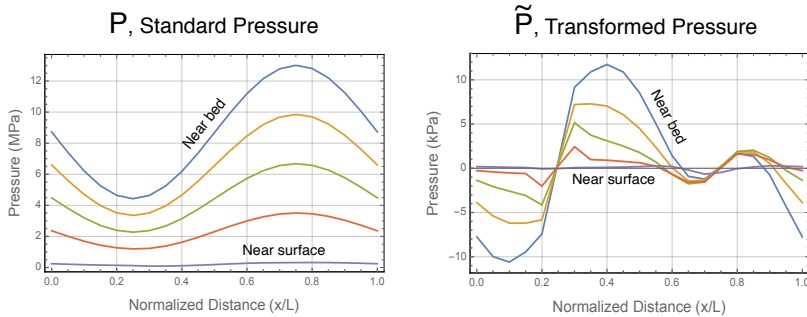


**Figure 2.** Standard pressure $P$ compared to the transformed pressure $\tilde{P}$ in Exp. B from

the ISMIP–HOM model intercomparison. Note that $P$ is in MPa while $\tilde{P}$ is in kPa.


The transformed pressure $\tilde{P}$ is again a Lagrange multiplier enforcing

incompressibility, i.e., it may be viewed as the effective pressure in the transformed
system. Alternatively, since $\tilde{P} = 0$ in the Blatter-Pattyn approximation, the definition of
$\tilde{P}$ from (18) may be written as $\tilde{P} = P - P_{BP}$, where

$$P_{BP} = -2\mu_n\left(\frac{\partial u}{\partial x} + \frac{\partial v}{\partial y}\right) + \rho g\left(z_s - z\right)$$



is the effective Blatter-Pattyn pressure (Tezaur et al., 2015). As a result, we have
$P = P_{BP} + \tilde{P}$, and therefore $\tilde{P}$ is actually the "Stokes" correction to the Blatter-Pattyn
pressure.

**3.2 The Transformed Stokes Equations**

Introducing (21), (22) into the Stokes system of equations (2)-(11) results in the
following transformed Stokes system:
$$\frac{\partial \tilde{\tau}_{ij}}{\partial x_j} - \hat{\xi}\frac{\partial \tilde{P}}{\partial x_i} - \rho g\frac{\partial z_s}{\partial x_{(i)}} = 0 \, , \tag{23}$$

$$\hat{\xi}\frac{\partial u_i}{\partial x_i} = 0 \, , \tag{24}$$

where quantities that are modified in the transformation are indicated by a tilde, e.g., $\tilde{P}$.
Corresponding to (4), the modified Stokes deviatoric stress tensor $\tilde{\tau}_{ij}$ is given by
$$\tilde{\tau}_{ij} = 2\tilde{\mu}_n\left(\tilde{\dot{\varepsilon}}_{ij} + \frac{\partial u_{(i)}}{\partial x_{(i)}}\delta_{ij}\right), \tag{25}$$

where $\delta_{ij}$ is the Kronecker delta, the modified strain rate tensor $\tilde{\dot{\varepsilon}}_{ij}$, corresponding to (6),
is given by
$$\tilde{\dot{\varepsilon}}_{ij} = \begin{bmatrix} \frac{\partial u}{\partial x} & \frac{1}{2}\left(\frac{\partial u}{\partial y}+\frac{\partial v}{\partial x}\right) & \frac{1}{2}\left(\frac{\partial u}{\partial z}+\xi\frac{\partial w}{\partial x}\right) \\[2mm] \frac{1}{2}\left(\frac{\partial u}{\partial y}+\frac{\partial v}{\partial x}\right) & \frac{\partial v}{\partial y} & \frac{1}{2}\left(\frac{\partial v}{\partial z}+\xi\frac{\partial w}{\partial y}\right) \\[2mm] \frac{1}{2}\left(\frac{\partial u}{\partial z}+\xi\frac{\partial w}{\partial x}\right) & \frac{1}{2}\left(\frac{\partial v}{\partial z}+\xi\frac{\partial w}{\partial y}\right) & -\left(\frac{\partial u}{\partial x}+\frac{\partial v}{\partial y}\right) \end{bmatrix} \tag{26}$$

and, corresponding to (5), the modified viscosity,
$$\tilde{\mu}_n = \eta_0\left(\tilde{\dot{\varepsilon}}^2\right)^{(1-n)/2n} \, , \tag{27}$$

is given in terms of the second invariant $\tilde{\dot{\varepsilon}}^2 = \tilde{\dot{\varepsilon}}_{ij}\tilde{\dot{\varepsilon}}_{ij}/2$, which, in expanded form becomes
$$\tilde{\dot{\varepsilon}}^2 = \left(\frac{\partial u}{\partial x}\right)^2 + \frac{\partial u}{\partial x}\frac{\partial v}{\partial y} + \left(\frac{\partial v}{\partial y}\right)^2 + \frac{1}{4}\left[\left(\frac{\partial u}{\partial y}+\frac{\partial v}{\partial x}\right)^2 + \left(\frac{\partial u}{\partial z}+\xi\frac{\partial w}{\partial x}\right)^2 + \left(\frac{\partial v}{\partial z}+\xi\frac{\partial w}{\partial y}\right)^2\right]. \tag{28}$$





The dummy variables $\xi = 1, \hat{\xi} = 1$ identify terms that are dropped in the Blatter-Pattyn
approximation, as explained below. Since (28) differs from (7) only by the use of
substitution (22), the transformation leaves the second invariant $\tilde{\dot{\varepsilon}}^2$ and viscosity $\tilde{\mu}_n$
unchanged provided the continuity equation (24) is satisfied, i.e., $\tilde{\dot{\varepsilon}}^2 = \dot{\varepsilon}^2$ and $\tilde{\mu}_n = \mu_n$,
and in particular, the transformed second invariant remains positive-definite, i.e., $\tilde{\dot{\varepsilon}}^2 \geq 0$.

The boundary conditions for the transformed equations, corresponding to (8)-(11),

are given by
BCs on $S_S$:
$$\tilde{\tau}_{ij}n_j^{(s)} - \tilde{\xi}\tilde{P}n_i^{(s)} = 0 \,, \tag{29}$$

BCs on $S_{B1}$:
$$u_i = 0 \,, \tag{30}$$

BCs on $S_{B2}$:
$$u_i n_i^{(b2)} = 0 \,, \tag{31}$$

$$n_z^{(b2)}\left(\tilde{\tau}_{(i)j}n_j^{(b2)} + \beta(x)u_{(i)}\right) - n_{(i)}^{(b2)}\left(\tilde{\tau}_{zj}n_j^{(b2)} + \beta(x)u_{(j)}n_{(j)}^{(b2)}\Big/n_z^{(b2)}\right) = 0 \,. \tag{32}$$

Equations (31), (32) constitute the three required boundary conditions for frictional
sliding (see Appendix A). Note that (32) differs from (11) because (14) has been used to
eliminate the vertical velocity component $w$ in favor of the horizontal velocity
components $u_{(i)}$.

The dummy variables $\xi, \hat{\xi}$ in (23)-(25) and (26)-(29) have been introduced to

identify the terms that are neglected in the two types of the Blatter-Pattyn approximation
that we consider in §3.4. Specifically, these two types are (a) the standard Blatter-Pattyn
approximation, $\xi = 0, \hat{\xi} = 0$, as originally derived (Blatter, 1995; Pattyn, 2003; DPL,
2010), which solves for just the horizontal velocity components $u, v$, and (b) the extended
Blatter-Pattyn approximation, $\xi = 0, \hat{\xi} = 1$, described more fully later, which contains the
standard approximation and also provides the additional equations for determination of
the consistent vertical velocity component $w$ and pressure $\tilde{P}$. Keeping all terms, i.e.,
$\xi = 1, \hat{\xi} = 1$, specifies the full transformed Stokes model.

The transformed system (25)-(32) and the standard Stokes system (2)-(11) yield

exactly the same solution. However, in common with the Blatter-Pattyn approximation,



transformation (21) implies the use of a gravity-oriented coordinate system because of the
particular form of the gravitational forcing term, while the standard Stokes model does
not have this restriction. This is only a minor limitation. A somewhat more restrictive
limitation is the appearance of $z_s(x,y)$, an implicitly single valued function, to describe
the vertical position of the upper surface of the ice sheet. This means that care must be
taken in case of reentrant upper surfaces (i.e., S-shaped in 2D) and sloping cliffs at the ice
edge, a restriction not present in the standard Stokes model. As noted earlier, we assume
that the upper and basal surfaces are connected by a vertical line of sight. With a
reentrant ice surface, such a vertical line must be broken up into individual segments with
each segment having its own "upper" surface location $z_s(x,y)$. Fortunately, such
situations do not normally arise in practice. Exactly these same limitations exist in the
Blatter-Patten model, which does not hinder its use in practice.

**3.3 The Transformed Stokes Variational Principle**

It is easy to verify that the transformed Stokes system (23)-(32) results from the variation
with respect to $u_i, \tilde{P}$ of the following functional:

$$\tilde{\mathcal{A}}[u_i, \tilde{P}] = \int_V dV \left[ \frac{4n}{n+1} \eta_0 \left(\tilde{\tilde{\varepsilon}}^2\right)^{(1+n)/2n} - \hat{\xi}\tilde{P}\frac{\partial u_i}{\partial x_i} + \rho g u_{(i)} \frac{\partial z_s}{\partial x_{(i)}} \right] + \frac{1}{2}\int_{S_{B2}} dS\, \beta(x)\left( u_{(i)}u_{(i)} + \left(u_{(i)}n_{(i)}^{(b2)}\big/n_z^{(b2)}\right)^2 \right),$$

(33)

where $\tilde{\tilde{\varepsilon}}^2$ is the transformed second invariant from (28). Basal boundary conditions in
(33) are imposed by direct substitution, as in (15). Alternatively, one could impose
boundary conditions using Lagrange multipliers, as in (13), but direct substitution is
preferred because it is simpler and involves fewer variables. The remarks made in §2.3
about replacing all values of $w$ that lie on the basal boundary segment $S_{B2}$ by (14) apply
here also.

**3.4 Two Blatter-Pattyn Approximations**
**3.4.1 The Standard Blatter-Pattyn Approximation**

The standard (or traditional) Blatter-Pattyn approximation (originally introduced by
Blatter, 1995; Pattyn, 2003; later by DPL, 2010; Schoof and Hewitt, 2013) is obtained by



setting $\xi = 0, \hat{\xi} = 0$. This yields the following Blatter-Pattyn variational functional in
terms of horizontal velocity components only,

$$
\mathcal{A}_{BP}[u_{(i)}] = \int_V dV \left[ \frac{4n}{n+1} \eta_0 \left( \dot{\varepsilon}_{BP}^2 \right)^{(1+n)/2n} + \rho g u_{(i)} \frac{\partial z_s}{\partial x_{(i)}} \right]
$$
$$
+ \frac{1}{2} \int_{S_{B2}} dS \, \beta(x) \left( u_{(i)} u_{(i)} + \varsigma \left( u_{(i)} n_{(i)}^{(b2)} / n_z^{(b2)} \right)^2 \right),
$$
(34)

where

$$
\dot{\varepsilon}_{BP}^2 = \left( \frac{\partial u}{\partial x} \right)^2 + \frac{\partial u}{\partial x} \frac{\partial v}{\partial y} + \left( \frac{\partial v}{\partial y} \right)^2 + \frac{1}{4} \left[ \left( \frac{\partial u}{\partial y} + \frac{\partial v}{\partial x} \right)^2 + \frac{\partial u}{\partial z}^2 + \frac{\partial v}{\partial z}^2 \right],
$$
(35)

and the corresponding Euler-Lagrange equations and boundary conditions are given by

$$
\frac{\partial \tau_{(i)j}^{BP}}{\partial x_j} - \rho g \frac{\partial z_s}{\partial x_{(i)}} = 0; \quad \begin{cases} \tau_{(i)j}^{BP} n_j^{(b2)} + \beta(x) \left( u_{(i)} + \zeta \left( u_{(j)} n_{(j)}^{(b2)} / n_z^{(b2)} \right) n_{(i)}^{(b2)} / n_z^{(b2)} \right) = 0 \\ \text{on } S_{B2}, \quad \tau_{(i)j}^{BP} n_j^{(s)} = 0 \text{ on } S_S, \quad u_{(i)} = 0 \text{ on } S_{B1}, \end{cases}
$$
(36)

where the Blatter-Pattyn stress tensor $\tau_{(i)j}^{BP}$ is

$$
\tau_{(i)j}^{BP} = \eta_0 \left( \dot{\varepsilon}_{BP}^2 \right)^{(1-n)/2n} \begin{bmatrix} 2 \left( 2 \frac{\partial u}{\partial x} + \frac{\partial v}{\partial y} \right) & \left( \frac{\partial u}{\partial y} + \frac{\partial v}{\partial x} \right) & \frac{\partial u}{\partial z} \\ \left( \frac{\partial u}{\partial y} + \frac{\partial v}{\partial x} \right) & 2 \left( \frac{\partial u}{\partial x} + 2 \frac{\partial v}{\partial y} \right) & \frac{\partial v}{\partial z} \end{bmatrix}.
$$
(37)

There is a new dummy variable $\zeta$ in (34) introduced to identify the basal boundary term
that is normally dropped $(\zeta = 0)$ in the standard Blatter-Pattyn approximation but which
was restored $(\zeta = 1)$ in Dukowicz et al. (2011) to better deal with arbitrary basal
topography.

The Blatter-Pattyn model is a well-behaved nonlinear approximate system for the
horizontal velocity components $u, v$ because in this case the variational formulation is
actually a convex optimization problem whose solution minimizes the functional. As
noted in the Introduction, the Blatter-Pattyn approximation is widely used in practice as
an economical and relatively accurate ice sheet model. If desired, the vertical velocity
component $w$ is computed a posteriori by means of the continuity equation.





**Remark #1**: The original formulation (e.g., Pattyn, 2003) also approximates the normal
unit vectors $n_i^{(b2)}$ on the frictional part of the basal boundary $S_{B2}$ by making the small
slope approximation (Dukowicz et al., 2011; Perego et al., 2012). However, this
additional approximation is unnecessary since any computational savings are negligible.

**3.4.2 The Extended Blatter-Pattyn Approximation**

A second form of the Blatter-Pattyn approximation is obtained from the transformed
variational principle (33) by making the assumption,

$$\left\|\frac{\partial w}{\partial x}\right\| \ll \left\|\frac{\partial u}{\partial z}\right\|, \quad \left\|\frac{\partial w}{\partial y}\right\| \ll \left\|\frac{\partial v}{\partial z}\right\|, \tag{38}$$

and therefore neglecting $\partial w/\partial x, \partial w/\partial y$ in the transformed second invariant $\tilde{\dot{\varepsilon}}^2$, or
equivalently, in the strain rate tensor $\tilde{\dot{\varepsilon}}_{ij}$ from (26), consistent with the original small
aspect ratio approximation (Blatter, 1995; Pattyn, 2003; DPL, 2010; Schoof and
Hindmarsh, 2008). This corresponds to setting $\xi = 0, \hat{\xi} = 1$ in the transformed Stokes
model. That is, we neglect vertical velocity gradients but keep the pressure Lagrange
multiplier term. This will be called the extended Blatter-Pattyn approximation (EBP)
because, in contrast to the standard Blatter-Pattyn approximation, all the variables, i.e.,
$u, v, w, \tilde{P}$, are retained. Notably, assumption (38) is equivalent to just setting $w = 0$ in
the second invariant $\tilde{\dot{\varepsilon}}^2$ in the full transformed Stokes model (i.e., with $\xi = 1, \hat{\xi} = 1$). In
other words, the extended BP approximation is obtained by neglecting vertical velocities
everywhere in (33) except where it occurs in the velocity divergence term. This aspect
of the transformed Stokes model will be exploited later to obtain approximations that
improve on Blatter-Pattyn. Thus, the extended Blatter-Pattyn functional is given by
$$\mathcal{A}_{EBP}[u_i, \tilde{P}] = \int_V dV \left[ \frac{4n}{n+1} \eta_0 \left( \dot{\varepsilon}_{BP}^2 \right)^{(1+n)/2n} - \tilde{P} \frac{\partial u_i}{\partial x_i} + \rho g u_{(i)} \frac{\partial z_s}{\partial x_{(i)}} \right]$$
$$+ \frac{1}{2} \int_{S_{B2}} dS \, \beta(x) \left( u_{(i)} u_{(i)} + \varsigma \left( u_{(i)} n_{(i)}^{(b2)} / n_z^{(b2)} \right)^2 \right), \tag{39}$$

where the Blatter-Pattyn second invariant $\dot{\varepsilon}_{BP}^2$ is given by (35). Taking the variation of
the functional, the resulting system of extended Blatter-Pattyn Euler-Lagrange equations
and their boundary conditions is given by



(1) Variation with respect to $u_{(i)}$ yields the horizontal momentum equation:
$$\frac{\partial \tau^{BP}_{(i)j}}{\partial x_j} - \frac{\partial \tilde{P}}{\partial x_{(i)}} - \rho g \frac{\partial z_s}{\partial x_{(i)}} = 0; \quad \left\{ \begin{array}{c} \tau^{BP}_{(i)j} n^{(s)}_j - \tilde{P} n^{(s)}_{(i)} = 0 \ \text{on} \ S_S, \quad u_{(i)} = 0 \ \text{on} \ S_{B1}, \\ \tau^{BP}_{(i)j} n^{(b2)}_j + \beta(x)\left( u_{(i)} + \zeta\left( u_{(k)} n^{(b2)}_{(k)} \big/ n^{(b2)}_z \right) n^{(b2)}_{(i)} \big/ n^{(b2)}_z \right) = 0 \\ \text{on} \ S_{B2}, \end{array} \right. \quad (40)$$

where $\tau^{BP}_{(i)j}$ is given by (37).
(2) Variation with respect to $w$ yields the vertical momentum equation:
$$-\frac{\partial \tilde{P}}{\partial z} = 0; \qquad \tilde{P} n^{(s)}_z = 0 \ \text{on} \ S_S, \qquad (41)$$

(3) Variation with respect to $\tilde{P}$ yields the continuity equation:
$$\frac{\partial w}{\partial z} + \frac{\partial u_{(i)}}{\partial x_{(i)}} = 0; \quad w = 0 \ \text{on} \ S_{B1}, \ \text{or} \ w = -u_{(i)} n^{(b2)}_{(i)} \big/ n^{(b2)}_z \ \text{on} \ S_{B2}. \qquad (42)$$

This appears to be a coupled system for the complete set of variables, $u, v, w, \tilde{P}$, just as in
the transformed Stokes model. However, it is apparent that the vertical momentum
equation system (41) is decoupled and results in $\tilde{P} = 0$, as was already shown in §3.1.
This eliminates pressure from the horizontal momentum equation (40), making it
identical to the standard Blatter-Pattyn system (36). Finally, having obtained the
horizontal velocities from the solution of (40), the continuity equation (42) may be solved
for the vertical velocity component $w$ (but see the comments regarding the discrete case
that follow (43)).

In summary, the extended Blatter-Pattyn model, (40)-(42), is equivalent to the
standard Blatter-Pattyn model, (36), for the horizontal velocities, $u, v$, except that it also
includes two additional equations that determine the pressure $\tilde{P}$ and the vertical velocity
$w$, which are usually ignored in the standard Blatter-Pattyn approximation when only the
horizontal velocity is of interest. Because of this, we distinguish between the *Blatter-*
*Pattyn model* that solves for just the two horizontal velocities (i.e., the standard Blatter-
Pattyn approximation, (36)), and the *Blatter-Pattyn system* that solves for all the variables
(i.e., the extended Blatter-Pattyn approximation, (40)-(42)). It may not be obvious why
we wish to deal with the extended Blatter-Pattyn system since we already know that it is
equivalent to the simpler Blatter-Pattyn model. As it turns out, the Blatter-Pattyn system
is needed for future applications, to be described in §6, because it allows for a dual-model





code and for easy switching between the Blatter-Pattyn and Stokes models, which may be
a useful feature in a general ice sheet code (e.g., ISSM, Larour et al., 2012), and because
it also enables an adaptive hybrid scheme where the cheaper Blatter-Pattyn
approximation is used locally within a Stokes model.

To complete the solution of the Blatter-Pattyn system once pressure $\tilde{P}$ and the

horizontal velocities $u, v$ are available, the continuity equation (42) needs to be solved for
the vertical velocity $w$. The use of the continuity equation to solve for the vertical
velocity $w$ is a novel feature of the Blatter-Pattyn approximation since the continuity
equation is not normally used for this purpose. Using Leibniz's theorem, the continuity
equation may be integrated starting from the bottom to obtain the vertical velocity in
terms of horizontal velocity components, as follows

$$w(u,v) = w_{z=z_b} - \int_{z_b}^{z} \frac{\partial u_{(i)}}{\partial x_{(i)}} dz' = u_{(i)} \frac{\partial z_b}{\partial x_{(i)}} - \int_{z_b}^{z} \frac{\partial u_{(i)}}{\partial x_{(i)}} dz' = -\frac{\partial}{\partial x_{(i)}} \int_{z_b}^{z} u_{(i)} dz' . \qquad (43)$$

Note that we have replaced $w_{z=z_b}$ by $u_{(i)} \partial z_b / \partial x_{(i)}$. This is valid for either of the basal
boundary conditions (9) or (10) (here (10) is in the form given by (14)). When solving
the Blatter-Pattyn system, the right-hand-side is known. However, (43) also works
symbolically when the horizontal velocities $u_{(i)}$ are not yet known, and therefore $w(u,v)$
is a functional of the unknown horizontal velocity distribution.

Thus far, we have only considered continuum results. A discrete finite element

formulation, however, may not be well behaved. The solution of the discretized Stokes
models and the associated Blatter-Pattyn approximations, and the ability to solve for the
vertical velocity as in (43), will depend on the choices made for the grids and for the
finite elements that are to be used. These issues will be discussed next.

**4. Finite Element Discretization**
**4.1 Standard and Transformed Stokes Discretizations**

In practice, both traditional Stokes and Blatter-Pattyn models are discretized using finite
element methods (e.g., Gagliardini et al., 2013; Perego et al., 2012). We follow this
practice except that here the discretization originates from a variational principle. This
has a number of advantages (see §2.3 and DPL, 2010). The following is a brief outline of
the finite element discretization. Additional details about the grid and the associated





discretization are provided in Appendix C. For simplicity, we confine ourselves to two
dimensions with coordinates $(x,z)$ and velocities $(u,w)$. Generalization to three
dimensions should be clear (an example of a three-dimensional grid appropriate for our
purpose is discussed in Appendix C). Further, we present only the simpler case of direct
substitution for the basal boundary conditions in the variational functional, i.e., (15) or
(33). The remarks in this Section apply to both the standard and transformed Stokes
models; for example, the discrete pressure variable $p$ may refer to either the standard
pressure $P$ or the transformed pressure $\tilde{P}$.

Consider an arbitrary grid with a total of $N = n_u + n_w + n_p$ unknown discrete
variables at appropriate nodal locations $1 \le i \le N$, with $n_u$ horizontal velocity variables,
$n_w$ vertical velocity variables, and $n_p$ pressure variables, such that
$U = \left\{ U_1, U_2, \cdots, U_N \right\}^T = \left\{ \left\{ u_1, u_2, \cdots, u_{n_u} \right\}, \left\{ w_1, w_2, \cdots, w_{n_w} \right\}, \left\{ p_1, p_2, \cdots, p_{n_w} \right\} \right\}^T = \left\{ u, w, p \right\}^T$ (44)
is the vector containing all the unknown discrete variables. These are the degrees of
freedom of the model. If using Lagrange multipliers for basal boundary conditions then
discrete variables corresponding to $\lambda_z, \Lambda$ must be added. Variables are expanded in
terms of shape functions $N_i^k(\mathbf{x})$ associated with each nodal variable $i$ in each element
$k$, such that $U^k(\mathbf{x}) = \sum_i U_i N_i^k(\mathbf{x})$ is the spatial variation of all the variables in element
$k$. The summation is over all variable nodes located in element $k$. Shape functions
associated with a given node may differ depending on the variable (i.e., $u, w,$ or $p$).
Substituting into the functional, (15) or (33), integrating and assembling the contributions
of all elements, we obtain a discretized variational functional in terms of the nodal
variable vectors $u, w, p$, as follows
$$\mathcal{A}(u, w, p) = \sum_k \mathcal{A}^k(u, w, p),$$ (45)
where $\mathcal{A}^k(u, w, p)$ is the local functional evaluated by integrating over element $k$. Since
the term in the functional involving the product of pressure and divergence of velocity is
linear in pressure and velocity, and the term responsible for gravity forcing is linear in
velocity, the functional (45) may be written in matrix form as follows
$$\mathcal{A}(u, w, p) = \mathcal{M}(u, w) + p^T \left( M_{UP}^T u + M_{WP}^T w \right) + u^T F_U + w^T F_W,$$ (46)





where the shorthand notation from (44) is used, i.e., $u = \left\{ u_1, u_2, \cdots, u_{n_u} \right\}^T$, etc. Parentheses
indicate a functional dependence on the indicated variables. Comparison with (15) and
(33) indicates that $\mathcal{M}(u, w)$ is a nonlinear positive-definite function of the velocity
components $u, w$, $M_{UP}$, $M_{WP}$ are constant $n_u \times n_p$ and $n_w \times n_p$ matrices, respectively,
arising from the incompressibility constraint in the functional, and $F_U$, $F_W$ are constant
gravity forcing vectors, of dimension $n_u$ and $n_w$, respectively. Note that $F_U = 0, F_W \neq 0$
in the standard Stokes model and $F_U \neq 0, F_W = 0$ in the transformed Stokes model. The
discrete functional $M(u, w)$ differs in the two models but it remains positive-definite in
both, which has important consequences, as will be seen in Appendix D.

Discrete variation of the functional corresponds to partial differentiation with

respect to each of the discrete variables in $U$. Thus, the discrete Euler-Lagrange
equations that correspond to the u-momentum, w-momentum, and continuity equations,
respectively, are given by
$$R(u, w, p) = \begin{bmatrix} R_U(u, w, p) \\ R_W(u, w, p) \\ R_P(u, w) \end{bmatrix} = \begin{bmatrix} \mathcal{M}_U(u, w) + M_{UP}p + F_U \\ \mathcal{M}_W(u, w) + M_{WP}p + F_W \\ M_{UP}^T u + M_{WP}^T w \end{bmatrix} = 0 , \qquad (47)$$

where $R(u, w, p)$ is the residual vector (actually, it is the negative of the usual definition
of the residual) with components $R_U(u, w, p) = \partial \mathcal{A}/\partial u$, $R_W(u, w, p) = \partial \mathcal{A}/\partial w$, and
$R_P(u, w) = \partial \mathcal{A}/\partial p$. The functionals $\mathcal{M}_U(u, w) = \partial \mathcal{M}/\partial u$, $\mathcal{M}_W(u, w) = \partial \mathcal{M}/\partial w$ are
nonlinear vectors of dimension $n_u$ and $n_w$, respectively. Altogether, (47) is a set of $N$
equations for the $N$ unknown discrete variables $U_i$. As explained previously, all
boundary conditions are already included in functional (46), and therefore are also
included in the discrete Euler-Lagrange equations (47).

Since the overall system (47) is nonlinear, it is typically solved using Newton-

Raphson iteration, expressed in matrix notation as follows
$$M(u^K, w^K) \Delta U^{K+1} + R(u^K, w^K, p^K) = 0 , \qquad (48)$$

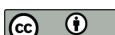
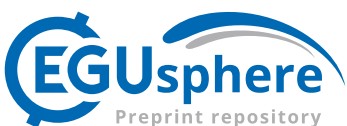

where $K$ is the iteration index, $M(u,w) = \partial^2 \mathcal{A}(U)/\partial U_i \partial U_j$ is a symmetric $N \times N$
Hessian matrix, and $\Delta^{K+1}$ is the column vector given by
$$\Delta U^{K+1} = \left[ u^{K+1} - u^K, w^{K+1} - w^K, p^{K+1} - p^K \right]^T .$$

Given $U_i^K$ from the previous iteration, (48) is a linear matrix equation that is solved for
the $N$ new variables $U_i^{K+1}$ at each iteration. In view of (46) and (47), the Hessian matrix
$M(u,w)$ may be decomposed into several submatrices, as follows
$$M(u,w) = \begin{bmatrix} M_{UU}(u,w) & M_{UW}(u,w) & M_{UP} \\ M_{UW}^T(u,w) & M_{WW}(u,w) & M_{WP} \\ M_{UP}^T & M_{WP}^T & 0 \end{bmatrix} . \tag{49}$$

Submatrices $M_{UW}(u,w) = \partial^2 \mathcal{M}/\partial u \partial w$, etc., depend nonlinearly on $u, w$. Thus,
$M_{UU}(u,w), M_{WW}(u,w)$ are square $n_u \times n_u$, $n_w \times n_w$ matrices, respectively, while
$M_{UW}(u,w)$ is a rectangular $n_u \times n_w$ matrix since $n_u, n_w$ may not be equal. As noted
earlier, $M_{WP}$ is a $n_w \times n_p$ matrix and therefore not square unless $n_p = n_w$. Additionally,
$M_{UU}(u,w)$ and $M_{WW}(u,w)$ are symmetric by definition.

**4.2 Blatter-Pattyn Discretizations**

For completeness, we express the Blatter-Pattyn approximations from §3.4 in matrix
form, as follows
(1) The standard Blatter-Pattyn model from §3.4.1 takes the simple form
$$R^{BP}(u) = \mathcal{M}_U(u,0) + F_U = 0 , \tag{50}$$

with the corresponding Newton-Raphson iteration given by
$$M^{BP}(u^K)\Delta u^{K+1} + R^{BP}(u^K) = 0 , \tag{51}$$

where the Blatter-Pattyn Hessian matrix is $M^{BP}(u) = M_{UU}(u,0)$.





(2) The extended Blatter-Pattyn approximation from §3.4.2 becomes
$$R^{EBP}\left(u,w,p\right)=\begin{bmatrix} \mathcal{M}_U\left(u,0\right)+M_{UP}p+F_U \\ M_{WP}p \\ M_{UP}^T u+M_{WP}^T w \end{bmatrix}=0 \ , \tag{52}$$

and the Newton-Raphson iteration is given by
$$M^{EBP}\left(u^K\right)\Delta U^{K+1}+R^{EBP}\left(u^K,w^K,p^K\right)=0 \ , \tag{53}$$

where the associated Hessian matrix is
$$M^{EBP}\left(u\right)=\begin{bmatrix} M_{UU}\left(u,0\right) & 0 & M_{UP} \\ 0 & 0 & M_{WP} \\ M_{UP}^T & M_{WP}^T & 0 \end{bmatrix} . \tag{54}$$


**4.3 Solvability Issues**

We now consider the solution of the three linear matrix problems (48), (51), (53). While
there is no issue in the continuous case, there may be problems in the discrete case
depending on the choice of the grid and the finite elements, as noted earlier.

**4.3.1 Solvability of the Standard and Transformed Stokes Models**

The Hessian matrix in the standard and transformed Stokes cases, (49), has the form
$$M\left(u,w\right)=\begin{bmatrix} A & B \\ B^T & 0 \end{bmatrix} , \tag{55}$$

where
$$A=A^T=\begin{bmatrix} M_{UU}\left(u,w\right) & M_{UW}\left(u,w\right) \\ M_{UW}^T\left(u,w\right) & M_{WW}\left(u,w\right) \end{bmatrix} , \quad B=\begin{bmatrix} M_{UP} \\ M_{WP} \end{bmatrix} , \quad B^T=\begin{bmatrix} M_{UP}^T & M_{WP}^T \end{bmatrix} .$$

The general form (55) is characteristic of Stokes-type problems, or more generally, of
constrained minimization problems using Lagrange multipliers. In finite element
terminology these are "mixed" problems, meaning that velocity components and the
pressure occupy different finite element spaces, or else they are "saddle point" problems
since the Hessian matrix $M\left(u,w\right)$ is symmetric but indefinite, with both positive and



negative eigenvalues. This can give rise to solution instabilities. To avoid this, elements
that are to be used must satisfy the so-called inf-sup or LBB condition constraining the
matrix $B$ in (55). There is a very large literature on the subject, e.g., Elman et al. (2014).
Testing for stability is not trivial. Both the standard and transformed Stokes models are
subject to these issues and in general must use inf-sup-stable finite elements. An
example of an inf-sup stable element is the popular second-order Taylor-Hood P2-P1
element with piecewise quadratic velocity and linear pressure (Hood and Taylor, 1973).
Both the standard and transformed Stokes models are stable using the Taylor-Hood
element. Some results involving the Taylor-Hood element are shown in Fig. 13 for Test
B, one of the test problems described in Appendix B that corresponds to Exp. B from the
ISMIP–HOM model intercomparison (Pattyn et al., 2008).

**4.3.2 Solvability of the Standard Blatter-Pattyn Model**

The standard Blatter-Pattyn approximation is not subject to these stability issues since
pressure, the Lagrangian multiplier, is absent in (51). As a result, the standard Blatter-
Pattyn variational formulation (34) is actually a well-behaved and stable positive-definite
minimization or optimization problem.

**4.3.3 Solvability of the Extended Blatter-Pattyn Model**

We noted earlier that the transformed Stokes model works well using the Taylor-Hood
element in Test B. Since the extended Blatter-Pattyn model has the same structure as the
transformed full-Stokes model and yields the same solution for horizontal velocity as the
standard Blatter-Pattyn model, one might expect its discrete implementation to behave
well. However, the extended Blatter-Pattyn model fails badly in this problem, with
nonsensical results for the vertical velocity. This may be because there is an additional
requirement for the stability of a Stokes-type problem that is not met in this case, namely,
the matrix $A$ in (55) must be elliptic on the whole $u, w$ space (Auricchio et al., 2017).
However, there is a much simpler explanation. Consider the vertical momentum
equation, the second of the extended Blatter-Pattyn model equations from (52). As is
seen in §3.4.2 or from the second of the equations in (52) in the extended Blatter-Pattyn
approximation, this equation is a decoupled linear system for the pressure. Since the
equation involves the $M_{WP}$ matrix, we have a decoupled set of $n_w$ equations that needs to





be solved for the $n_p$ pressure variables. This is not possible unless the matrix $M_{WP}$ is
square. For the same reason, the third of the equations in (52) cannot be solved for $w$
unless matrix $M_{WP}^T$ is invertible. In other words, the extended Blatter-Pattyn model (52)
only works when $n_w = n_p$, which is not the case in a Taylor-Hood discretization. This is
because in finite element discretizations of Stokes problems, the pressure approximation
is typically one degree lower than the velocity approximation, which leads to fewer
pressure variables than velocity variables. In the case of the Taylor-Hood element, the
difference is very large and we have $n_w \gg n_p$ (see §7 for more details). This means that
in the extended Blatter-Pattyn model vertical velocity is greatly underdetermined, which
accounts the problem in the Taylor-Hood calculation. This problem also manifests itself
in Taylor-Hood discretizations of Stokes models but to a much lesser extent. For
example, mass is poorly conserved in the Taylor-Hood discretization of the standard
Stokes model (Boffi et al., 2012). In the transformed Stokes case there tend to be
velocity oscillations that tend to go away when using a grid in which $n_p = n_w$ (see Fig. 13,
Panels E and F).

**4.3.4 The Solvability Condition**

Summarizing, the extended Blatter-Pattyn approximation is problematic unless we have
$$n_p = n_w. \tag{56}$$

In addition, the resulting square matrix $M_{WP}$ must be non-singular, which we assume to be
the case for a reasonable finite element discretization. This makes it possible to solve for
the pressure in the extended Blatter-Pattyn system (52) because $M_{WP}$ is square and
invertible. We henceforth refer to (56), together with non-singularity, as the solvability
condition for the pressure. This is a characteristic or a property associated with the
discrete grid and the boundary conditions. In Appendix C, we consider several grids that
exhibit this property. The specific solvability condition given by (56) applies when direct
substitution is used for basal boundary conditions. The number of unknown pressures $n_p$
must be augmented if Lagrange multipliers are used and (56) becomes $n_p + \lambda_z + \Lambda = n_w$
(See Appendix C, §C2).



The solvability condition has an additional implication. If matrix $M_{WP}$ is square
and invertible due to (56), then its transpose $M_{WP}^{T}$ is also square and invertible. This
implies that the continuity equation in (47) and (52), that is,

$$M_{UP}^{T}u + M_{WP}^{T}w = 0 \,,\qquad(57)$$

is solvable for the vertical velocity $w$ in terms of the horizontal velocities, as follows

$$w(u) = -M_{WP}^{-T}M_{UP}^{T}u \,,\qquad(58)$$

where the matrix $M_{WP}^{-T}$ is defined by

$$M_{WP}^{-T} = \left(M_{WP}^{T}\right)^{-1} = \left(M_{WP}^{-1}\right)^{T} \,.\qquad(59)$$

Note that (58) is the discrete form of equation (43). Thus, since the invertibility of $M_{WP}$
implies the invertibility of $M_{WP}^{T}$, the solvability condition (56) implies the solvability of
the continuity equation (58), and vice-versa. As we shall see, this property is not just a
useful property but it is necessary for the new Stokes approximations that improve on the
Blatter-Pattyn approximation, as discussed in §6.2.

Perhaps the main reason for the importance of the solvability condition is
demonstrated in Appendix D. Appendix D shows that a variational principle that
complies with the solvability condition is equivalent to an optimization or minimization
problem, which is sufficient for the stability of the corresponding Stokes model. Thus,
for example, the extended Blatter-Pattyn model fails with a Taylor-Hood P2-P1 grid,
which does not satisfy the solvability condition, but works well with a variant, the P2-E1
grid, shown in Fig. 13A, that does satisfy the solvability condition. Several finite
elements that satisfy the condition are presented in Appendix C. One particular element,
the P1-E0 element, is particularly useful for use with the transformed Stokes model
because the solvability condition is satisfied locally, i.e., along individual vertical grid
lines, as shown in Appendix C. This element is used in most of the 2D test problems
featured here.

**5. Comparison of the Standard and Transformed Stokes Models**

To compare the standard and transformed Stokes models we use two 2D test problems,
namely, Exp. B from the ISMIP-HOM benchmark (Pattyn et al, 2008), and Exp. D*, a
modified version of Exp. D from the ISMIP-HOM suite. A description of these tests is



provided in Appendix B, where they are referred to as Test B and Test D[*]. Test B
involves no-slip boundary conditions on a sinusoidal bed, and Test D[*] evaluates sliding
of the ice sheet along a flat bed in the presence of sinusoidal friction. The tests are
discretized using P1-E0 elements on a regular grid composed of $n$ quadrilaterals in the
$x$-direction and $m$ quadrilaterals in the $z$-direction, with each quadrilateral divided into
two triangles as illustrated in Figs. C3 and described in Appendix D. The results
presented in this Section are for a relatively coarse 40x40 grid, i.e., $m = n = 40$, except
when we consider the convergence of the models with grid refinement.

**5.1 Convergence of Solutions with Grid Refinement**

We first look at the convergence of the transformed and standard Stokes models as the
grid is refined in Fig. 3. In particular, we look at the convergence of ice transport through
a vertical cross section of the ice sheet at $x = L$. The ice transport $T$ is defined by
$$T = \int_{z_b}^{z_s} u(L,z)\, dz \,, \tag{60}$$

where the vertical profile $u(L,z)$ is plotted in Fig. 4 for several cases at the 40x40
resolution. Fig. 3 plots the absolute value of the transport error $E = \left\| T - T_R \right\|$ as a function
of the resolution $r$, where $r$ is the number of quadrilaterals in either direction (since
$r = m = n$) and $T_R$ is the converged value of the transport obtained by Richardson
extrapolation using the two highest resolution values. The transport is evaluated at
various resolutions $r = 5, 10, 15, 20, 30, 40$, and plotted at two domain lengths, $L = 5$ and
10 km. Trying to estimate the rate of convergence in this way is highly uncertain, as
discussed in §7, but estimating the error is a more reasonable thing to do. Both models
are consistent with second order convergence, as expected from the use of linear
elements, but they behave quite differently in the two test problems. The transformed
Stokes model (TS) is some two orders of magnitude more accurate at all resolutions than
the standard Stokes model (SS) in Test B calculations although they start from the same
initial conditions. However, the accuracy of the two models is quite similar in Test D[*]
calculations, with the SS error actually somewhat smaller than the TS error. This is
confirmed when we compare the details of the $u$-velocity solutions in Figs. 4 and 5 at the
40x40 resolution. The TS and SS profiles differ noticeably from each other but are quite
similar in the Test D[*] case. However, the standard and transformed Stokes models do
eventually converge to the same solution.





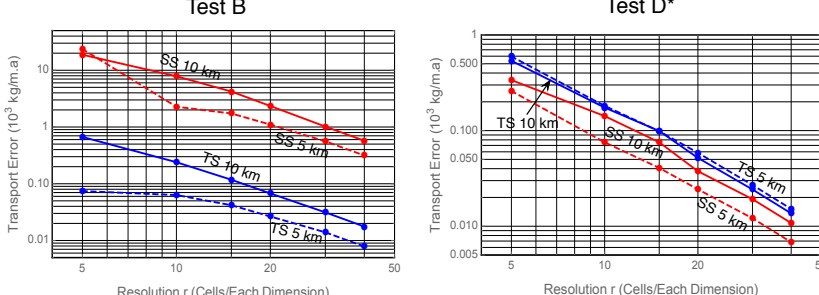


**Figure 3.** Convergence of ice transport in Tests B and D* with grid refinement.

Transformed Stokes plots are in blue and standard Stokes plots are in red.

**5.2 The Vertical Profile of Solutions**

Fig. 4 shows the vertical profiles of the horizontal velocity $u$ at $x = L$ for the 40x40
resolution in the transformed and standard Stokes models. There is a noticeable
difference in the two profiles in Test B, as is to be expected from Fig. 3 results where we
see that the SS calculation is not yet as well converged as the TS case at this resolution.
Also shown in Fig. 4 are profiles from the two frictional sliding problems, Tests D and
D*. The Test D profile, i.e., Exp. D from the ISMIP-HOM benchmark, is almost
vertically constant, indicating that the originally chosen value for basal friction is too
small, i.e., more appropriate for a shallow-shelf approximation. This motivated the
modification of Test D to Test D*, as described in Appendix B. In contrast to the Test B
case, the standard and transformed frictional Test D and D* plots cannot be visually
distinguished from each other, as might be expected from the similar error convergence
for the Test D* results in Fig. 3.

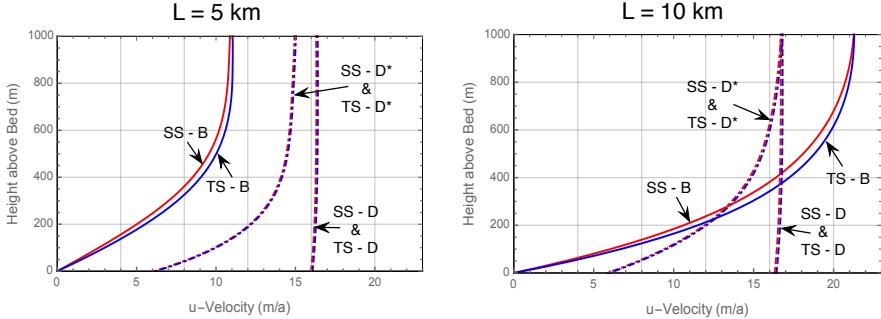


**Figure 4.** The u-velocity profile at location $x = L$ as a function of height from the bed.





### 5.3 The Upper Surface Horizontal Velocity

Figs. 5 and 6 show the u-velocity at the upper surface at the 40x40 resolution for Tests B and D[*], respectively. This is the basic benchmark used in ISMIP-HOM to compare the different ice sheet models. Here we compare four cases: the standard Stokes model (SS), the transformed Stokes model (TS), the Blatter-Pattyn (BP) model, and for reference, the very high resolution full-Stokes calculation "oga1" presented in the ISMIP-HOM paper (SS-HR). The SS-HR calculation is also available independently in Gagliardini and Zwinger (2008). Results are presented for two domain lengths, $L = 5$ km and 10 km, to observe the behavior of the SS and TS models in the aspect ratio range where the Blatter-Pattyn model begins to fail.

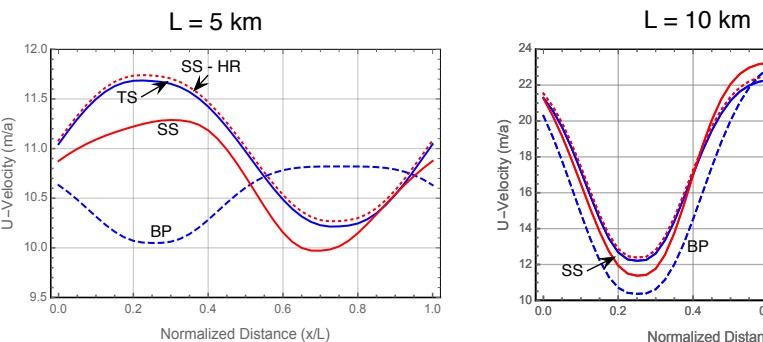

**Figure 5.** Upper surface u-velocity, $u(x, z_s)$ - Test B, No-slip boundary conditions.

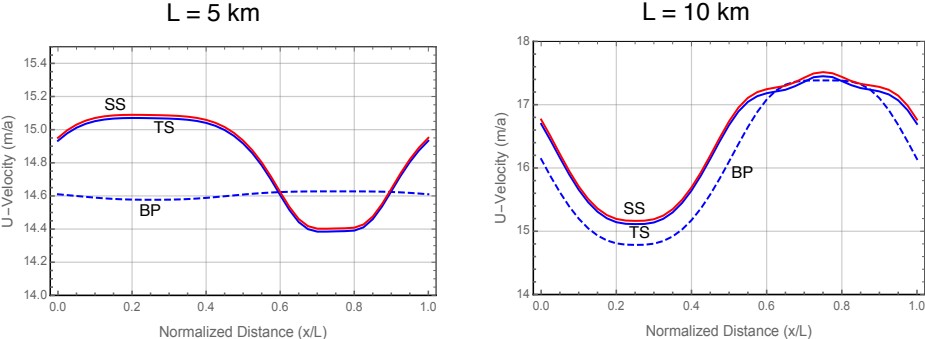

**Figure 6.** Upper surface u-velocity, $u(x, z_s)$ - Test D[*], Modified frictional sliding case.

The TS and the SS-HR plots in Fig. 5 lie on top of one another (the SS-HR plot (dotted) has been slightly offset upward for clarity), indicating that the transformed





Stokes model is already fully converged, and confirming that the standard and
transformed Stokes models do indeed converge to the correct Stokes solution. We again
observe that the SS results are not yet converged in Test B at this resolution, particularly
at $L = 5$ km. As also seen in the ISMIP-HOM benchmark paper, the Blatter-Pattyn
calculation (BP) shows large deviations from the Stokes results, especially so at $L = 5$
km where surface velocity is entirely out of phase with the Stokes results. Test D*
frictional sliding results follow a similar pattern in Fig. 6. Since convergence of the SS
and TS models is very similar in the frictional case, the SS and TS plots overlie one
another (the SS plot has been slightly offset upward for visibility), confirming that the
two Stokes models converge to the same solution. As was seen in Test B, the Blatter-
Pattyn error is quite large at $L = 10$ km, and dramatically so at $L = 5$ km.

**6. Some Applications of the Transformed Stokes Model**
**6.1 Adaptive Switching between Stokes and Blatter-Pattyn Models**

One way of reducing the cost of a full Stokes calculation is to use it adaptively with a
cheaper approximate model in a given problem. That is, one may use the cheaper model
in those parts of a problem where it is accurate, and the more expensive full Stokes model
where the approximate model loses accuracy. One example of such an adaptive approach
is the tiling method by Seroussi et al. (2012). However, there are drawbacks to such
methods, such as the difficulty of incorporating two or more presumably quite different
models into a single model, and the additional complexity of a transition zone in order to
couple the disparate models.

Using the transformed Stokes model in such an adaptive role is attractive because

it may be switched between the Stokes and Blatter-Pattyn cases simply by switching the
parameter $\xi \in \{0,1\}$ between its two values. To avoid complications and more difficult
programming it is essential that both the Stokes and the Blatter-Pattyn parts of the code
have the same number of discrete variables. This implies that the extended Blatter-Pattyn
approximation $(\hat{\xi} = 1)$ must be used, which therefore implies the use of a grid that
satisfies the solvability condition for reasons discussed in §4 and Appendix C. To do
this, we will discretize using the P1-E0 element. To demonstrate the idea of adaptive
switching with a transformed Stokes model, we introduce a new test problem, Test O,
described in Appendix B and illustrated in Fig. B1. This consists of an inclined ice slab
whose movement is obstructed by a thin obstacle protruding 20% of the ice depth up



from the bed. No-slip boundary conditions are applied along the bed and on the obstacle
itself. Because of the localized nature of the obstacle, the conditions for the Blatter-
Pattyn approximation to be valid, (38), must fail near the obstacle and therefore the full
Stokes model is needed for good accuracy, at least locally.

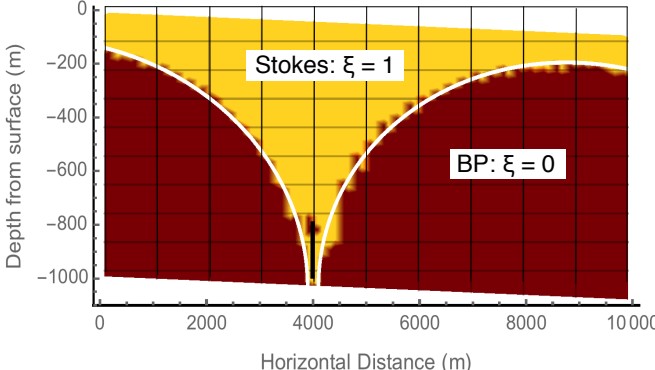


**Figure 7.** Mask function (white curve, $z = F_M(x)$) to indicate where the Stokes and BP
models are activated in the adaptive hybrid 20% obstacle test problem. The dark brown
region delineates the region where $|\partial w/\partial x| \leq 0.1|\partial u/\partial z|$ in a Blatter-Pattyn calculation.

To implement this idea, we first use a Blatter-Pattyn calculation to outline regions
where $|\partial w/\partial x| \leq 0.1|\partial u/\partial z|$, approximately localizing where the Blatter-Pattyn
approximation is valid. This determines a mask function $z = F_M(x)$, illustrated in Fig. 7
by the two white curves, that specifies where the two models must be used. Defining the
centroid of a triangular element by $(x_C, z_C)$, the code makes he following selection in
each element,
$$z_C \leq F_M(x_C) \quad \Rightarrow \quad \text{Set } \xi = 0, \quad \text{i.e., the Blatter-Pattyn region,}$$
$$z_C > F_M(x_C) \quad \Rightarrow \quad \text{Set } \xi = 1, \quad \text{i.e., the Stokes region.}$$

Somewhat counterintuitively, the Stokes region occupies the upper part of the domain in
Fig. 7 and includes the obstacle, while the Blatter-Pattyn region occupies much of the
bottom part of the domain. It would be possible to introduce a transition zone, e.g.,
$0 \leq \xi(x,z) \leq 1$, but this was not deemed necessary and it was not done in the present
calculation.





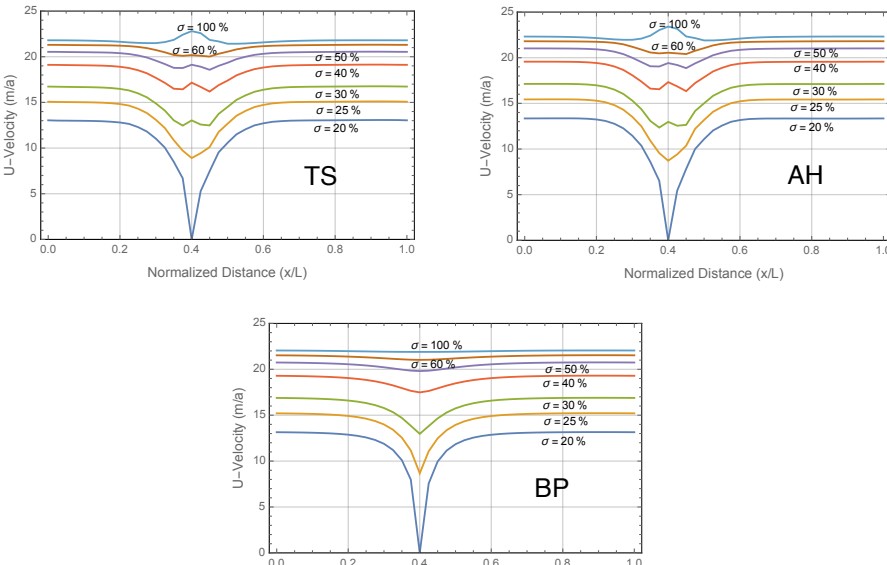


**Figure 8.** Comparing results for the Transformed Stokes (TS, i.e., the exact Stokes),

the Adaptive-Hybrid (AH), and the Blatter-Pattyn (BP) models for Test O.


The Adaptive-Hybrid results are shown in Fig. 8, which shows curves of the

horizontal velocity $u$ at seven different vertical positions specified as a percentage of the
distance between top and bottom, such that $\sigma = 100\%$ is at the top surface. The top right
panel shows the results for the adaptive-hybrid model. For comparison, the top left panel
and the bottom panel show results for the full Stokes and the Blatter-Pattyn calculations,
respectively. All calculations are at the 40x40 resolution. The Adaptive-Hybrid results
are very similar to the full Stokes results, reproducing most features of the velocity
profiles, including the velocity bump at the top surface, indicating that even the top
surface feels the presence of the obstacle. The Blatter-Pattyn results are much less
accurate; they completely miss the details of the flow near the obstacle. We also
calculate a measure of the error relative to the transformed Stokes results, the overall
RMS u-Error, defined as follows
$$\text{RMS u-Error} = \sqrt{\sum_{k=1}^{n_u} \left( u_k - u_k^{TS} \right)^2 \Big/ n_u} \ , \qquad (61)$$
where $u_k^{TS}$ is the transformed Stokes horizontal velocity discrete variable. The overall
RMS u-Error in the Blatter-Pattyn case is 0.493 m/a while the Adaptive-Hybrid error is
0.440 m/a, smaller in the Blatter-Pattyn case, as expected, but the difference is not big



and not as striking as the visual differences in Fig. 8. Nevertheless, the adaptive-hybrid
method can be judged successful by the results presented in Fig. 8 alone. Unfortunately,
a reasonable estimate of the computational cost savings cannot be made because of the
small-scale nature of these calculations that were carried out on a personal computer.

**6.2. Two Stokes Approximations Beyond Blatter-Pattyn**

As shown in §3.4, simply setting $w = 0$ in the second invariant $\tilde{\varepsilon}^2$ in the transformed
functional $\tilde{\mathcal{A}}$, given by (28) and (33), respectively, results in the Blatter-Pattyn system of
equations. This suggests that approximating the vertical velocity $w$ in the transformed
functional would be a good way to create approximations that improve on the Blatter-
Pattyn approximation since providing no information at all, i.e., $w = 0$, already produces
an excellent approximation. We will look at only two such methods in this Section even
though many other variations are possible. The first method, to be called the BP+
approximation, improves the Blatter-Pattyn approximation simply by using a lagged
value of the vertical velocity in the functional (33). It is implemented using a
combination of Newton and Picard iterations such that at each Newton iteration the
variational functional is evaluated using the known vertical velocity $w^K$ from the
previous iteration, where $K$ is the iteration index. The vertical velocity, $w^K = w(u^K)$, is
obtained by using (58) together with a grid that is consistent with an invertible continuity
equation, such as the P1-E0 grid from Appendix C. The second method, to be called the
Dual-Grid approximation, approximates the transformed Stokes model by discretizing the
continuity equation on a coarser grid. Since vertical velocity $w$ is to be determined by
inverting the continuity equation, this has the effect of approximating the vertical velocity
while at the same time reducing the number of pressure and vertical velocity variables.
The degree of grid coarsening determines the accuracy of the resulting approximation.

**6.2.1 An Improved Blatter-Pattyn or BP+ Approximation**

To prepare, we introduce a pair of 2D variational quasi-functionals, $\tilde{\mathcal{A}}_{PS1}[u, w]$ and
$\tilde{\mathcal{A}}_{PS2}[\tilde{P}]$. Noting that $\tilde{P} = 0$ in the Blatter-Pattyn approximation, we drop the pressure
term from the transformed functional (33) and define a new functional,



$$\tilde{\mathcal{A}}_{PS1}[u,w] = \int_V dV \left[ \frac{4n}{n+1} \eta_0 \left( \tilde{\varepsilon}^2 \right)^{(1+n)/2n} + \rho g u \frac{\partial z_s}{\partial x} \right]$$

$$+ \frac{1}{2} \int_{S_{B2}} dS\, \beta(x) \left( u^2 + \zeta \left( u\, n_x^{(b2)} \big/ n_z^{(b2)} \right)^2 \right), \qquad (62)$$

where

$$\tilde{\varepsilon}^2 = \left( \frac{\partial u}{\partial x} \right)^2 + \frac{1}{4} \left( \frac{\partial u}{\partial z} + \frac{\partial w}{\partial x} \right)^2 . \qquad (63)$$

Since the continuity equation has been eliminated, we introduce incompressibility
separately by defining another functional,

$$\tilde{\mathcal{A}}_{PS2}[p] = \int_V dV\, p \left( \frac{\partial u}{\partial x} + \frac{\partial w}{\partial z} \right) . \qquad (64)$$

Since direct substitution is used for boundary conditions, then (9) and (14) are the
appropriate basal boundary conditions needed to specify $w$ in (64); no boundary
condition is required for the pressure. Here we are effectively viewing the pressure $p$ as
a "test function" in the finite element sense. This gives us great flexibility to create
elements that satisfy the solvability condition (56). In a triangulation, for example, some
pressures may be assigned to every two triangles, as in a P1-E0 grid, while others may be
assigned to a single triangle to achieve an equal number of pressure and vertical velocity
variables.

The discrete variation of $\tilde{\mathcal{A}}_{PS1}[u,w]$ with respect to $u$, results in a set of $n_u$ Euler-
Lagrange equations,

$$\hat{R}_U(u,w) = \frac{\partial \tilde{\mathcal{A}}_{PS1}(u,w)}{\partial u} = M_U(u,w) + F_U = 0 . \qquad (65)$$

This may be recognized as the standard Blatter-Pattyn model, (50), when $w = 0$. The
discrete variation of $\tilde{\mathcal{A}}_{PS2}[p]$ with respect to $p$, results in the continuity equation, (57),

$$\hat{R}_P(u,w) = \frac{\partial \tilde{\mathcal{A}}_{PS2}(p)}{\partial p} = M_{UP}^T u + M_{WP}^T w = 0 . \qquad (66)$$

These two systems are now combined to form the BP+ approximation, as follows

$$\hat{R}(u,w) = \left[ \hat{R}_U(u,w), \hat{R}_P(u,w) \right]^T = 0 . \qquad (67)$$

This is a single system of $n_u + n_p$ equations to determine the $n_u + n_w$ discrete velocities
$u, w$, implying that (67) is viable only on grids satisfying the solvability condition,



$n_p = n_w$. Just as in the standard Blatter-Pattyn approximation in §3.4.1, the vertical
momentum equation is missing, but instead of neglecting $w$, the vertical velocity is now
obtained consistently from the continuity equation.

There are two ways of solving the BP+ system (67), as follows

(1) BP+, Newton/Picard iteration version:

If $w = \hat{w}(x_i)$ is some arbitrary specified function of position, then (65) becomes a

nonlinear set of $n_u$ equations that may be solved for the horizontal velocity $u$ using
Newton iteration, as follows
$$\hat{M}_{UU}\left(u^K, \hat{w}\right)\Delta u + \hat{R}_U\left(u^K, \hat{w}\right) = 0 , \qquad (68)$$

where $\hat{M}_{UU}\left(u, \hat{w}\right) = \partial \mathcal{M}_U\left(u, \hat{w}\right)/\partial u$, $\Delta u = u^{K+1} - u^K$, and $K$ is the iteration index. In
particular, if we choose $\hat{w} = w^K$, where $w^K$ is the horizontal velocity from the previous
iteration (i.e., $w^K = w\left(u^K\right)$ from (58), where $u^K$ is the horizontal velocity from the
previous iteration), we obtain the following Picard iteration:

Starting from $K = 0$, *c*hoose an initial guess, $u^0 \neq 0$,

Do: $\mathrm{w}^K = w\left(u^K\right) = M_{PW}^{-1} M_{PU} u^K$,

Solve $\hat{M}_{UU}\left(u^K, \mathrm{w}^K\right)\Delta u + \hat{R}_U\left(u^K, \mathrm{w}^K\right) = 0,$        (69)

$u^{K+1} = u^K + \Delta u,$

$K = K + 1,$

         Repeat until convergence.

The advantage of this method is that iteration is rapid since each iteration step is
equivalent to the short Newton step of the standard Blatter-Pattyn model, (36). On the
other hand, as a Picard iteration, its convergence is expected to be only linear.

(2) BP+, Quasi-variational, Newton iteration version:

Although a variational principle does not exist, it is still possible to make use of

Newton-Raphson iteration to obtain second order convergence. To do this, we treat (67)
as a single multidimensional nonlinear system and solve it using Newton-Raphson
iteration, as follows



$$\left[ \begin{array}{cc} \hat{M}_{UU}\left(u^{K},w^{K}\right) & \hat{M}_{UW}\left(u^{K},w^{K}\right) \\ M_{PU} & M_{PW} \end{array} \right] \left[ \begin{array}{c} \Delta u \\ \Delta w \end{array} \right] + \left[ \begin{array}{c} \hat{R}_{U}\left(u^{K},w^{K}\right) \\ \hat{R}_{P}\left(u^{K},w^{K}\right) \end{array} \right] = 0 \;, \qquad (70)$$


where $\hat{M}_{UU}\left(u,w\right)=\partial\hat{R}_{U}\left(u,w\right)/\partial u$ and $\hat{M}_{UW}\left(u,w\right)=\partial\hat{R}_{U}\left(u,w\right)/\partial w$. The convergence is
quadratic once in the basin of attraction but each iteration is more expensive than in the
Picard version because the linear system (70) is approximately double the size of the one
in (69). It remains to be seen which version proves to be preferable in practice.

Both BP+ versions converge to the same solution. Fig. 9 compares the upper
surface u-velocity from the improved Blatter-Pattyn (BP+) approximation to the standard
Blatter-Pattyn approximation and to a reference exact Stokes calculation. The RMS u-
Error of the BP+ approximation relative to the exact Stokes case is shown in Fig. 12. The
BP+ approximation is noticeably more accurate than the BP approximation, especially so
in the $L=5$ km case where the Blatter-Pattyn solution bears no resemblance to the
correct solution while the BP+ approximation retains very good accuracy. This is
confirmed by the RMS u-Error results in Fig. 12.

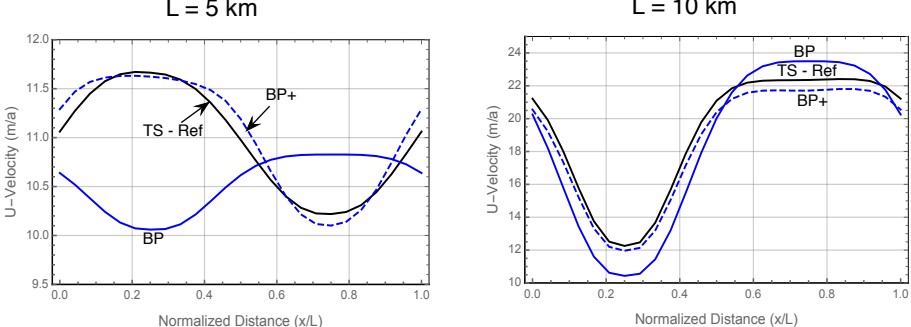


**Figure 9.** Comparing Approximations. Test B, Upper surface u-velocity.
TS-Ref: Transformed Stokes; BP: Blatter-Pattyn; BP+: Improved Blatter-Pattyn.

Resolution: 24x24.


The two versions depend either on solving the continuity equation to obtain
$w=w\left(u\right)$, or the use of a grid that incorporates such a solvable continuity equation.
Solution of the continuity equation to obtain $w$ may already be available for the purpose
of temperature advection in production code packages that either incorporate or are based
on the Blatter-Pattyn approximation. Thus, these new approximations, and particularly
the Newton/Picard version, may be especially attractive for use in such codes since they



substantially improve the accuracy of the basic Blatter-Pattyn model, as seen in Fig. 9, at
little or no additional cost.

**6.2.2 A "Dual-Grid" Transformed Stokes Approximation**

In §6.2.1, the BP+ approximation was based on directly approximating or lagging the
vertical velocity $w$ in the second invariant $\tilde{\varepsilon}^2$ in the transformed functional $\tilde{\mathcal{A}}$. Here we
take a different approach and instead approximate the continuity equation in the
transformed Stokes model, which indirectly approximates $w$. To do this we discretize
the continuity equation on a grid that is coarser than the one used for the momentum
equations and then interpolate the vertical velocity to the appropriate locations on the
finer grid. This reduces the number of unknown variables in the problem, making it
cheaper to solve but hopefully without much loss of accuracy. As described in Appendix
B, our test problem grids are logically rectangular, divided into $n$ cells horizontally and
$m$ cells vertically, thus allowing considerable freedom to specify the coarse grid. The
coarse grid is constructed by dividing the fine grid into $s$ equal segments in each
direction. This presupposes that the integers $n$ and $m$ are each divisible by $s$, such that
there are $s^2$ coarse cells in total, with each coarse cell containing $nm/s^2$ fine cells. The
primary grid (i.e., the fine grid) was chosen to have $n = m = 24$, resulting in a reference
$24 \times 24$ fine grid, so as to maximize the number of different coarse grids that may be
used for this test. Coarse grids were constructed using $s = 2,3,4,6$, and this resulted in
fine/coarse grid combinations labeled by $24 \times 12, 24 \times 8, 24 \times 6, 24 \times 4$, respectively.
Similar to a P1-E0 fine grid, coarse grid vertical velocities $w$ are located at vertices and
pressures at vertical edges. Fig. 10 illustrates the case of a single coarse and four fine
quadrilateral cells for a grid fragment with $n = m = 2$ and $s = 1$. For the Test B problem,
using direct substitution for basal boundary conditions, there will be $nm$ u-variables and
$nm/s^2$ w- and p-variables each, for a total of $nm\left(1 + 2/s^2\right)$ unknown variables,
considerably fewer than the $3nm$ variables in the full resolution (i.e., fine grid) case,
depending on the value of $s$. The coarse grid terms in the functional that are affected,
$\tilde{P}\left(\partial u/\partial x + \partial w/\partial z\right)$ and $\partial w/\partial x$, are computed using coarse grid variables and
interpolated to the fine grid. We will consider two versions of the approximation
depending on how the coarse grid terms are calculated and distributed on the fine grid.





(1) Approximation A, Bilinear interpolation:

Referring to Fig. 10, the four velocities at the vertices of the coarse grid

quadrilateral, i.e., $u_1, u_3, u_7, u_9$ and $w_1, w_2, w_3, w_4$, are used to obtain $u, w$ at the remaining
five vertices of the fine grid by means of bilinear interpolation. Thus, the five velocities
$u_2, u_4, u_5, u_6, u_8$ are obtained in terms of vertex velocities $u_1, u_3, u_7, u_9$, and similarly for the
$w$ velocities. The resulting complete set of fine grid variables, interpolated from coarse
grid variables, are used calculate the divergence $D = (\partial u/\partial x + \partial w/\partial z)$ and the quantity
$\partial w/\partial x$ in each of the eight triangular elements $t_1, t_2, \cdots, t_8$ of the fine grid. Coarse grid
pressures $\tilde{P}_1, \tilde{P}_2$ are associated with the coarse grid triangles $T_1, T_2$. The products $\tilde{P}_1 D$ in
elements $t_1, t_2, t_3, t_5$ and $\tilde{P}_2 D$ in elements $t_4, t_6, t_7, t_8$ are then accumulated over the entire
grid to obtain $\tilde{P}(\partial u/\partial x + \partial w/\partial z)$ for use in the transformed functional $\tilde{\mathcal{A}}$. Similarly, the
quantity $\partial w/\partial x$ is computed in the fine grid elements from coarse grid variables for use
in the second invariant $\tilde{\dot{\varepsilon}}^2$.

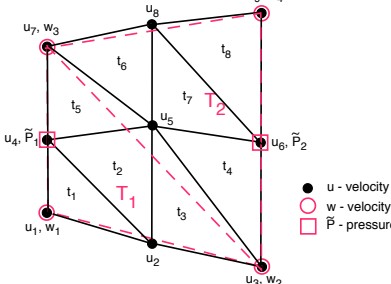



**Figure 10.** A Sample of a Coarse/Fine P1-E0 Grid for the Dual-Grid Approximation.

Resolution: $n = m = 2, s = 1$. Coarse grid is in red, fine grid in black.


(2) Approximation B, Linear interpolation:

In this version, the three velocities at the vertices of the two coarse grid triangles

$T_1$ and $T_2$, i.e., $u_1, u_3, u_7$ and $w_1, w_2, w_3$ in $T_1$, and $u_7, u_3, u_9$ and $w_3, w_2, w_4$ in $T_2$,
approximate the divergence $D = (\partial u/\partial x + \partial w/\partial z)$ and the quantity $\partial w/\partial x$ as constant
values in the two coarse triangles. The constant quantities $\tilde{P}_1 D$, $\tilde{P}_2 D$ are then
accumulated over the entire grid. The constant quantity $\partial w/\partial x$ in each coarse triangle is





973   then distributed to each of the eight fine grid elements $t_1, t_2, \cdots, t_8$ depending on whether

974   the centroid of the fine triangular element is in $T_1$ or $T_2$. As in the previous case, this is

975   then used in the second invariant $\tilde{\varepsilon}^2$ when evaluating the transformed functional $\tilde{\mathcal{A}}$.


977     While the number and type of unknown variables is the same in the two versions,

978   they differ considerably in accuracy, as is seen in Figs. 11 and 12. Fig. 11 compares the

979   upper surface u-velocity in both version, Approximations A and B, for the four coarse

980   grid combinations and the reference 24x24 fine grid calculation. Fig. 12 compares the

981   overall accuracy the same way by means of the RMS u-Error. As might be expected, the

982   accuracy of Approx. A is better than the accuracy of Approx. B, particularly in the case

983   when $L = 10$ km. Both versions are more accurate than the Blatter-Pattyn and BP+

984   approximations, except at the lowest 24x4 resolution when only the Approx. A version

985   retains that distinction.

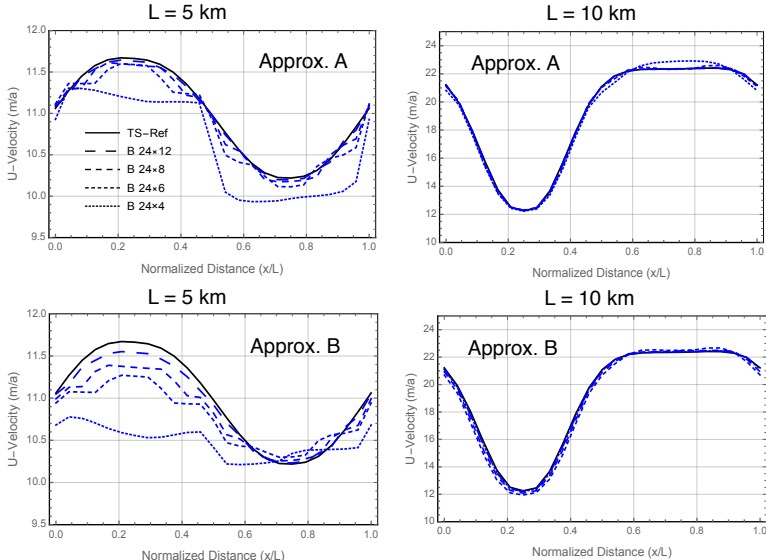

987   **Figure 11.** Comparing Approximations A and B. Test B. Upper surface u-velocity.

988   TS-Ref: Reference Stokes 24x24; Fine/Coarse resolutions (r x R): 24xR, R=12, 8, 6, 4.

990     In summary, the dual-grid approximation improves on the Blatter-Pattyn

991   approximation in both versions and at all resolutions, as seen in Fig. 12. Compared to the

992   BP+ approximations, here the vertical momentum equation is retained, although in

993   approximated form. In fact, the solution procedure here is very similar to that of the

994   unapproximated Stokes model except that the dimensions of the pressure and the vertical



velocity variables are reduced. Despite the differences with the unapproximated case, the
arguments in Appendix D regarding stability extend to the case $n_u > n_w = n_p$ appropriate
for the dual-grid approximation. As argued in Appendix D, provided the solvability
condition $n_w = n_p$ holds on the coarse grid, the "reduced" continuity equation may be
solved for the coarse vertical velocity in terms of the fine horizontal velocity
variables, $w = w(u)$, and in turn, the coarse pressure may be obtained in terms of the fine
horizontal velocity variables, $p = p(u)$, as in (79). As a result, pressure may be
eliminated in the dual grid version of the functional, converting the variational
formulation into a stable minimization problem. Thus, the solvability condition still
applies, but this time it applies to the coarse grid.

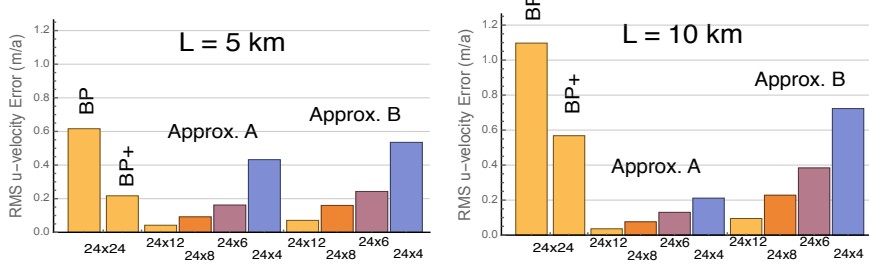


**Figure 12.** Comparing RMS u-Error in Different Approximations, Test B,
Resolutions (r x R): Approx. BP, BP+: 24x24; Approx. A, B: 24xR, R=12, 8, 6, 4.

**7. Second-Order Discretizations**

So far we have been using first-order elements, primarily P1-E0. However, in current
practice Stokes models are often based on the popular second-order Taylor-Hood P2-P1
element (Leng et al., 2012; Gagliardini et al., 2013). The two-dimensional P2-P1
element, illustrated in Fig. 13A, has velocities on element vertices and edge midpoints
and pressures on element vertices, resulting in a quadratic velocity and linear pressure
within the element. The element satisfies the conventional inf-sup stability condition
(Elman et al., 2014) but not the solvability condition (56). For example, in Test B with
direct substitution for basal boundary conditions, the number of vertical velocity
variables in the Taylor-Hood element, $n_w = 4nm$, is typically much larger than the
number of pressure variables, $n_p = n(m+1)$, where $n, m$ have been defined previously.

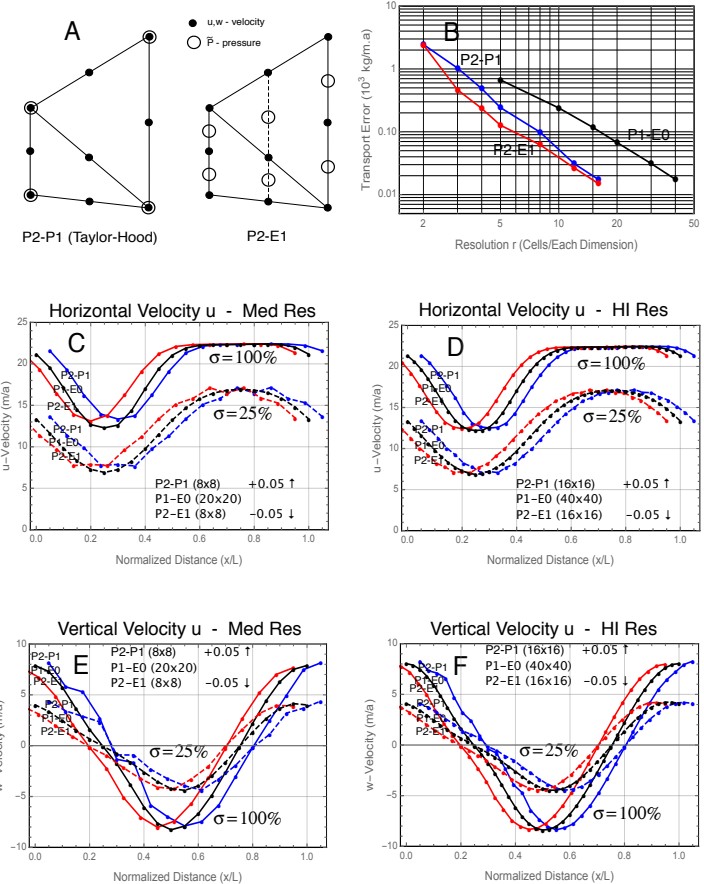

**Figure 13.** Comparing second-order discretizations based on the P2-P1 and P2-E1 elements from panel A to first-order discretizations using the P1-E0 element running Test B with L=10 km. For simplicity, only transformed Stokes calculations are compared; standard Stokes results behave similarly. Panel B compares the relative accuracy of the various schemes with increasing resolution, while panels C through F compare the horizontal and vertical velocities at medium and maximum resolutions, i.e., $r = 8, 16$ for second-order and $r = 20, 40$ for first-order cases. Plots labeled $\sigma = 100\%$ indicate the upper surface while dashed plots labeled $\sigma = 25\%$ indicate surfaces a quarter of the way up from the bottom.

Stokes models work well with a Taylor-Hood grid, as illustrated in Fig. 13, where both P2-P1 and P1-E0 models converge to a common Test B solution, but models that require the solvability condition (56) will not work on a P2-P1 grid, as discussed in connection with the extended Blatter-Pattyn approximation in §4.3.3. For these





applications an alternative will be needed if one wishes to use a second order
discretization. An alternative second-order element, consistent with an invertible
continuity equation, can be created by modifying the Taylor-Hood element to produce the
P2-E1 element illustrated in Fig. 13A. This element is second-order for velocities and
linear for pressure, just like the P2-P1 element, but the pressure is edge-based, as in the
P1-E0 element. The pressure is located midway between the velocities on the vertical
cell edges, including an "imaginary" vertical edge joining the velocities in the middle of
the vertical column as shown in Fig. 13A. Since pressures are collinear with vertical
velocities along vertical grid edges as in the P1-E0 element, the analysis in Appendix C,
§C2, demonstrates that element P2-E1 also satisfies the solvability condition (56).
Preferably, as explained in Appendix C, §C3, a P2-E1 grid is constructed using vertical
columns of quadrilaterals. A three-dimensional analog of this element exists and is
presented in Appendix C.

**Remark #2**: In addition to the P2-E1 element, it is possible to construct other elements
that feature an invertible continuity equation with second-order accurate velocities. Thus,
noting that there are $2nm$ triangular elements in a Test B problem grid, it is sufficient
that each triangular element contains two pressures, resulting in the same total number of
vertical velocity and pressure variables, namely, $n_w = n_p = 4nm$. The pressure will not be
linear within the element but this is unimportant since, as noted before, pressure has no
physical significance.

Fig. 13B shows the approximate error of the ice transport $T$ from (60) as a

function of grid refinement for the second-order P2-P1 and P2-E1 grids in transformed
Stokes Test B calculations, together with similar results for the first-order P1-E0 grid
from Fig. 3, for comparison. Calculation of the error $E = \left\| T - T_R \right\|$, as defined in §5.1, is
difficult because we do not have the converged value of the transport $T_R$. To estimate it,
we use Richardson extrapolation, assuming a rate of convergence proportional to $r^{-c}$,
where $r$ is the resolution and $c$ is the order of convergence, taken to be either $c = 2$ in a
first order model and $c = 3$ in a second order model. This gives a reasonable estimate of
the magnitude of the error as plotted in Fig. 13B. We note that both second order models
show approximately the same error at resolution $r = 16$ as the first order P1-E0 model at
resolution $r = 40$, and similarly for coarser resolutions such as $r = 8$ and $r = 20$,
respectively. However, although here the computational costs are not representative, it is





safe to say that these second-order calculations are considerably more expensive than the
first-order calculations at comparable resolution or accuracy.

Panels C, D in Fig. 13 compare the u-velocities, and panels E, F compare the w-

velocities, respectively, from several Test B calculations using the two second-order
models in comparison with first-order P1-E0 model results from Fig. 3. Each panel
shows results from the upper surface ($\sigma = 100\%$) in solid lines and results from a surface
a quarter of the way up from the bottom ($\sigma = 25\%$) in dashed lines. Panels C, E show
results from medium resolution calculations ($r = 8, 20$ in the second-order and first-order
calculations, respectively) and panels D, F show the corresponding results from the
higher resolution calculations ($r = 16, 40$). At these resolutions the accuracy of the first-
and second-order calculations is very similar so for clarity the second-order results are
displaced horizontally from the first-order results by 0.05 nondimensional units. The P2-
E1 results in magenta are displaced to the left and the P2-P1 results in blue are displaced
to the right. In general, models satisfying the solvability condition, namely the P1-E0
and P2-E1 models, are better behaved than the Taylor-Hood model, particularly in the
vertical velocity results, panels E and F, where velocity oscillations are present in the P2-
P1 results. This is presumably related to the well-known "weak" mass conservation of
the Taylor-Hood element. This problem is greatly improved by "enriching" the pressure
space with constant pressures in each triangular element (Boffi et al., 2012). In the 2D
Test B problem this increases the number of pressure variables from $n_p = n(m+1)$ in the
basic Taylor-Hood element to $n(3m+1)$, much closer to the $4nm$ needed to satisfy the
solvability condition. On the other hand, it should be noted that the pressure in the P2-E1
case is highly oscillatory while in the P2-P1 case it is well behaved. However, this is not
at all concerning since, as mentioned earlier in Remark #2, the transformed pressure, a
Lagrange multiplier, has no physical significance.

**8. Summary**

This paper introduces two main innovations. Together, the two innovations expand the
scope of traditional methods used in ice sheet modeling. The first innovation is a
transformation of the ice sheet Stokes equations into a form that closely resembles the
Blatter-Pattyn approximate model. This creates the ability to easily convert from one
model to the other. The variational formulation of the Blatter-Pattyn approximation



differs from the corresponding formulation of the transformed Stokes model only by the
absence of the vertical velocity $w$ in the second invariant of the strain rate tensor. This
makes it possible to create new Stokes approximations by focusing on the smallness of
vertical velocity compared to other terms in the variational functional. Two such
approximations are presented, the BP+ approximation and the dual-grid approximation,
which are cheaper than full-Stokes and more accurate than Blatter-Pattyn. Both
approximations are based on using an approximate vertical velocity that is obtained
inexpensively for this purpose, in general by solving the continuity equation for the
vertical velocity in terms of the horizontal velocity components. In the variational
formulation, the continuity equation is obtained by variation with respect to the pressure,
yielding a system of $n_p$ equations to solve for the $n_w$ vertical velocity variables. Thus,
vertical velocity can only be obtained from the solution of the discrete continuity
equation if the number of unknown vertical velocity variables is equal to the number of
unknown pressure variables, i.e., $n_w = n_p$. This is called the solvability condition.

The second innovation is the introduction of finite element grids in which the

solvability condition is satisfied. These grids incorporate a decoupled and invertible
discrete continuity equation. This has two important consequences. The first is that it
allows for the numerical solution of the continuity equation for the vertical velocity in
terms of the horizontal velocity components, $w = w(u,v)$, which is a prerequisite in the
different approximations made possible by the transformed Stokes formulation. A
second very important consequence is that invertibility of the continuity equation and the
availability of the vertical velocity in terms of the horizontal velocity components can be
used to remove the need for pressure as a Lagrange multiplier. Removing the pressure
from the system of Stokes equations, or from the variational functional, means that a
Stokes problem discretized with such a grid becomes a well-behaved minimization
problem rather than a mixed or saddle-point problem. This eliminates the need for the
inf-sup or LBB condition that is normally required to be satisfied in finite element
formulations. Some examples of such grids for use in both 2D and 3D are given in
Appendix C. An important case is the P1-E0 grid that has been used in most of the test
problems in this paper. To construct such grids we can focus on the term involving
pressure in the variational functionals (15) and (33) in isolation from the other terms, as is
done in (64). The pressure may then be considered a finite element "test function",
allowing us to construct appropriate test functions that yield $n_w$ independent equations



corresponding to the linear system of continuity equations (57), which is sufficient to
solve for the vertical velocity in terms of the horizontal velocity components. This is
already done in MALI (Hoffman et al., 2018), an ice sheet model based on the Blatter-
Pattyn approximation, to obtain the vertical velocity $w$ needed for the advection of ice
temperature (Mauro Perego, private communication).

We have also introduced some minor innovations in the implementation of the

frictional tangential sliding boundary condition that is often challenging to implement
numerically. Implementation directly into the Stokes equations involves the formation of
the normal component of the stress force at the boundary. This is extremely complex
(e.g., see DPL, 2010). Appendix A describes an alternative that avoids this complication.
The variational formulation makes it possible to also implement this boundary condition
using Lagrange multipliers, but this may not be desirable because it introduces extra
variables. A much more attractive alternative is the use of the no-penetration condition in
the form given by (14) to eliminate the vertical velocity by direct substitution along the
frictional portion of the basal boundary, as discussed in connection with the functional
(15). This automatically enforces both the frictional sliding condition and the no-
penetration condition.

Finally, we need to point out that no cost comparisons have been presented. This

is because the present calculations were made on a personal computer using the program
Mathematica, which is not at all representative of the computer hardware or the methods
that are used in practical ice sheet modeling. Furthermore, no effort was made to
optimize the calculations or to take advantage of parallelization. As a result, cost
comparisons would have been highly misleading.

**Code Availability**

All calculations were made using the Wolfram Research, Inc. program Mathematica in a
development environment. No production code is available.

**Competing Interests**

The author has acknowledged that there are no competing interests.



**Acknowledgements**

I am grateful to Steve Price, and especially to Mauro Perego and William (Bill)
Lipscomb for many helpful comments and suggestions that helped to improve the paper.

**References**

Auricchio, F., da Veiga, L.B., Brezzi, F., and Lovadina, C.: Mixed Finite Element
Methods, In Encyclopedia of Computational Mechanics Second Edition (Eds E. Stein, R.
de Borst, and T.J.R. Hughes), John Wiley & Sons, Ltd., 2017.

Blatter, H.: Velocity and Stress Fields in Grounded Glaciers: A Simple Algorithm for
Including Deviatoric Stress Gradients, J. Glaciol., 41, 333-344, 1995.

Boffi, D., Cavallini, N., Gardini, F., and Gastaldi, L.: Local Mass Conservation of Stokes
Finite Elements, J. Sci. Comput., 52, 383–400, 2012.
Cheng, G., Lötstedt, P., and von Sydow, L.: A Full Stokes Subgrid Scheme in Two
Dimensions for Simulation of Grounding Line Migration in Ice Sheets Using Elmer/ICE
(v8.3), Geosci. Model Dev., 13, 2245-2258, 2020.

Dukowicz, J.K., Price, S.F., and Lipscomb, W.H.: Consistent Approximations and
Boundary Condition for Ice Sheet Dynamics from a Principle of Least Action, J. Glaciol.,

56, 480-496, 2010.


Dukowicz, J.K., Price, S.F., and Lipscomb, W.H.: Incorporating Arbitrary Basal
Topography in the Variational Formulation of Ice Sheet Models, J. Glaciol., 57, 461-467,

2011.


Dukowicz, J.K.: Refolmulating the Full-Stokes Ice Sheet Model for a More Efficient
Computational Solution, The Cryosphere, 6, 21-34, 2012.

Elman, H.C., D.J. Silvester, and A.J. Wathen, 2014: *Finite Elements and Fast Iterative*
*Solvers: With Applications in Incompressible Fluid Dynamics*, 2nd Ed., Oxford
University Press, 494 pp.




Gagliardini, O., and Zwinger, T.: The ISMIP-HOM Benchmark Experiments Performed
Using the Finite-Element Code Elmer, The Cryosphere, 2, 67–76, 2008.

Gagliardini, O., Zwinger, T., Gillet-Chaulet. F., Durand, G., Favier, L., de Fleurian, B.,
Greve, R., Malinen, M., Martín, C., Råback, P., Ruokolainen, J., Sacchettini, M., Schäfer,
M., Seddik, H., and Thies, J.: Capabilities and Performance of Elmer/Ice, a New-
Generation Ice Sheet Model, Geosci. Model Dev., 6, 1299–1318, doi:10.5194/gmd-6-

1216 1299-2013, 2013.


Greve, R. and Blatter, H.: Dynamics of Ice Sheets and Glaciers, Springer-Verlag, Berlin
Heidelberg, 2009.

Heinlein, A., Perego, M., and Rajamanickam, S.: FROSch Preconditioners for Land Ice
Simulations of Greenland and Antarctica, SIAM J. Sci. Comput., 44, V339-B367, doi:
10.1137/21M1395260, 2022.

Hoffman, M. J., Perego, M., Price, S. F., Lipscomb, W. H., Zhang, T., Jacobsen, D.,
Tezaur, I., Salinger, A. G., Tuminaro, R., and Bertagna, L.: MPAS-Albany Land Ice
(MALI): A Variable-Resolution Ice Sheet Model for Earth System Modeling Using
Voronoi Grids, Geosci. Model Dev., 11, 3747–3780, doi:10.5194/gmd-11-3747-2018,

2018.


Hood, P. and Taylor, C.: Numerical Solution of the Navier-Stokes Equations Using the
Finite Element Technique, Comput. Fluids, 1, 1-28, 1973.

Larour, E., Seroussi, H., Morlighem, M., and Rignot, E.: Continental scale, high order,
high spatial resolution, ice sheet modeling using the Ice Sheet System Model (ISSM), J.
Geophys. Res., 117, 1–20, doi:10.1029/2011JF002140, 2012.

Leng, W., Ju, L., Gunzburger, M., Price, S., and Ringler, T.: A Parallel High-Order
Accurate Finite Element Nonlinear Stokes Ice Sheet Model and Benchmark Experiments,
J. Geophys. Res., 117, 2156–2202, doi:10.1029/2011JF001962, 2012.




Lipscomb, W.H., Price, S.F., Hoffman, M.J., Leguy, G.R., Bennett, A.R., Bradley, S.L.,
Evans, K.J., Fyke, J.G., Kennedy, J.H., Perego, M., Ranken, D.M., Sacks, W.J., Salinger,
A.G., Vargo, L.J., and Worley, P.H.: Description and Evaluation of the Community Ice
Sheet Model (CISM) v. 2.1, Geosci. Model Dev., 12, 387-424, 2019.

Nowicki, S.M.J. and Wingham, D.J.: Conditions for a Steady Ice Sheet-Ice Shelf
Junction, *Earth Planet. Sci. Lett.,* **265**(1-2), 246-255, 2008.

Pattyn, F.: A New Three-Dimensional Higher-Order Thermomechanical Ice Sheet
Model: Basic Sensitivity, Ice Stream Development, and Ice Flow across Subglacial
Lakes, J. Geophys. Res., 108(B8), 2382, 2003.

Pattyn, F., Perichon, L., Aschwanden. A., Breuer, B., de Smedt, B., Gagliardini, O.,
Gudmundsson, G.H., Hindmarsh, R.C.A., Hubbard, A., Johnson, J.V., Kleiner, T.,
Konovalov, Y., Martin, C., Payne, A.J., Pollard, D., Price, S., Ruckamp, M., Saito, F.,
Soucek, O., Sugiyama, S., and Zwinger, T.: Benchmark Experiments for Higher-Order
and Full-Stokes Ice Sheet Models (ISMIP–HOM*),* The Cryosphere, 2, 95-108, 2008.

Perego, M., Gunzburger. M., and Burkardt, J.: Parallel Finite-Element Implementation
for Higher-Order Ice-Sheet Models, J. Glaciol., 58, 76-88, 2012.

Rückamp, M., Kleiner, T., and Humbert, A.: Comparison of ice dynamics using full-
Stokes and Blatter-Pattyn approximation: application to the Northeast Greenland Ice
Stream, The Cryosphere, 16, 1675-1696, 2022.

Schoof, C.: Coulomb friction and other sliding laws in a higher order glacier flow model,
Math. Models. Meth. Appl. Sci., 20(1), 157–189, 2010.

Schoof, C. and Hewitt, I.: Ice-Sheet Dynamics, Annu. Rev. Fluid Mech., 45, 217–239,

2013.




Schoof, C. and Hindmarsh, R.C.A.: Thin-Film Flows with Wall Slip: An Asymptotic
Analysis of Higher Order Glacier Flow Models, Quart. J. Mech. Appl. Math, 63, *73-114*,

2010.


Seroussi, H., Ben Dhia, H., Morlighem, M., Latour, E., Rignot, E., and Aubry, D.:
Coupling Ice Flow Models of Varying Orders of Complexity with the Tiling Method, J.
Glaciol., 58, 776-786, 2012.

Tezaur, I. K, Perego, M., Salinger, A. G., Tuminaro, R. S., and Price, S. F.:
Albany/FELIZ: A Parallel, Scalable and Robust, Finite Element, First-Order Stokes
Approximation Ice Sheet Solver Built for Advanced Analysis, Geosci. Model Dev., 8,

1197-1220, 2015.


**Appendix A: The Frictional Sliding Boundary Condition**

The frictional sliding boundary condition requires the specification of the tangential
component of the frictional stress force. Dukowicz et al. (2010) obtain this by defining
the frictional stress force at the basal surface as follows

$$\sigma_{ij} n_j^{(b2)} = \left( \tau_{ij} - P\delta_{ij} \right) n_j^{(b2)} = -f_i$$

where $\sigma_{ij}$ is the stress tensor, $\delta_{ij}$ is the Kronecker delta, and $f_i$ is the frictional sliding
force vector from §2.2, and then subtracting out the normal component. The result is

$$\left( \tau_{ij} - \tau_n \delta_{ij} \right) n_j^{(b2)} + f_i = 0 \tag{71}$$

where $\tau_n = n_i^{(b2)} \tau_{ij} n_j^{(b2)}$ is the normal component of the stress force. However, the three
components of (71) are not independent because they already satisfy the tangency
condition at the basal surface. Since we already have one component of the basal
frictional boundary condition, namely, the tangency condition (10), we therefore need
only two more conditions and these are typically taken to be the two horizontal
components of (71). This option is problematic because of the need to form the highly
complex quantity $\tau_n$.



A simpler alternative is obtained by simply using the unneeded vertical
component of (71) to eliminate $\tau_n$ from the two horizontal components. The vertical
component of (71) gives
$$\tau_n n_z^{(b2)} = \tau_{zj} n_j^{(b2)} + f_z . \tag{72}$$

Substituting this into (71), we obtain the desired two conditions, as follows
$$n_z^{(b2)}\left(\tau_{(i)j} n_j^{(b2)} + f_{(i)}\right) - n_{(i)}^{(b2)}\left(\tau_{zj} n_j^{(b2)} + f_z\right) = 0 . \tag{73}$$

This is boundary condition (11) as used in §2.2.

Alternatively, one could use of a Lagrange multiplier $\Lambda$ in the variational
principle, as is done in (13) and in Dukowicz et al. (2011). This yields the tangency
condition (10) together with
$$\tau_{ij} n_j^{(b2)} + \left(\Lambda - P\right) n_i^{(b2)} + f_i = 0 . \tag{74}$$

Equation (74) provides three conditions, which, together with (10), is one too many.
However, one of these conditions must be used to determine the quantity $\Lambda - P$.
Contracting (74) with $n_i^{(b2)}$, and using the fact that $f_i$ is tangential to the basal surface,
gives us $\Lambda - P = -\tau_n$, which, when substituted into (74) gives us agreement with (71).
Alternatively, employing the vertical component of (74) to determine $\Lambda - P$, yields
$\Lambda - P = -\left(f_z + \tau_{zj} n_j^{(b2)}\right)\Big/ n_z^{(b2)}$. Substituting this into (74) gives the preferred boundary
condition (73).

**Appendix B: Test Problems**

We will use three two-dimensional test problems to demonstrate the new methods. The
geometrical configuration of the three test problem grids is illustrated in Fig. B1. The
first problem, Test B, is actually Exp. B from the ISMIP-HOM benchmark suite (Pattyn
et al., 2008); it features a no-slip condition (infinite friction) on a sinusoidal basal surface.
The second problem, Test D*, featuring sinusoidal friction along a uniformly sloped
plane basal surface, is a replacement with modified parameters for Exp. D from the
benchmark suite. This is because the ice flow in Exp. D is very nearly vertically uniform
(as seen in Fig. 4), which is more characteristic of a shallow-shelf approximation.
Increasing basal friction in Test D* rectifies this. These two test problems, Tests B and




D*, are used to illustrate and compare the performance of the new transformation versus
the traditional Stokes formulation.

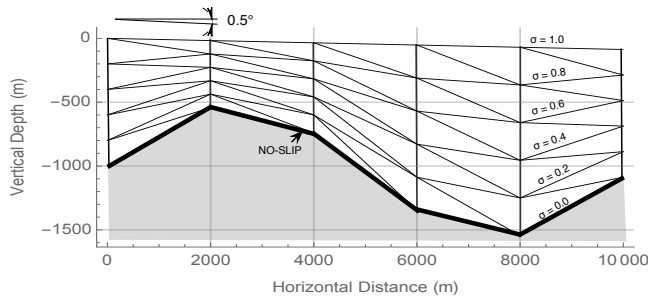

ISMIP-HOM Test Problem B - No Slip

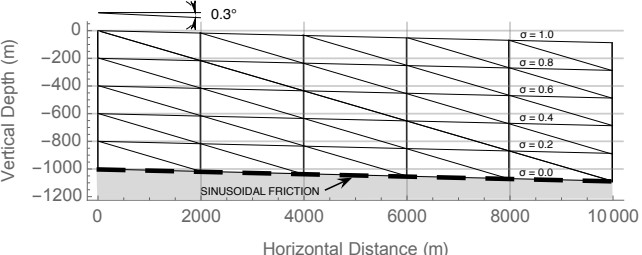

Test Problem D* - Sinusoidal Friction

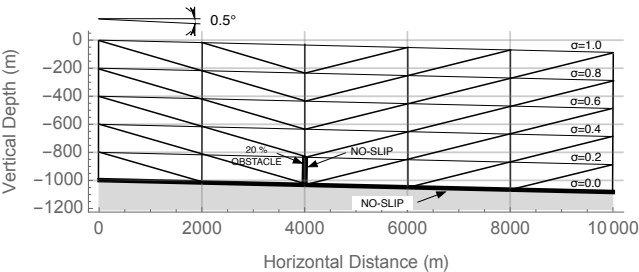

Test Problem O - 20% Obstacle - No Slip


**Figure B1.** Test problem grids. For clarity, a very coarse 5x5 configuration is used.

A third problem, Test O (for "Obstacle") has been introduced to illustrate
adaptive switching between the transformed Stokes and the extended Blatter-Pattyn
model in a problem where the small aspect ratio assumption underlying the Blatter-Pattyn
approximation fails locally. Test O has a unique feature, namely, a thin no-slip obstacle,
located at $x = 4\,km$ and extending vertically $200\,m$ from the bed (20 % of the ice sheet
thickness), as illustrated in Fig. B1, which forces the ice flow near the obstacle to adjust
abruptly. Because of the no-slip boundary conditions along the obstacle surface, a



triangular element in the lee of the obstacle, with one vertical edge and one edge along
the bed, would be a "null" element since all vertex velocities would be zero. This would
create zero stress and therefore a local singularity in ice viscosity in the element. To
avoid this, all elements at the back of the obstacle are "reversed" as compared to the ones
at the front of the obstacle, as shown in Fig. B1.

All tests feature a sloping flat upper surface, given by
$$z_s(x) = -x\,\mathrm{Tan}(\theta), \tag{75}$$

where $\theta = 0.5°$ for Tests B and O, and $\theta = 0.3°$ for Test D$^*$ (note that this differs from the
$0.1°$ slope in Test D), with a free-stress upper boundary condition in all cases. The
sinusoidal bottom surface elevation for Test B is specified by
$$z_b(x) = z_s(x) - H_0 + H_1 \mathrm{Sin}(\omega x), \tag{76}$$

where the depth $H_0 = 1000\,m$, $H_1 = 500\,m$, $\omega = 2\pi/L$, and $L$ is the perturbation
wavelength, which is also the domain length. The bottom surface in Tests D$^*$ and O is
parallel to the upper surface so the bottom surface elevation is
$$z_b(x) = z_s(x) - H_0. \tag{77}$$

The length $L$ in the ISMIP-HOM suite ranges from $5\,km$ to $160\,km$, but here we
consider only the two cases at the high end of the aspect ratio $H_0/L$ range, namely,
$L = 5\,km$ and $L = 10\,km$, where the inaccuracy of the Blatter-Pattyn approximation
becomes noticeable. Lateral boundary conditions in all cases are periodic. The spatially
varying friction coefficient for Test D$^*$ is given by
$$\beta(x) = \beta_0 + \beta_1 \mathrm{Sin}(\omega x), \tag{78}$$

where the friction coefficients are $\beta_0 = \beta_1 = 10^4\,Pa\,a\,m^{-1}$ (these are an order of
magnitude higher than in Test D). Physical parameters used for the test problems are the
same as in ISMIP-HOM, namely, ice-flow parameter $A = 10^{-16}\,Pa^{-3}a^{-1}$, ice density
$\rho = 910\,kg\,m^{-3}$, and gravitational constant $g = 9.81\,ms^2$. In general, units are MKS,
except where time is given per annum, which is convertible to per second by the factor
$3.1557 \times 10^7\,s\,a^{-1}$.



**Appendix C: Grids Satisfying the Solvability Condition**

**C1  A Solvable Continuity Equation**

As discussed in §4, the invertibility of the discrete continuity equation, at least in the simplest case of direct substitution for basal boundary conditions, requires a special grid that satisfies the solvability condition (56), i.e., $n_p = n_w$. Here we discuss several such grids and their properties.

The finite element discretization of our test problems, described in Appendix B and illustrated in Fig. B1, is constructed using vertical columns of quadrilaterals that are subdivided into triangles. Fig. C1 illustrates three different two-dimensional elements on triangles or quadrilaterals that may be used to construct grids that may be used to satisfy the solvability condition (56) in certain circumstances. The P1-E0 element is quite general and satisfies the solvability condition along each vertical grid edge, as will be demonstrated in Appendix C, §C2. As noted before, it has velocities located at triangle vertices, resulting in a linear velocity distribution within the triangle (P1), and pressure is located on the vertical edge of each triangle, resulting in constant pressure over the two triangles that share that edge (E0). A second order version of the P1-E0 element, the P2-E1 element, is illustrated in Fig. 13A. The two other elements in Fig. C1, i.e., the P1-Q0 and Q1-Q0 elements, satisfy the solvability condition when used in the grids for our test problems, Tests B and D*, but may not do so in other problems. The P1-Q0 element also has velocities on triangle vertices for a linear velocity distribution within the triangle (P1), but pressure is constant within the two triangles that form a quadrilateral (Q0). The element Q1-Q0 has velocities located at quadrilateral vertices and pressure centered in the quadrilateral, resulting in a bi-quadratic velocity distribution and a constant pressure within the quadrilateral (Q0).

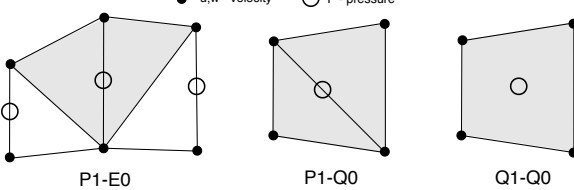

**Figure C1.** Three first-order 2D elements that may be used to satisfy the solvability condition, (56), in Tests B and D*.



Fig. C2 shows the convergence of ice transport with grid resolution for Test B
calculations using these three elements. The solutions are stable and they all converge to
the same value for the ice transport. The pressure distribution is smooth in the P1-E0
case, but contains very small fluctuations near the surface in the P1-Q0 and Q1-Q0 cases
that tend to disappear as the resolution is increased. The Q1-Q0 element is attractive
because of its simplicity but it has the potential for a pressure null space, resulting in
pressure checkerboarding (Elman et al., 2014, where the element is called Q1-P0). As a
result, apparently it is only used in a stabilized form. Here, however, the Q1-Q0 grid
satisfies the solvability condition in Test B and behaves well. Overall, these results
confirm our expectation of stability for grids when they satisfy the solvability condition
as will be discussed in Appendix D. The P1-E0 element is somewhat special because the
solvability condition (56) is satisfied individually along each vertical edge in grids that
are composed of this element, as opposed to being satisfied over the entire grid as in the
other two elements, as we discuss next.

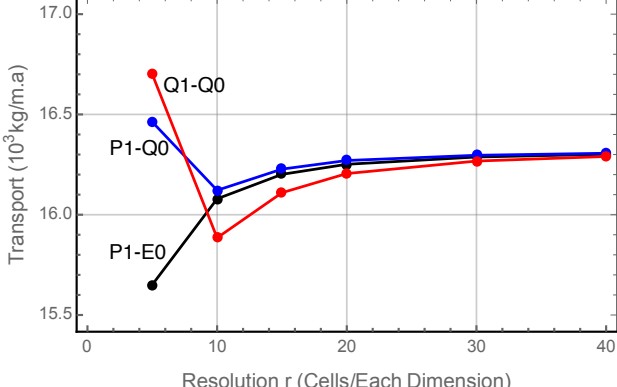


**Figure C2.** Convergence of Test B ice transport for grids using the three elements
from Fig. C1. All discretizations are stable and converge to the same solution.

**C2  Proving that the P1-E0 Element Satisfies the Solvability Condition**
The P1-E0 element from Fig. C1 is used in an example grid in Fig. C3. Note that
the grid is composed of vertical columns subdivided into triangular elements. To
demonstrate that the element meets the solvability condition (56) it is sufficient to
consider a single vertical edge extending from the bottom to the top. Assuming there are
$m$ edge segments in the vertical direction, there will be $m+1$ discrete $w$ variables and $m$
discrete $\tilde{P}$ variables, such that each $\tilde{P}$ variable is located between a pair of $w$ variables.
Since the $w$ variable at the bed is specified as a boundary condition, either directly as a
no-slip condition or in terms of the horizontal velocity component as part of a no-

penetration condition, there will be only $m$ unknown $w$ variables, and therefore $n_w = n_p$ along each vertical grid edge, and hence over the entire grid, as desired. In case Lagrange multipliers are used, there will be $m+1$ unknown discrete $w$ variables (since now the basal vertical velocity $w$ is also an unknown). This is matched by $m$ unknown $\tilde{P}$ variables, supplemented by one $\lambda_z$ or one $\Lambda$ unknown Lagrange multiplier variable, depending on the type of boundary condition. Thus, again the number of unknown variables equals the number of equations along every vertical edge, thereby satisfying the solvability condition whether Lagrange multipliers are used or not. Importantly, this means that this element can be used to satisfy the solvability condition irrespective of the boundary conditions on quite arbitrary grids, as illustrated in Fig. C3. These arguments apply for other versions of the P1-E0 element as well, such as the second order version P2-E1 in Fig. 13A or the 3D version in Fig. C4.

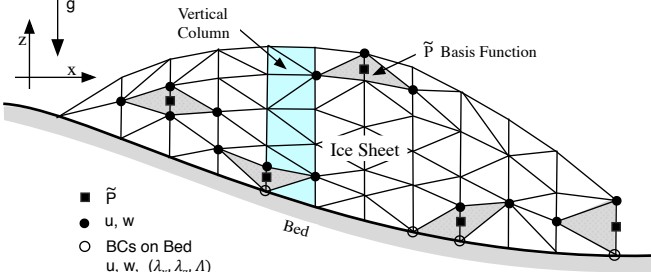

**Figure C3.** An illustration of a 2D edge-based P1-E0 grid, composed of vertical columns randomly subdivided into triangles. Pressures are located on the vertical edges. The triangulation and the configuration of the associated pressure basis functions (shown in gray) is quite general, allowing for a flexible triangulation of the domain.

**C3 Two- and Three-Dimensional Meshes Based on the P1-E0 Element**

The P1-E0 element has been used on the simple test problem grids in Fig. B1 and performs well. Moreover, the element has great geometric generality so it may be used for quite complicated grids, as in Fig. C3. Generally, there are two triangles associated with a pressure variable, one on each side of a vertical edge, except in situations as in Fig. C3 where the ice sheet ends at a vertical face. Even in this unusual situation there is no problem since the pressure is simply associated with the single triangle on one side of the vertical face.



The two-dimensional P1-E0 element has a relatively simple three-dimensional
counterpart, shown in Fig. C4. The mesh again consists of vertical columns, this time
composed of hexahedra. Each hexahedron is subdivided into six tetrahedra such that
each vertical edge is surrounded by from as few as four to as many as eight tetrahedra.
As in the two-dimensional case, velocity components are collocated at vertices, yielding a
piecewise-linear velocity distribution in each tetrahedral element, and pressures are
located in the middle of each vertical edge so that pressure is constant in the tetrahedra
surrounding that edge. Lagrange multipliers, if used, are located at the vertices on the
basal surface, yielding a piecewise linear distribution on the basal triangular facet. This
arrangement also satisfies the solvability condition (56) since pressures and vertical
velocities are again intermingled along a single line of vertical edges from top to bottom,
as in the 2D case. Thus, the solvability argument used in the two-dimensional case
applies, confirming that the 3D version of the P1-P0 element also satisfies the solvability
condition.

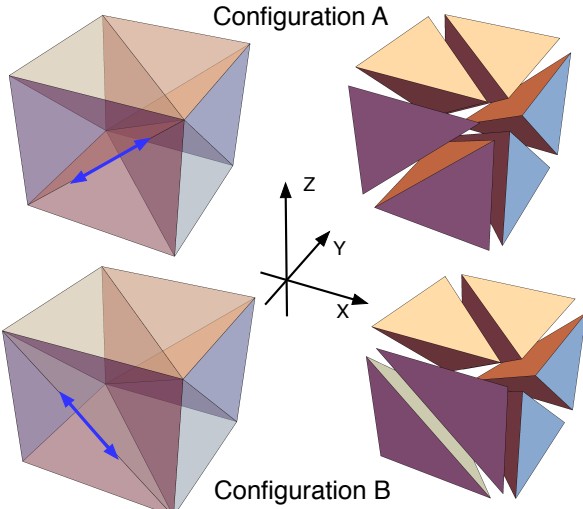

**Figure C4.** Three-dimensional P1-E0 tetrahedral elements that generalize the 2D
P1-E0 element of Fig. C1. Configurations A and B differ by having an internal
triangular face rotated, as indicated by the blue arrows. Both configurations
satisfy the solvability condition.
Fig. C4 shows two of the several possible configurations of a typical hexahedron,
including an exploded view of each configuration for clarity. The two configurations
differ in having the internal face of the two forward-facing tetrahedra rotated, creating
two different forward facing tetrahedra. The remaining six tetrahedra are undisturbed.





Since edges must align when hexahedra (or tetrahedra) are connected, this demonstrates
that the three-dimensional mesh can be flexibly reconnected and rearranged, just as in the
two-dimensional case.

**Remark #3**: A closely related and perhaps simpler three-dimensional P1-E0 element is
one based on the P2-P1 prismatic tetrahedral element used in Leng et al. (2012). A grid
of these elements is composed of vertical columns of triangular prisms, with triangular
faces at the top and bottom, which are then each subdivided into three tetrahedra. As in
Fig. C4, pressures are located on the vertical prism edges.

Meshes composed of P1-E0 elements have another useful property. Since
pressure and vertical velocity variables alternate along vertical grid lines, the matrix-
vector products $M_{WP}p$, $M_{WP}^{T}w$ in (47), corresponding to $\partial \tilde{P}/\partial z$ and $\partial w/\partial z$ in the
vertical momentum and continuity equations, respectively, consist of simple decoupled
bi-diagonal one-dimensional difference equations along each vertical grid line for
determining pressure, as in (79), and the vertical velocity, as in (58). This should be
particularly advantageous for parallelization.

Just as the two-dimensional second-order P2-E1 element in Fig. 13A is a
generalization of the P1-E0 element, a three-dimensional second-order P2-E1 element
may be constructed as a generalization of the P1-E0 element illustrated in Fig. C4.
Velocities are to be located at the vertices and at midpoints of the tetrahedral edges, and
pressures are to be located halfway between the velocities on vertical edges, including the
imaginary vertical edges through the midpoints of the tetrahedral edges, in the same way
as in the 2D case in Fig. 13A. The P2-E1 element, both 2D and 3D, also satisfies the
solvability condition since the arguments in Appendix C, §C2, apply here also because
pressures are again located midway between vertical velocities along all vertical edges.

**Appendix D: Proving the Stability of a Stokes Problem with an**
**Invertible Continuity Equation**
Here we show that a discretization of a Stokes problem is stable on a grid that
satisfies the solvability condition (56), or equivalently, one that is consistent with an
invertible continuity equation, i.e., (58). This is because such a discretization is
equivalent to the formulation of an unconstrained problem, i.e., a problem without the use
of pressure as a Lagrange multiplier. In fact, such a problem is also equivalent to an





optimization problem, or more specifically, to a minimization problem. To demonstrate
this, consider the full set of discrete Euler-Lagrange equations (47). Recall that the
solvability condition implies the invertibility of $M_{WP}$, and therefore also the invertibility
of its transpose, $M_{WP}^T$, i.e., (59). This means that we can solve for the pressure from the
vertical momentum equation, the second equation in (47), to obtain
$$p = -M_{WP}^{-1}\left(M_W\left(u, w(u)\right) + F_W\right), \tag{79}$$

where we would use $w(u)$ from (58). Using (79) to eliminate the pressure in the
horizontal momentum equation, we obtain
$$M_U\left(u, w(u)\right) - M_{UP}M_{WP}^{-1}\left(M_W\left(u, w(u)\right) + F_W\right) + F_U = 0 . \tag{80}$$

This is a nonlinear set of equations for just the horizontal velocity $u$, similar in this
respect to the standard Blatter-Pattyn formulation in that it is no longer a mixed or
saddle-point problem because pressure is absent. As a result, although still a rather
complicated nonlinear problem, it should not suffer from the stability issues discussed in
§4.3.1. Alternatively, using $w = w(u)$ in the functional (46) eliminates the pressure term
because continuity is already satisfied, and one obtains a reduced functional,
$$\mathcal{A}(u) = \mathcal{M}\left(u, w(u)\right) + u^T F_U + w(u)^T F_W . \tag{81}$$

This implies that $\mathcal{A}(u)$ is a positive-definite functional involving only the horizontal
velocity components because $\mathcal{M}\left(u, w(u)\right)$ is positive-definite (see §4.1), which means
that now the Stokes variational formulation represents an optimization, or more
specifically, a minimization problem. It is therefore n a well-defined and stable problem
for the horizontal velocities (albeit numerically very expensive). We conclude that the
solution of a Stokes model on a grid satisfying the solvability condition, or equivalently,
one that allows for an invertible discrete continuity equation is stable and well behaved.

Note that the arguments here and in §4 apply to arbitrary values of $n_u, n_w, n_p$, and
in particular, they apply in the case $n_u > n_w = n_p$ that is relevant to the "dual-grid"
approximation of §6.2.2. As a result, we conclude that the dual-grid approximation is
also stable provided the solvability condition (56) holds on the coarse grid.





**Remark #4**: Instead of the standard formulations of the Stokes problem that include the
pressure, such as (46) or (47), one could consider using the corresponding pressure-free
formulation, (80) or (81), to solve for $u$, followed by (58) and (79) if one is interested in
the vertical velocity and pressure. This corresponds to a discrete version of the pressure-
free formulation attempted analytically by Dukowicz (2012). However, this formulation
couples together large parts of the grid and produces a dense Hessian matrix when using
Newton-Raphson iteration, thus making the conventional numerical solution extremely
costly and therefore impractical, particularly for large problems.