# Peer review of "A Novel Transformation of the Ice Sheet Stokes Equations and Some of its Properties and Applications"

_EGUsphere, 2024_

## Referee Comment (RC2)

A Novel Transformation of the Ice Sheet Stokes Equations and Some of its Properties and Applications

J. Dukowicz

Review by Christian Schoof, University of British Columbia

**Overview**

The paper builds a unified variational formulation for Stokes flow and the somewhat misnamed Blatter-Pattyn model for ice flow. To get the trivia out of the way, I will call this the Herterich-Blatter-Pattyn model from this point onwards. History may be written by the winners, and Herterich clearly hasn't been one of them, *but* the first instance of the Herterich-Blatter-Pattyn model being formulated was (to my knowledge) the following paper:

- Herterich, K. 1987. On the flow within the transition zone between ice sheet and ice shelf, *in* Dynamics of the West Antarctic Ice Sheet. Proceedings of a workshop held in Utrecht, May 6–8, 1985, pp.!185–202. D. Reidel, Dordrecht,

This predates the commonly used Blatter (1995) and Pattyn (2003) references by eight and sixteen years, respectively. If we're going to fall into the trap of naming things after people, we probably owe it to ourselves to do that accurately. Not that I've been able to convince anyone of this point so far.

Back to the present. It took me a bit of effort in stripping away detail and side notes to understand that the primary advance of the paper is a numerical formulation in which one can dynamically switch between the simpler Herterich-Blatter-Pattyn and the more complete Stokes flow model. The latter applies to flows with arbitrary aspect ratios and abrupt changes in boundary conditions, while the former requires a "shallow" flow.

At face value, such a "switchable" model makes a lot of sense, since ice sheet flow is shallow in most places, but often contains boundary layers (at ice divides, ice stream margins, grounding lines etc) where the Stokes equations must be solved. The present paper uses the variational structure of both models to create a unified formulation, by rewriting the Lagrangian for the Stokes equations so that it takes the form of the Herterich-Blatter-Pattyn Lagrangian with a few extra terms retained. The unified formulation then consists of introducing a flag that activates or deactivates these Stokes correction terms. This is an intriguing idea and deserves to be published. I am not entirely convinced that *The Cryopshere* is the right vehicle as there are a number of technical issues that deserve more thorough scrutiny; *Geoscientific Model Development* would have seemed a more appropriate journal in the EGU stable to use, but really I would have been inclined to go with something like *J. Comp. Phys.*.

As far as I can tell, the modified Stokes Lagrangian does not alter the saddle point strucutre of the Stokes flow problem (in the sense that the solution maximizes the modified Stokes functional with respect to $\tilde{p}$ and minimizes with respect to $\boldsymbol{u}$). Still, the modified Stokes Lagrangian is a fairly ugly object that obscures the natural symmetries of the original Stokes flow Lagrangian and introduces a much larger null space to the elliptic operator. For the usual Stokes flow Lagrangian, that null space consists of rigid body rotations modulo any such motions that are precluded by the boundary conditions (by way of restrictions on the space of admissible functions). In the modified Stokes Lagrangian of the present paper, the vertical velocity $w$ can be changed by adding an arbitrary function of the vertical coordinate $z$ only while leaving the elliptic part of the Lagrangian uchanged; such functions of $z$ are then penalized through the incompressibility constraint.

That ugliness appears to be an unavoidable part of a unified formulation. It does however mean that you would probably not choose to use the modified Lagrangian for the analysis of general Stokes flow problems, except to unify the Stokes and Herterich-Blatter-Pattyn problem computationally. The paper can probably streamlined by focusing on that aspect at the exclusion of some of the more peripheral commentary, and the title of the paper could probably be made more informative.

In terms of pursuing that unification of Stokes and Herterich-Blatter-Pattyn models, there are two significant issues that I can see:

1. While the variational formulation of the Stokes flow problem *is* a saddle point problem, the same is not true of the Herterich-Blatter-Pattyn problem. The latter naturally wants to be solved as an unconstrained minimization problem, with the incompressibilty condition solved a posteriori for the vertical velocity component $w$. Enforcing incompressibility through a Lagrange multiplier as part of the variational formulation leads to problems that occupy most of the technical material in the paper: unlike the modified Stokes Lagrangian, the elliptic part of the Herterich-Blatter-Pattyn Lagrangian does not contain $w$ at all, which must instead be solved for through the (hyperbolic) incompressibility condition alone — for which the standard function spaces used in Stokes flow solvers are unsuitable. This is perhaps unsurprising, as the standard function spaces used in finite elements are problematic for hyperbolic equations, and discontinuous basis functions (as used in discontinuous Galerkin methods) might be preferable for $w$ in the Herterich-Blatter-Pattyn problem; they might at least alleviate issues such as $\nabla \cdot \boldsymbol{u} = 0$ being an overdetermined problem if we assume that $\boldsymbol{u}$ is represented by P1 basis functions, which work well for the force balance part of the Herterich-Blatter-Pattyn problem.[1]
* * *
[1] By contrast, the "P1-E0" spaces advocated here go in a different direction: $w$ is still represented by piecewise linear, and therefore continuous, basis functions, but the incompressibility constraint is weakened through the choice of a "coarser" basis function for the Lagrange multiplier that enforces incompressibility, averaging over two adjacent elements.

The main challenge identified in the paper (which motivates the choice of those P1–E0 soaces, and the others discussed in sections 6,7 and appendix C) is to make sure that the choice of basis functions for the "Blatter-Pattyn system" (that is, the Blatter-Pattyn equations and the incompressibility condition) can also be used for the modified Stokes system. Unfortunately, I think that there are some issues with how this has been addressed in the paper, and I suspect that there is a mis-understanding of what the inf-sup ("LBB") condition really does. More on this below.

2. More practically, I am not convinced that the method developed here will be adopted widely. Consider this: unlike its depth-integrated variants developed by Richard Hindmarsh, the Herterich-Blatter-Pattyn requires a three-dimensional do-main to be resolved. This leads to a large number of computational degrees of freedom. In fact, the only advantage relative to a Stokes flow model is that there are fewer variables to be solved for in *the same* computational domain, as you can solve for $(u, v)$ using the elliptic solver, and the compute $w$ and $p$ *a posteriori* if required. As far as I can tell, the Seroussi et al (2012, cited in the manuscript) tiling method makes use of that. The method proposed here, of solving for $(u, v, w, p)$ using the same basis functions for both models without explicitly using the sim-pler structure of the Herterich-Blatter-Pattyn model, seems to get rid of that last advantage. Which begs the question, why bpther, given that the Stokes model is preferable in terms of the physics it represents. Unfortunately, the last paragraph of the conclusion puts paid to any hope that the paper might address whether the new approach is actually going to be computationally competitive.

To follow up on the first point above, I am concerned that appendix D is misleading. Apologies if this gets a little long below; I am not as much of an expert at this as I'd like to be, so it took me a bit more explanation. My understanding of the inf-sup or Ladyzhenskaya-Babuška-Brezzi condition for Stokes flow problems is the following. Take the Hessian matrix in (55) of the present paper

$$\mathbf{M} = \begin{pmatrix} \mathbf{A} & \mathbf{B} \\ \mathbf{B}^{\mathrm{T}} & \mathbf{0} \end{pmatrix} \tag{1}$$

and we have to solve $\mathbf{Au} + \mathbf{Bp} = \mathbf{f}$, $\mathbf{B}^{\mathrm{T}}\mathbf{u} = \mathbf{0}$ where $\mathbf{f}$ is the relevant residual of the Stokes equations. $\mathbf{A}$ is positive semi-definite, which makes the solution of this problem (if it exists) equivalent to a saddle point: to find

$$(\mathbf{p}, \mathbf{u}) = \arg\max_{\mathbf{q}} \min_{\mathbf{v}} \left( \frac{1}{2}\mathbf{v}^{\mathrm{T}}\mathbf{Av} - \mathbf{v}^{\mathrm{T}}\mathbf{f} + \mathbf{vBq} \right). \tag{2}$$

The *purely discrete* inf-sup condition for the existence of a unique saddle point then becomes that i) $\mathbf{A}$ is elliptic[2] on the nullspace of $\mathbf{B}^{\mathrm{T}}$, and that ii) $\mathbf{B}$ is of full rank.
* * *
[2]that is, $\mathbf{v}^{\mathrm{T}}\mathbf{Av} > c\mathbf{v}^{\mathrm{T}}\mathbf{v}$ for some fixed $c > 0$ and any $\boldsymbol{v}$ that satisfies $\mathbf{B}^{\mathrm{T}}\boldsymbol{v} = 0$

Appendix D does not address the ellipticity of the matrix $\mathbf{A}$ (which, for the Stokes flow model, is not really an issue), but correctly identifies that $\mathbf{B}$ is of full rank if it can be written as

$$\mathbf{B}^{\mathrm{T}} = \left(\mathbf{M}_{UP}^{\mathrm{T}}, \mathbf{M}_{WP}^{\mathrm{T}}\right) \tag{3}$$

and $\mathbf{M}_{WP}$ is invertible. There is a bit of a sleight of hand here, as the text equates invertibility with $\mathbf{M}_{WP}$ being square (which is obviously a necessary but not a sufficient condition). The text in appendix D actually doesn't invoke saddle point theory directly, but instead eliminates the pressure variable from the elliptic part of the problem using a Schur complement. However, the "proof" in appendix most likely stands as a demonstration of *solvability* for the discretized problem, if we overlook the fact that a square matrix $\mathbf{M}_{WP}$ might still not be invertible.

Solvability is likely to be insufficient, however. As far as I understand (and I would caution that I've only recently revived an interest in mixed finite element methods), *stability* of mixed finite element formulations is not the same as solvability of the discretized problem. Spurious pressure oscillations occur in finite element solutions of the Stokes equations using basis functions that do not satisfy the inf-sup conditions even though (as far as I know) the discretized problem can remain solvable. The pressure oscillations should then be indicative of a lack of convergence under mesh refinement, not of a lack of solvability of the discrete problem.

If we set $n = 1$ for simplicity, then the discretized Stokes flow saddle point problem becomes finding $(\boldsymbol{u}_h, p_h) \in V_h \times Q_h$

$$A(\boldsymbol{u}_h, \boldsymbol{v}_h) + b(v_h, \tilde{p}_h) = \langle \boldsymbol{v}_h, f \rangle \tag{4}$$
$$b(\boldsymbol{u}_h, q_h) = 0. \tag{5}$$

for all $(\boldsymbol{v}_h, q_h) \in V_h \times Q_h$, where $V_h$ and $Q_h$ are finite-dimensional subspaces of $V = \{\boldsymbol{v} \in [H^1(\Omega)]^3 : \boldsymbol{v} \cdot \boldsymbol{n} = 0 \text{ on } \partial\Omega_b\}$ and $Q = L^2(\Omega)$, respectively, endowed with their usual norms. In the weak form of the transformed Stokes problem, $A : V \times V \mapsto \mathbb{R}$ is the elliptic operator

$$A(\boldsymbol{u}, \boldsymbol{v}) = \int_\Omega \mu \left(\nabla\boldsymbol{u} : \nabla\boldsymbol{v} + (\nabla \cdot P\boldsymbol{u})(\nabla \cdot P\boldsymbol{v})\right) \mathrm{d}\Omega + \int_{\Omega_b} \beta\boldsymbol{u} \cdot \boldsymbol{v} \, \mathrm{d}\Gamma, \tag{6}$$

$P\boldsymbol{v}$ being the projection of a vector onto the $x_1x_2$-plane, $P(v_1, v_2, v_3) = (v_1, v_2, 0)$. $b : V \times Q \mapsto \mathbb{R}$ is the incompressibility constraint in weak form

$$b(\mathbf{v}, q) = \int_\Omega q\nabla \cdot \boldsymbol{u} \, \mathrm{d}\Omega \tag{7}$$

My understading is that, in order to ensure convergence of $(\boldsymbol{u}_h, \tilde{p}_h)$ to a weak solution of the Stokes problem, we are looking for a family of subspaces $(V_h, Q_h)$ such that

$$||\boldsymbol{v}_h - \boldsymbol{v}|| + ||q_h - q|| < C_h(||v|| + ||q||) \tag{8}$$

where $C_h \to 0$ as mesh size $h \to 0$, *and* that the inf-sup condition holds in the following "infinite-dimensional" form: 1) $A$ is uniformly elliptic on the kernel of $b$, $A(\boldsymbol{v}, \boldsymbol{v}) > c \, ||\boldsymbol{v}||^2$ for some fixed $c > 0$ and all $\boldsymbol{v} \in V$ such that $b(\boldsymbol{v}, q) = 0$ for all $q \in Q$ and 2) $\inf_{q \in Q_h, q \neq 0} \sup_{\boldsymbol{v} \in V_h, \boldsymbol{v} \neq \boldsymbol{0}} b(\boldsymbol{v}_h, q_h)/(||\boldsymbol{v}_h|| \, ||q_h||) \geq \beta$ **where $\beta$ is independent of resolution $h$.**

If I omitted the last half-sentence and allowed $\beta = \beta_h$ to depend on resolution $h$ rather than enforcing a uniform bound, then condition 2) would simply be equivalent to corresponding block matrix $\boldsymbol{B}^{\mathrm{T}}$ having full rank (that is, not having a non-trivial nullspace.[3] Again, as far as I understand, it is the uniformity of the inf-sup condition under grid refinement that makes establishing *stability* of mixed finite element schemes non-trivial (and more difficult than the dimensional counting argument at the heart of appendix D).

Why is all of this relevant? Key to the proposed scheme is that I *can* find suitable function spaces $(V_h, Q_h)$ that work equally well for the Stokes problem (for which I have constructed the discussion above) and for the Herterich-Blatter-Pattyn problem (for which the invertibility of $\mathbf{M}_{WP}$ is an actual necessary condition for the solvability of the discretized problem). My point is that invertibility of $\mathbf{M}_{WP}$ and stability of the resulting mixed problem for the Stokes equations are probably not the same thing, at least in my understanding of the theory of saddle point problems. This affects pretty much the entire discussion of finite element basis functions in the paper.[4] My point about choice of journals is relevant here: I think this should really be read by reviewers that are expert at mixed finite element problems, rather than glaciological dabblers such as myself.

I hope I have been clear to say that I may be wrong. If I am, the text of the paper should be more explicit about the technical issues that I have raised; it's currently fairly vague in spelling out what the "LBB" (inf-sup) condition actually is, what it does, and whether the basis functions described in sections 6–7 and appendix C satisfy the inf-sup condition. (Also, if you're going to introduce a concept like that, please spell out what the acronym "LBB" stands for before using it, and maybe cite the original places where it comes from.)

My other main points would be the following (partially elaborated under "specific points")

1. The notation in the paper is fairly idiosyncratic, which made it more time-consuming for me to follow various pieces. There is a mix of standard subscript notation, a very much nonstandard "round bracket around subscript means projection onto the horizontal plane" variant to subscript notation, and the use of explicit component
* * *
[3]since I would otherwise have $B(\boldsymbol{v}_h, q_h) = 0$ for all $\boldsymbol{v}_h$ and some $q_h$

[4]There is another issue in play here: the inf-sup condition provides sufficient conditions for a convergence, but not necessary ones. You might also decide that you don't care about convergence of the discretized pressure solution $\tilde{p}_h$, but only of $\boldsymbol{u}_h$, which may conceivably relax the conditions you need to impose. I am not in a position to comment on that, however — you need a real expert reviewer.

notation as in equations (7), (28) and (35) to write scalar quantities that should really be denoted by contractions over subscripts. The paper makes enough assumption about the reader's level of mathematics that I would recommend streamlining this as I don't imagine much of an audience who will be helped rather than hindered by the nonstandard notation. In particular, I would discourage the round bracket notation in favour of a more standard projection operator: if $P$ is the projection onto the horizontal plane, $P(v_1, v_2, v_3) = (v_1, v_2, 0)$, then you can simply replace $a_i$ by $P_i(\boldsymbol{a})$ and retain summation over indices from 1 to 3. I say this even though the author has used it previously: the use of nonstandard notation it may make the paper harder to decipher for numerical analysts, who should really be encouraged to delve deeper into the theory relevant to the paper.

The use of superscripts $^{(s)}$, $^{(b_1)}$ and $^{(b_2)}$ on surface normals is also redundant, as you specify the parts of the domain boundary to which the stated boundary conditions apply. Keep this as simple as possible, because it it certainly isn't simple.

2. The paper is quite ungenerous to the prior literature. I can see the appeal of referencing only your own paper ("DPL" in the present case) as the default reference because you know its content extremely well, but many of the introductory concepts in this manuscript have been developed in other places, which remain uncited. That may put off other practicioners who should read this paper (and who might in fact get some sort of notification, alerting them to your paper if you were to cite them!). It's also unhelpful to any reader who wants to make sense of the field: the may conclude that, really, only DPL is relevant as a prior publication. In particular, I see almost no reference to the extensive numerical analysis literature on Herterich-Blatter-Pattyn and Stokes flow models in glaciology (except my ca. 2010 own effort in that direction, which does get a citation somewhere).

3. The "Improved Blatter-Pattyn or BP+ Approximation" in section 6.2.1. I am unconvinced that it makes sense to include this in the present paper, unles you want to "patent" the idea for eternity. (Who knows, maybe someone will in future refer this approximation by someone else's name, the same fate that befell Herterich?) The reason why I am unconvinced is that this material further breaks the flow of the paper, without being robust. The supposedly improved approximation remains an *ad hoc*, partial retention of higher order terms in the Stokes flow model, higher order being in sense of the aspect ratio. There is no theoretical justification for doing that, as a partial retention of higher order terms in a model comes with no guarantee of a reduction in model error. In fact, it can make the model error worse. We do see a reduction in model error for the "BP+" model ("HBP+" / "D"?) in the single test that the "improved" model has been subjected to (ISMIP-HOM Test B, figure 9 of the manuscript). However, that is not really a robust demonstration of an "improved" model. It's also unclear whether the supposedly improved model is really competitive relative to solving the full Stokes model in terms of the tradeoff

between accuracy and computational effort. (See my second major point above.)

**Specific points**

A number of specific points, some of which will replicate elements of what I've written above.

- p3, line 81: "finite element grid" — "grid" might be seen to imply regularity. "finite element basis functions"? I mean, closer scrutiny does indicate that the basis functions used here are restricted to quite regular meshes, so perhaps the terminology is not wrong here, but that required regularity actually needs to be discussed somewhere.

- p3 line 84 "these two elements are so-named because they employ edge-based pressures" — I actually found that confusing. The basis function for pressure is not really defined on edges (since there are plenty of edges in the mesh that don't have a pressure defined on them). They are really P0 basis functions in which two adjacent triangles of the triangulation are assigned the same pressure.
  Also, to make it clear that you are inventing these basis functions, avoid the passive voice here. "I have named these E0 to indicate that pressure is defined on select element edges" or similar. It took me a while to realize that I would look in vain for references to these elements in the literature.

- p 3 line 89 onwards " A conventional ice sheet Stokes model discretized on such a grid is numerically equivalent to an inherently stable positive-definite minimization (i.e., optimization) problem, as demonstrated in Appendix D. This is in contrast to the ubiquitous Stokes finite element practice of needing to use elements that satisfy the "inf-sup" or "LBB" condition for stability (see Elman et al., 2014, and the brief discussion in §4.3.1)." — This honestly confused me. If you're saying that current practice in solving the Stokes equations in ice sheet dynamics ("A conventional ice sheet Stokes model") is to use unconstrained optimization, then please provide a reference. It's very hard to do unconstrained optimization for Stokes flow as you need divergence-free basis functions. If you mean that *your new approach* is equivalent to constructing such a divergence-free basis, then say that instead. However, you really also need to show that the reduction of the single Newton iteration step to a problem for $(u_1, \ldots, u_N, w_1, \ldots, u_N)$ only in appendix D (taking the dimensionality argument for $\mathbf{M}_{WP}$ at face value) really is equivalent to a convergent solution (under mesh refinement) of the unconstrained Stokes optimization problem (that is, the problem in which the incompressibility constraint is directly imposed on admissible $\boldsymbol{u}$). I don't think this is entirely trivial, see also my discussion of the inf-sup criterion above. (Put this another way, for any set of basis functions that allows the problem to be solved, I must be able in principle to

eliminate the discretized pressure variable in favour of retaining on the nodal values of velocity, even if I would not choose to do the matrix manipulations involved. By your logic, any such scheme should therefore be robust. Is that true?) I may be mistaken about what you're trying to say here, but it may therefore be worth finding a different way of saying it.

- p 4 line 116. $S = S_{B1} \cup S_{B2}$?

- p 4 line 121–123. See above re: notation.

- eqs (8), (10, (11), (13)–(15). The superscripts on your unit normals seem to be redundant. There is only one (outward-pointing) unit normal to each of these parts of the boundary.

- p6, line 170 "the simplest representation" — I beg to differ. There is a standard way of writing shear stress, as the traction $\Sigma_i = \sigma_{ij}n_j$ minus the normal component of traction $n_i\Sigma_j n_j$, or
$$(\delta_{ij} - n_i n_j)\sigma_{jk}n_k.$$
That is equation (71) in appendix A, and I think you'd be well advised to just use that (standard) form. (The point about weak solutions is surely that you never need to actually evaluate the shear stress itself from the stress tensor and the normal to the surface; you just need to know the constitutive relation, which here is just

$$(\delta_{ij} - n_i n_j)\sigma_{jk}n_k. = \beta u_i.$$

I think what the paper says isn't wrong, but manipulations don't seem they will help the unwitting reader, who is just wrapping their head around the basic model formulation. Especially at this point, where you haven't motivated your use of the bracketed indices at all yet, in terms of the mathematics you're doing.

- p 6 line 174, "$f_i$ is a specified frictional sliding force vector." — $f_i$ has the wrong units for a force. It's a traction, but why not just call it an interfacial shear stress? $\tau_i$ would probably be a more common symbol.

- p 7 line 180 "...can easily be added" → "...can esaily be added to equation (10)"?

- p 7 "(see DPL, 2010, for a fuller description of the variational principle applied to ice sheet modeling)" — you should refer to
Chen, Q, M. Gunzburger and M. Perego. 2013. Well-posedness results for a nonlinear Stokes problem arising in glaciology. *SIAM J. Math. Anal.*, 45(5), 2710-2733. This is probably the most comprehensive analysis of Stokes flows with sliding in glaciology (and you'll find the relevant functional there, too, naturally).

- p 7 line 200 "...(see Schoof, 2010, in connection with the Blatter-Pattyn model)" — I feel a little unfairly singled out here. There are earlier papers by Coling and Rappaz (MSAN, 1999), Glowinski and Rappaz (2003), Chow et al 92004) Rappaz and Reist (M3AS, 2005) that deserve an equal mention as having contributed to the analysis of the Herterich-Blatter-Pattyn model

- p8 line 208 " As in DPL (2010), arguments enclosed in square brackets, here $u_i$ , $P$, $\lambda_i$ ,$\Lambda$ , indicate those variables that are used in the variation of the functional." — this is an odd way of putting it. A functional is simply a mapping (or function) from some vector space (or perhaps an affine space) into the real numbers. So why not just say that you're enclosing the arguments of the functional (which is a function and therefore takes arguemnts like any other function![5]) in square brackets.

- p 8 line 210 onwards. I don't think this correct, or at least, it seems misleading. In my understanding, there are two ways of imposing Dirichlet conditions: by Lagrange multiplier, or by restricting the space of admissible functions. The latter is not the same as the kind of explicit substitution you're doing here. If I take the functional defined in equation (15) and I *don't* separately impose the constraint $\boldsymbol{u} \cdot \boldsymbol{n} = 0$ at $\S_b$ on the arguments $u_i$, then I don;t see that taking the first variation of the functional will recover that constraint. If you impose the constraint $\boldsymbol{u} \cdot \boldsymbol{n} = 0$, then you do not need to make the substitution in equation (14) in the friction term of the Stokes Lagrangian
The substitution here is again quite nonstandard, with little advance warning or real motivation. I think you're really trying to lay the groundwork for the modified Stokes variational principle later (by eliminating $w$ where that's going to be necessary) but here is an awkward place to do it, unless you explain why you're eliminating $w$.

- p 8 line 218 " Here we use $z_b$ as a shorthand notation for $z_b(x, y)$' — this is surely redundant? Same on line 259.

- page 9 line 232 "...the specified values of velocity are then obtainable a posteriori from (9) or (14)" — this is a strange way of putting it. What is a specified value? One that is prescribed? I don't think that's what you mean. I think you mean that the taking the first variation does not recover the boundary condition $\boldsymbol{u} \cdot \boldsymbol{n} = 0$, see previous point. However, you can't impose that "a posteriori" since you cannot solve the Euler-Lagrange equations for the functional in (15) without using the Dirichlet condition on velocity. The standard way of putting this (I believe) is to restrict the functional to the vector space of suitable smooth velocities (in $[W^{1,1+1/n}(\Omega)]^3$) *that satisfy the Dirichlet condition.*
* * *
[5]Unless you insist that a function have to take arguments in $\mathbb{R}^n$ for some finite $n$.

- page 9 line 254 "... as is done in the Blatter-Pattyn approximation (see DPL, 2010)." — is the apropriate reference not one of Herterich (1987), Blatter (1995) or Pattyn (2003(?

- page 10, line 265 "The standard Stokes pressure $P$ is some three orders of magnitude larger than the transformed pressure $\tilde{P}$" — asymptotic analysis of the problem will show that, more generally, the "Stokes correction" $\tilde{P}$ scales as $O(\varepsilon)$ when there is signfiicant sliding, where $\varepsilon$ is aspect ratio, and of $O(\varepsilon^{1+1/n})$ in the absence of sliding (see the Schoof and Hindmarsh 2010 reference).

- page 12 line 296. It would be a good idea to explain the role of the flag parameters $\xi$ and $\hat{\xi}$ earlier. I spent a page half guessing what they were.

- page 13, equation (32). As above, that is pretty ugly. I can't tell if there is a less confusing form, but might be worth trying.

- p 13 line 353 "The standard (or traditional) Blatter-Pattyn approximation (originally introduced by Blatter, 1995; Pattyn, 2003; later by DPL, 2010; Schoof and Hewitt, 2013)" — as above, Herterich (1987) was the person who introduced this. I don't think the Schoof and Hewitt review paper did much more than describe the theory, as oppposed to contributing to it. If I did anything here, then Schoof and Hindmarsh (2010) provided the first self-consistent asymptotic analysis of the model. If you want to list DPL for its description of the variatoinal formulation, you probably ought to cite the numerical analysis papers I have listed above (Coling and Rappaz, Glowinski and Rappaz, Chow et al, Rappaz and Reist; full citations in my 2010 paper that is in the reference list), who previously dealt with the same variational formulation.

- page 15 line 376 Remark # 1: seems like splitting hairs, especially as you wouldn't bother with this if you wrote down the weak form.

- page 15 line 419 "In summary, the extended Blatter-Pattyn model, (40)-(42), is equivalent to the standard Blatter-Pattyn model, (36), for the horizontal velocities, u,v , except that it also includes two additional equations that determine the pressure P ! and the vertical velocity w , which are usually ignored in the standard Blatter-Pattyn approximation when only the horizontal velocity is of interest. Because of this, we distinguish between the Blatter- Pattyn model that solves for just the two horizontal velocities (i.e., the standard Blatter- Pattyn approximation, (36)), and the Blatter-Pattyn system that solves for all the variables (i.e., the extended Blatter-Pattyn approximation, (40)-(42))." — again, this seems like splitting hairs. Anyone who needs to solve for temperature in an ice sheet (which is a standard part of any ice sheet simulation code) has to solve for the vertical velocity component, so I don't think insisting on a difference between the Blatter-Pattyn model and system is helpful. Solving for $\tilde{P}$ is not particularly relevant as we

know $\tilde{P} = 0$ for the Blatter-Pattyn model. For the sake of not going down rabbit holes, I would omit this.

- p 17 line 437 "The use of the continuity equation to solve for the vertical velocity w is a novel feature of the Blatter-Pattyn approximation since the continuity equation is not normally used for this purpose." — as per my previous comment, I don't think this is true or novel. The vertical velocity component is a pretty important quantity to be able to compute in ice sheet codes.

- p 17 lie 439 onwards. The use of a depth-integrated mass conservation equation is a boilerplate approach for thin film models, so a lengthy discussion without citation seems unnecessary, especially as this doesn't really lead anywhere in developing the novel material in the paper.

- p 17 line 482 "Additional details about the grid and the associated discretization..." — I flagged this above (grid versus mesh), but again: I think it's worth pointing out somewhere that some of the basis functions developed here really do require a regular mesh (grid?) rather than an unstructured triangulation. For the P1-E0 basis functions, I have to be able to match every triangle with precisely one neighbour, not leaving any neighbourless triangles. I don't think that's possible with every fully unstructured mesh. (?)

- page 19 line 504. Italicize the variables $u$ and $w$.

- page 21 line 526, "...or else they are "saddle point" problems since the Hessian matrix $M(u, w)$ is symmetric but indefinite, with both positive and negative eigenvalues..." — you can be more definitive about this I think: the "saddle point" terminology refers to the fact that there really is a saddle point, where you mimiize the quadratuc form associated with the matrix with respect to the velocity variables and maximize with respect to pressure.

- p 22 line 565 Please define the acronym "LBB" before using it.

- p 23 line 601 Taylor-Hood elements have gone from being an illustration of an element that satisfies the inf-sup condition (line 569) to becoming the standard reference for stable elements. Is it true in general that all elements that satisfy the LBB condition for the Stokes problem will leave $M_{WP}^{\mathrm{T}}$ non-invertible on dimensional grounds alone? If so, that is important to point out here.

- Sections 5.1–5.3 You're focusing on velocity solutions here. If there are stability issues with your choice of basis functions, I'd expect these to show up in the pressure solution. Can we see resultss for some of those? Obviously not relevant for Herterich-Blatter-Pattyn, but for the transformed and original Stokes flow problems. Especially pressure along the bed would be useful, but also convergence or lack thereof.

- Figure 7: this is fine, but more convincing would be a grounding line, where you have to worry about a normal stress constraint determining the grounding line location (and that normal stress is in general not cryostatic as assumed by Herterich-Blatter-Pattyn)

---

## Author Comment (AC1)

**Response to Reviewer RC1 – Prof. Ed Bueler**

**Egusphere-2024-1052 Article**

Author Responses in Red
I thank Prof. Bueler for a detailed and helpful review!

Summary:  This paper rewrites the standard glaciological (Glen law) Stokes model in a form which resembles a shallow approximation, the Blatter-Pattyn (BP) model.  This expresses the saddle-point structure of the Stokes problem in a form close to the unconstrained-optimization form of the BP model. The stability and finite element (FE) analysis of the new form is addressed, and new mixed FE pairs for vertically-extruded meshes are propsed.  Small-scale experiments are presented, and then prospective applications at larger scale are discussed.  The resulting essentially-theoretical paper is both frustrating and promising.  The manuscript's current form is notably inefficient, with 1500 lines of text.  The presentation is likely to be hard to read for those who have not already done battle with BP equations and related technical matters.  Despite doing numerical experiments, the author provides no open-source code basis for further development by readers, a clear demerit in 2024.  The manuscript avoids the function-space understanding of the Stokes and BP problems---this is the viewpoint from which these problems are known to be well-posed and by which they are solved by mainstream finite element libraries---but then it labors to build a fragmented substitute for this viewpoint.  Despite these flaws, the paper illuminates important matters.  It shows how the (transformed) Stokes equations are close to an "extended Blatter-Pattyn" (EBP) form, and thereby how the solvability conditions of the Stokes model work in practice over vertically-extruded meshes.  The EBP model has similar numerical and stability issues as the Stokes problem, which is actually clarifying because the numerical and FE character of the standard BP and Stokes models otherwise appear very different.  The inf-sup stability of the mixed Stokes problem is recognized here, when the mesh is extruded and when one simultaneously wants the EBP model to be solvable on the same mesh, as the requirement of unique solvability of the continuity (incompressibility) equation for the vertical velocity from the horizontal velocity.  A necessary condition for this to work is that the number of vertical velocity and pressure unknowns must be exactly the same, or rather that a particular matrix in the blockwise form of the discrete equations must be invertible.

Recommendation:  A manuscript which made the same points in half the length, and which provided open source code in a widely-used language, facilitating further

August 28, 2024

development, would be an excellent paper. Of course it is not realistic to expect re-coding at that level. However, significant revisions should be attempted. A much-shortened abstract is offered below, along with several other suggestions for trimming.

An effort has been made to tighten and shorten the manuscript while preserving the content. The line count has been reduced to 1340 while preserving most of the content. Unfortunately, it is not possible to provide open source code in a widely used language because of the piecemeal way that the work was carried out using the Mathematica program, as pointed out in the paper.

Specific Comments on Manuscript lines 9-35: This long abstract could be halved without losing meaning, by removing the sales pitches and by other simple edits. However, changes are also needed to clearly identify the models (systems) under consideration. The following is a guess/suggestion for an abstract which meets these objectives. It has 191 words vs 371 in the original: """We introduce a novel transformation of the Stokes equations into a form that resembles the shallow Blatter-Pattyn (BP) equations. The two forms only differ by a few additional terms, and the variational formulations differ only by a single term in each horizontal direction, but the BP form also lacks the vertical velocity in the second invariant of the strain rate tensor. The transformed Stokes model has the same type of gravity forcing as the BP model, determined by the ice surface slope. An apparently intermediate "extended Blatter-Pattyn" (EBP) form is identified, which is actually the same as the standard BP model although it retains a pressure variable. The role played by the vertical velocity in the transformed Stokes and EBP forms, reflected in the block-wise structure of their discrete equations, motivates the construction of new finite element velocity/pressure pairs for vertically-extruded meshes. With these new pairs, examples of which are demonstrated in 2D and 3D, the discrete continuity equation can be uniquely and stably inverted for the vertical velocity. We describe how to incorporate the new forms into codes that adaptively switch between Stokes and BP models, where the latter would lose accuracy."""

I have rewritten the abstract using many of these suggestions. Thank you.

August 28, 2024

line 41: "full" is unnecessary.

Removed line 52-72: The style of glaciology, used at unnecessary length in these lines, says some models are shallow and some are higher order. It is more accurate to say all are shallow, and to not claim some are "higher-order" because the order depends on which scaling argument is use.

I have used the term "shallow" only as part of the accepted names of some simple approximations. The term "higher-order" is commonly applied to the Blatter-Pattyn and other more accurate approximations.

line 99: "THE LOWER BOUNDARY OF an ice sheet ...". (A 3D ice sheet can't be divided the way the text says.) (1)

Don't quite understand what the problem is. This is an idealized situation of course. I will be glad to make whatever change is required.

lines 103-105: This "vertical line of sight" phrase appears here and later. Surely one can just say: "We assume the glacier's geometry is described by an upper surface function $z_s(x,y)$ and a lower surface function $z_b(x,y)$."

This was intended to mean that there should not be various indentations so that various multiple upper and lower surfaces would exist along a vertical line. Although unlikely, these could be handled but would complicate things considerably. I have changed this to say that there should be just one upper and one lower surface.

lines 105-106: There is nothing about the rest of the paper, in my reading, that excludes the techniques being used for floating ice. (Put $f\_i=0$ in equation (11)?) It is true that there must be sufficient drag--see the inequality in Schoof (2006)--*somewhere at the base* so that the velocity field is unique, but the techniques apply across grounding lines.

I have modified the sentence to say that ice shelves can be handled.

lines 112--126 Briefer notation is surely possible.

I have simplified by removing superscripts on unit normal vectors. Not sure what else can be done.

August 28, 2024

line 149: "positive-definite" --> "nonnegative"
Changed to "a positive quantity"

line 178-180: Whether or not the surface kinematical equations can be added "easily", the way this is said here is silly. The whole paper assumes fixed ice geometry.
Yes, fixed geometry is assumed. What this says is that flux inflows or outflows are allowed through a fixed geometry (which may be a crude representation of melting or refreezing at the bed).

lines 192-195: I don't know what this means. "There are some stress boundary conditions and it is easier for the author to think about them in the variational formulation."? No need for this?
This means that evaluating derivatives at boundaries is less accurate or more complicated because one-sided formulas have to be used due to the absence of information from across the boundary. I have changed the wording to make this clearer.

lines 197-200: No need for this.
I think this needs to be pointed out because most people use the weak formulation method and may not be familiar with the variational method.

lines 204-209: Is this option ever used later in the paper? (Line 233 suggests not.) If not, it can be removed and replaced with a simple declaration that the boundary conditions can be weakly imposed if desired.
I have indeed used it but most computations were done using direct substitution, as stated on Line 233. Of course, there is no difference in the results. However, it is a useful option and some people may prefer it. There are some consequences when Lagrange multipliers are used. For example, the "solvability condition" must be modified (see Line 626 in the originally submitted paper). For this reason, I prefer to leave this section as is.

lines 238-252: This is a valuable observation, namely form (17) which shows ~P solves a trivialized problem. If this observation is original, then great. Otherwise cite it more clearly; did it appear in DPL 2010? (The nearby citations to DPL do not refer to this main idea as far as I can tell.)

August 28, 2024

It is original but only insofar as it refers to the transformed pressure ($\tilde{P}$ not $P$) in the Blatter-Pattyn approximation. It does not appear in DPL, 2010, since the new transformation was not invented yet.

Figure 2: This basic point is greatly appreciated: The deviation from hydrostatic is relatively small. However, in this and almost all figures, the fonts are too small! (Also these figures are bad on a monochrome printer, but I suppose that train has left ...) Changing all figures would be difficult. Should be OK for young eyes ….

line 282: I don't think (22) is actually used *here*.
Yes, it is used in the strain rate tensor (6) and in the second invariant (7). See (26) and (28).

around line 282: Warn the reader that "dummy variables" ("flag variables"?) are about to be used. As the text is written, they are finally explained on the next page.
Done lines 286 and onward: I find "modified" really unpleasant here. For (25) the tensor ~\tau_ij is actually modified; it is not equal to the original. But in (26) the tensor is merely rewritten; neither "modified" nor the tilde have the same meaning as they do in the equation above. Similarly (27) and (28) are not "modified" but merely rewritten, as far as I can tell. I therefore would not say "modified" or add a tilde; just write out the new form. Equality means equality.
I must disagree here. Equations (26), (27), (28) are indeed modified because $\partial w / \partial z$ is replaced by $-\left(\partial u / \partial x + \partial v / \partial y\right)$ according to (22). They may have the same numerical value at convergence but they are discretized differently, so they are "modified". It is also important to distinguish quantities in the transformed Stokes equations from the standard or traditional Stokes to avoid confusion.

line 325: "implies the use of" --> "uses"
Done lines 327-336: This is a rambling paragraph that can be shortened to something like "As noted earlier we require the upper and lower surfaces of the glacier to be functions of the

August 28, 2024

horizontal coordinates x,y. That is, as expected in glacier modeling, overhangs are not permitted."

Thank you, this is better. Text has been changed.

line 344-348: Repetitive. Say *once* (earlier, presumably) that one could impose boundary conditions weakly, and that you won't do that.

Shortened, but did mention can use Lagrange multipliers, if desired.

line 360: Help the reader by referencing/comparing (23).

I have referenced (23) and (25) following (37).

lines 361 and 404: Separate these into 2 displays. (Or better, just be more efficient. Use vector notation?)

I have done it this way in an effort to be more compact (long paper!) I think it's quite clear that I have combined equations and boundary condition. Vector notation would not be good because the rest of the paper uses Cartesian tensors.

lines 437-439: This use of the continuity equation is completely mainstream in glaciology. It applies in all shallow theories including BP. (And the current manuscript illuminates it!) Please say this some other way.

This has been reworded.

lines 459-460: Again, deriving FE discretizations from variational principles is the normal way to do business. Why "except"?

My understanding is that the normal way to do FE business is by means of the weak formulation.

ine 475: There is no reason to use capital "U" here, and it is a source of confusion because capital U is used shortly in subscripts with a different meaning.

I have changed U to V.

line 495: "u, w, AND M_{UP}, M_{WP}"
Section 4.3: This section needs editing most. The main point of the entire paper is made in subsection 4.3.3, I believe. Roughly-speaking the main point is that, for the

August 28, 2024

transformed Stokes or EBP equations, the block M_{WP} must be invertible, thus square, when an extruded mesh with z-aligned cells is used. This point is buried after laborious and repetitive text. The main point of the paper *does* require a block-wise presentation of the Newton step equations, so the text will necessarily be somewhat technical, but it doesn't have to bury the main idea. There would seem to be no reason not to start a section with (47) and (48); the notation here is obvious. In any case, this reader had to get 600 lines into the document before getting to the key lines (roughly starting at line 596), and only then have an "oh ... that is what he is trying to say ..." moment.

lines 596-600: The main point of the paper, right? Which this reader appreciates! The blockwise form of the EBP model is therefore the central object of the paper, and could be put much earlier and more prominently.

Section 4.3 has been completely rewritten. I believe it may now address these comments.

lines 616-618: I would not permit my undergrad linear algebra students to say what is said here. The necessary condition is that *M_{WP} must be non-singular*, from which it *follows logically* that it must be square. The text literally says that non-singularity is "in addition" to squareness, thereby asserting that square matrices are invertible! (Line 1521 is worse.) Equation (56) could instead say "M_{WP} is non-singular"; one is allowed to put text in displayed LaTeX equations.

I have been careless here. In Section 4.3.2 it now says: "matrix $M_{WP}^T$ must be invertible and so it must be square and full rank. Since in general $M_{WP}^T$ is an $n_p \times n_w$ matrix, for solvability this requires that $n_p = n_w$".

Section 5: I think the paper would be improved by removing this section. I understand that the transformed Stokes model is the same as the Stokes model, and the EBP model is the same as the BP model. So recapitulating the ISMIP-HOM purpose, which is (I suppose) to examine how close BP results are to Stokes results, should not come out any differently here, and thus it is not worth doing. Of course it is true that different numerical approaches generate different results in detail. But what exactly should the reader know about this numerical comparison? Can this be summarized in a sentence or two?

I have shortened this section considerably, keeping the figures and only a minimum amount of text to describe them.

August 28, 2024

lines 778-780:  For efficiency I assume that BP is first used everywhere, then some criteria is applied, and then Stokes is used where the criteria applies.  But do you want to demonstrate that the Stokes calculation everywhere gives the nearly same criteria-satisfying region?

I think this is done visually in Fig. 8.  It is quite obvious that the Adaptive (AH) and Stokes (TS) calculations are quite close while the Blatter-Pattyn calculation is not very accurate in the details up through the column in the vicinity of the obstacle.

line 785:  Is the "counterintuitive" aspect of this explained by noting that the effective viscosity is often actually largest in the top of the ice column, which implies the greatest longitudinal and bridging stress transmission up there?  I often find that visualizing the effective viscosity, in these shear-thinning flows, illuminates where stresses de-localize the problem.

It is counterintuitive because I would have expected the Stokes calculation to be needed just in the vicinity of the obstacle and not far away at the top of the domain.  Your explanation is probably correct but it would need a more detailed analysis to verify than is justified in this paper.

line 811-813:  It is not the personal computer etc. which stops an analysis of the cost savings, but rather the lack of a performance model for the solver.  This could be added, but it requires a bit of thinking.

Yes, but a more realistic calculation on representative computer hardware would be able to provide believable information on cost savings.

Subsection 6.2 and Section 7:  This seems like tedious overkill.  If a reader gets the main points of the paper then they can probably imagine lagging the Newton iteration and/or dual grids and/or higher order.  In any case, another 300 lines are burned before the summary.  If these are important enough then they could be a separate paper?  Otherwise most readers won't have the endurance; really I don't.

In introducing the new transformation I stated that I wanted to bring out two of its applications (although there may be more): Adaptive switching and improved approximations that are more accurate than BP.  I think both are equally important.  Breaking it up into two papers is possible but it would lose some continuity.  Honestly, I would not have the stamina to do that.  Readers can always skip over parts that don't interest them.

August 28, 2024

Section 8 (Summary):  Too long.
Substantially shortened.

Appendix A-C:  On and on.

Appendix D:  The manipulations shown in (79) and (80) are again very close to the main novel point of the paper.  I see no reason why they can't be written into a new and prominent form which makes subsubsection 4.3.3 into the central material.

line 1521:  Again, please don't say that all square matrices are invertible.  (Literally the text says "the solvability condition [$n_u = n_p$] implies the invertibility of $M_{WP}$".  Just no.)

Appendices A and D eliminated.  Material from Appendix D shortened and transferred to subsection 4.3.2.  Sloppiness re matrix invertibility has been corrected.

August 28, 2024

[revised manuscript text omitted]

approximation in the adaptive switching method. The second approximation simply approximates the vertical velocity by discretizing the continuity equation on a coarser grid than the rest of the model.

The second innovation involves the introduction and use of finite element discretizations that feature a decoupled invertible continuity equation permitting the

[revised manuscript text omitted]

August 30, 2024

---

## Author Comment (AC2)

**Response to Reviewer RC2 – Prof. Christian Schoof**

**Egusphere-2024-1052 Article**

Author Responses in Blue

I thank Prof. Schoof for a detailed and insightful review!

**Overview**

The paper builds a unified variational formulation for Stokes flow and the somewhat misnamed Blatter-Pattyn model for ice flow. To get the trivia out of the way, I will call this the Herterich-Blatter-Pattyn model from this point onwards. History may be written by the winners, and Herterich clearly hasn't been one of them, *but* the first instance of the Herterich-Blatter-Pattyn model being formulated was (to my knowledge) the following paper:

- Herterich, K. 1987. On the flow within the transition zone between ice sheet and ice shelf, *in* Dynamics of the West Antarctic Ice Sheet. Proceedings of a workshop held in Utrecht, May 6–8, 1985, pp.!185–202. D. Reidel, Dordrecht,

This predates the commonly used Blatter (1995) and Pattyn (2003) references by eight and sixteen years, respectively. If we're going to fall into the trap of naming things after people, we probably owe it to ourselves to do that accurately. Not that I've been able to convince anyone of this point so far.

I was not familiar with this reference (I doubt many people are) so I looked into this model to compare it to the BP model. It turns out that the Herterich model is NOT equivalent to the Blatter-Pattyn model. Instead, it is the same as the BP+ approximation that's described in §6.2.1. I've attached a derivation of this result at the end after these responses, in an Appendix, and I've also added a short discussion in §6.2.1 regarding this. Needless to say, I will not be changing the name of the Blatter-Pattyn model. Thank you for pointing out this reference.

Back to the present. It took me a bit of effort in stripping away detail and side notes to understand that the primary advance of the paper is a numerical formulation in which one can dynamically switch between the simpler Herterich-Blatter-Pattyn and the more complete Stokes flow model. The latter applies to flows with arbitrary aspect ratios and abrupt changes in boundary conditions, while the former requires a "shallow" flow.

At face value, such a "switchable" model makes a lot of sense, since ice sheet flow is shallow in most places, but often contains boundary layers (at ice divides, ice stream margins, grounding lines etc) where the Stokes equations must be solved. The present paper uses the variational structure of both models to create a unified formulation, by re-writing the Lagrangian for the Stokes equations so that it takes the form of the Herterich-Blatter-Pattyn Lagrangian with a few extra terms retained. The unified formulation then consists of introducing a flag that activates or deactivates these Stokes correction terms. This is an intriguing idea and deserves to be published. I am not entirely convinced that

August 27, 2024

You are right, this is an important advance introduced in the paper. However, I also want to point out the additional innovations, such as the new approximations that improve on Blatter-Pattyn and the introduction and use of grids, such as P1-E0, that permit the solution of the continuity equation for $w(u,v)$. To this effect, I have rewritten Section 8 (I hope you don't mind that I used some of your words to do this).

This is an intriguing idea and deserves to be published. I am not entirely convinced that *The Cryopshere* is the right vehicle as there are a number of technical issues that deserve more thorough scrutiny; *Geoscientific Model Development* would have seemed a more appropriate journal in the EGU stable to use, but really I would have been inclined to go with something like *J. Comp. Phys.*.

I did consider Geosci. Model Dev. and J. Comp. Phys. Unfortunately, I was not able to comply with GMD's code availability requirement. JCP did not seem appropriate because the paper is too specialized to ice sheet modeling.

As far as I can tell, the modified Stokes Lagrangian does not alter the saddle point strucutre of the Stokes flow problem (in the sense that the solution maximizes the modified Stokes functional with respect to $\tilde{p}$ and minimizes with respect to $\boldsymbol{u}$). Still, the modified Stokes Lagrangian is a fairly ugly object that obscures the natural symmetries of the original Stokes flow Lagrangian and introduces a much larger null space to the elliptic operator. For the usual Stokes flow Lagrangian, that null space consists of rigid body rotations modulo any such motions that are precluded by the boundary conditions (by way of restrictions on the space of admissible functions). In the modified Stokes Lagrangian of the present paper, the vertical velocity $w$ can be changed by adding an arbitrary function of the vertical coordinate $z$ only while leaving the elliptic part of the Lagrangian uchanged; such functions of $z$ are then penalized through the incompressibility constraint.

That ugliness appears to be an unavoidable part of a unified formulation. It does however mean that you would probably not choose to use the modified Lagrangian for the analysis of general Stokes flow problems, except to unify the Stokes and Herterich-Blatter-Pattyn problem computationally. The paper can probably streamlined by focusing on that aspect at the exclusion of some of the more peripheral commentary, and the title of the paper could probably be made more informative.

You are quite right. I do not advocate the transformed Stokes formulation for the solution of general Stokes problems. As presented in the paper I view it primarily as a means to (a) "unify the Stokes and Blatter-Pattyn problem computationally", §6.1, and (b) as a means of developing new approximations that improve on BP, §6.2. On the other hand, the transformed Stokes model can perform better computationally than the standard Stokes model. Section 5 compares some computational results for the standard and transformed Stokes formulation. Fig. 3 shows that the transformed model is more accurate than the standard model for Test B (no-slip) calculations at all resolutions. However, this is probably a fortuitous result because the two cases in the Test D* (frictional sliding) calculations show similar accuracy.

In terms of pursuing that unification of Stokes and Herterich-Blatter-Pattyn models, there are two significant issues that I can see:

1. While the variational formulation of the Stokes flow problem *is* a saddle point problem, the same is not true of the Herterich-Blatter-Pattyn problem. The latter naturally wants to be solved as an unconstrained minimization problem, with the incompressibilty condition solved a posteriori for the vertical velocity component $w$. Enforcing incompressibility through a Lagrange multiplier as part of the variational formulation leads to problems that occupy most of the technical material in the paper: unlike the modified Stokes Lagrangian, the elliptic part of the Herterich-Blatter-Pattyn Lagrangian does not contain $w$ at all, which must instead be solved for through the (hyperbolic) incompressibility condition alone — for which the standard function spaces used in Stokes flow solvers are unsuitable. This is perhaps unsurprising, as the standard function spaces used in finite elements are problematic for hyperbolic equations, and discontinuous basis functions (as used in discontinuous Galerkin methods) might be preferable for $w$ in the Herterich-Blatter-Pattyn problem; they might at least alleviate issues such as $\nabla \cdot \boldsymbol{u} = 0$ being an overdetermined problem if we assume that $\boldsymbol{u}$ is represented by P1 basis functions, which work well for the force balance part of the Herterich-Blatter-Pattyn problem.[1]

   The main challenge identified in the paper (which motivates the choice of those P1–E0 soaces, and the others discussed in sections 6,7 and appendix C) is to make sure that the choice of basis functions for the "Blatter-Pattyn system" (that is, the Blatter-Pattyn equations and the incompressibility condition) can also be used for the modified Stokes system. Unfortunately, I think that there are some issues with how this has been addressed in the paper, and I suspect that there is a misunderstanding of what the inf-sup ("LBB") condition really does. More on this below.
* * *
   [1] By contrast, the "P1-E0" spaces advocated here go in a different direction: $w$ is still represented by piecewise linear, and therefore continuous, basis functions, but the incompressibility constraint is weakened through the choice of a "coarser" basis function for the Lagrange multiplier that enforces incompressibility, averaging over two adjacent elements.

I don't have a quarrel with what is said here so I will postpone a response to where the inf-sup condition is discussed later.

2. More practically, I am not convinced that the method developed here will be adopted widely. Consider this: unlike its depth-integrated variants developed by Richard Hindmarsh, the Herterich-Blatter-Pattyn requires a three-dimensional domain to be resolved. This leads to a large number of computational degrees of freedom. In fact, the only advantage relative to a Stokes flow model is that there are fewer variables to be solved for in *the same* computational domain, as you can solve for $(u, v)$ using the elliptic solver, and the compute $w$ and $p$ *a posteriori* if required. As far as I can tell, the Seroussi et al (2012, cited in the manuscript) tiling method makes use of that. The method proposed here, of solving for $(u, v, w, p)$ using the same basis functions for both models without explicitly using the simpler structure of the Herterich-Blatter-Pattyn model, seems to get rid of that last advantage. Which begs the question, why bpther, given that the Stokes model is preferable in terms of the physics it represents. Unfortunately, the last paragraph of the conclusion puts paid to any hope that the paper might address whether the new approach is actually going to be computationally competitive.

I think this comment refers to the use of the transformed model in a "unified formulation". There is no question that the BP model is much cheaper, although less accurate, than a full-Stokes model. This is the reason why BP rather than Stokes is used in several production code packages (e.g., ISSM, MALI, CISM). Depth-integrated models are still cheaper but even less accurate. The approximations presented in §6.2 are at the high end of this scale; they are more accurate and more expensive than BP but still cheaper than Stokes. There is a tradeoff between accuracy and computational cost so a choice has to be made depending on the application.

I would not say that the standard Stokes formulation is preferable in terms of the physics it represents. The two formulations represent exactly the same physics and provide the same solutions to a given problem. They are, after all, mathematically related by a transformation. They differ, if at all, by their numerical behavior.

The adaptive switching example of §6.1 makes use of the "Blatter-Pattyn system", i.e., solving for $(u, v, w, p)$, for simplicity and just to illustrate the method, even though this is more expensive than it needs to be. In a practical application it should be possible to switch to using the Blatter-Pattyn model instead, i.e., solving for $(u, v)$, and coupling to the Stokes model using $p = 0$ and $w = w(u, v)$ at the interface, even though this would involve more complicated programming. I have added some sentences to this

August 27, 2024

effect in the paper.

    To follow up on the first point above, I am concerned that appendix D is misleading. Apologies if this gets a little long below; I am not as much of an expert at this as I'd like to be, so it took me a bit more explanation. My understanding of the inf-sup or Ladyzhenskaya-Babuška-Brezzi condition for Stokes flow problems is the following. Take the Hessian matrix in (55) of the present paper

$$\mathbf{M} = \begin{pmatrix} \mathbf{A} & \mathbf{B} \\ \mathbf{B}^{\mathrm{T}} & \mathbf{0} \end{pmatrix} \tag{1}$$

and we have to solve $\mathbf{Au} + \mathbf{Bp} = \mathbf{f}$, $\mathbf{B}^{\mathrm{T}}\mathbf{u} = \mathbf{0}$ where $\mathbf{f}$ is the relevant residual of the Stokes equations. $\mathbf{A}$ is positive semi-definite, which makes the solution of this problem (if it exists) equivalent to a saddle point: to find

—

—

—

Long discussion of inf-sup condition

—

—

—

    I hope I have been clear to say that I may be wrong. If I am, the text of the paper should be more explicit about the technical issues that I have raised; it's currently fairly vague in spelling out what the "LBB" (inf-sup) condition actually is, what it does, and whether the basis functions described in sections 6–7 and appendix C satisfy the inf-sup condition. (Also, if you're going to introduce a concept like that, please spell out what the acronym "LBB" stands for before using it, and maybe cite the original places where it comes from.)
* * *
[3] since I would otherwise have $B(\boldsymbol{v}_h, q_h) = 0$ for all $\boldsymbol{v}_h$ and some $q_h$

[4] There is another issue in play here: the inf-sup condition provides sufficient conditions for a convergence, but not necessary ones. You might also decide that you don't care about convergence of the discretized pressure solution $\tilde{p}_h$, but only of $\boldsymbol{u}_h$, which may conceivably relax the conditions you need to impose. I am not in a position to comment on that, however — you need a real expert reviewer.

    I have eliminated Appendix D and rewritten Section 4 to bypass the difficult issue of the inf-sup condition. The inf-sup condition is relevant only so as to point out that the standard and transformed Stokes models are subject to it and that one must use one of the many inf-sup-stable elements available in the literature in the discretization.

    I hope I have made it clear that the inf-sup condition does not apply to problems using elements satisfying the "solvability condition" because they are no longer constrained problems since incompressibility is built-in when using $w(u,v)$.

August 27, 2024

My other main points would be the following (partially elaborated under "specific points")

1. The notation in the paper is fairly idiosyncratic, which made it more time-consuming for me to follow various pieces. There is a mix of standard subscript notation, a very much nonstandard "round bracket around subscript means projection onto the horizontal plane" variant to subscript notation, and the use of explicit component notation as in equations (7), (28) and (35) to write scalar quantities that should really be denoted by contractions over subscripts. The paper makes enough assumption about the reader's level of mathematics that I would recommend streamlining this as I don't imagine much of an audience who will be helped rather than hindered by the nonstandard notation. In particular, I would discourage the round bracket notation in favour of a more standard projection operator: if $P$ is the projection onto the horizontal plane, $P(v_1, v_2, v_3) = (v_1, v_2, 0)$, then you can simply replace $a_i$ by $P_i(\boldsymbol{a})$ and retain summation over indices from 1 to 3. I say this even though the author has used it previously: the use of nonstandard notation it may make the paper harder to decipher for numerical analysts, who should really be encouraged to delve deeper into the theory relevant to the paper.
The use of superscripts $^{(s)}$, $^{(b_1)}$ and $^{(b_2)}$ on surface normals is also redundant, as you specify the parts of the domain boundary to which the stated boundary conditions apply. Keep this as simple as possible, because it it certainly isn't simple.

The second invariant was written in subscript notation, $\dot{\varepsilon}^2 = \dot{\varepsilon}_{ij}\dot{\varepsilon}_{ij}/2$ , just before equations (7) and (28) but then expanded for clarity.

I prefer using the nonstandard horizontal-index notation, i.e., $u_{(i)}$ vs $P_i(u)$, because it is more compact. However, I've added a sentence to clarify this in the paragraph following Fig. 1.

I have removed superscripts on surface normals as per your suggestion. Thank you.

2. The paper is quite ungenerous to the prior literature. I can see the appeal of referencing only your own paper ("DPL" in the present case) as the default reference because you know its content extremely well, but many of the introductory concepts in this manuscript have been developed in other places, which remain uncited. That may put off other practicioners who should read this paper (and who might in fact get some sort of notification, alerting them to your paper if you were to cite them!). It's also unhelpful to any reader who wants to make sense of the field: the may conclude that, really, only DPL is relevant as a prior publication. In particular, I see almost no reference to the extensive numerical analysis literature on Herterich-Blatter-Pattyn and Stokes flow models in glaciology (except my ca. 2010 own effort in that direction, which does get a citation somewhere).

I am happy to add additional or more appropriate references. Any suggestions?

August 27, 2024

3. The "Improved Blatter-Pattyn or BP+ Approximation" in section 6.2.1. I am unconvinced that it makes sense to include this in the present paper, unles you want to "patent" the idea for eternity. (Who knows, maybe someone will in future refer this approximation by someone else's name, the same fate that befell Herterich?) The reason why I am unconvinced is that this material further breaks the flow of the paper, without being robust. The supposedly improved approximation remains an *ad hoc*, partial retention of higher order terms in the Stokes flow model, higher order being in sense of the aspect ratio. There is no theoretical justification for doing that, as a partial retention of higher order terms in a model comes with no guarantee of a reduction in model error. In fact, it can make the model error worse. We do see a reduction in model error for the "BP+" model ("HBP+" / "D"?) in the single test that the "improved" model has been subjected to (ISMIP-HOM Test B, figure 9 of the manuscript). However, that is not really a robust demonstration of an "improved" model. It's also unclear whether the supposedly improved model is really competitive relative to solving the full Stokes model in terms of the tradeoff between accuracy and computational effort. (See my second major point above.)

I wonder if this comment is relevant now that we understand that the BP+ approximation is the same as the Herterich model? I doubt that it's necessary to have a full scale analysis when introducing a new approximation. For example, the Blatter-Pattyn model did not have a scale analysis for 15 years until Schoof and Hindmarsh (2010).

**Specific points**

A number of specific points, some of which will replicate elements of what I've written above.

- p3, line 81: "finite element grid" — "grid" might be seen to imply regularity. "finite element basis functions"? I mean, closer scrutiny does indicate that the basis functions used here are restricted to quite regular meshes, so perhaps the terminology is not wrong here, but that required regularity actually needs to be discussed somewhere.

Changed "grids" to "discretizations"

- p3 line 84 "these two elements are so-named because they employ edge-based pressures" — I actually found that confusing. The basis function for pressure is not really defined on edges (since there are plenty of edges in the mesh that don't have a pressure defined on them). They are really P0 basis functions in which two adjacent triangles of the triangulation are assigned the same pressure.
  Also, to make it clear that you are inventing these basis functions, avoid the passive voice here. "I have named these E0 to indicate that pressure is defined on select element edges" or similar. It took me a while to realize that I would look in vain for references to these elements in the literature.

I have changed the wording as follows "these two elements are novel and are so-named because they employ pressures located on vertical grid edges"

- p 3 line 89 onwards " A conventional ice sheet Stokes model discretized on such a grid is numerically equivalent to an inherently stable positive-definite minimization (i.e., optimization) problem, as demonstrated in Appendix D. This is in contrast to the ubiquitous Stokes finite element practice of needing to use elements that satisfy the "inf-sup" or "LBB" condition for stability (see Elman et al., 2014, and the brief discussion in §4.3.1)." — This honestly confused me. If you're saying that current practice in solving the Stokes equations in ice sheet dynamics ("A conventional ice sheet Stokes model") is to use unconstrained optimization, then please provide a reference. It's very hard to do unconstrained optimization for Stokes flow as you need divergence-free basis functions. If you mean that *your new approach* is equivalent to constructing such a divergence-free basis, then say that instead. However, you really also need to show that the reduction of the single Newton iteration step to a problem for $(u_1, \ldots, u_N, w_1, \ldots, u_N)$ only in appendix D (taking the dimensionality argument for $\mathbf{M}_{WP}$ at face value) really is equivalent to a convergent solution (under mesh refinement) of the unconstrained Stokes optimization problem (that is, the problem in which the incompressibility constraint is directly imposed on admissible $\boldsymbol{u}$). I don't think this is entirely trivial, see also my discussion of the inf-sup criterion above. (Put this another way, for any set of basis functions that allows the problem to be solved, I must be able in principle to eliminate the discretized pressure variable in favour of retaining on the nodal values of velocity, even if I would not choose to do the matrix manipulations involved. By your logic, any such scheme should therefore be robust. Is that true?) I may be mistaken about what you're trying to say here, but it may therefore be worth finding a different way of saying it.

As mentioned on Page 5 of this response, I have eliminated Appendix D and rewritten Section 4 and I hope that this now addresses the issue. No, my approach is not equivalent to constructing divergence-free basis functions. In fact, I doubt that such a basis exists (Boffi et al. (2008), a new reference added in the paper). What I am trying to say is that given an arbitrary discrete horizontal velocity field $\left(u\right)$ on a grid that satisfies the solvability condition (i.e., square, full rank matrix $M_{WP}^{T}$), I can always find an "incompressible" velocity field $\left(u, w\left(u\right)\right)$. This, when substituted into a constrained functional (Jacobian?) for variables $\left(u, w, p\right)$ converts it into an equivalent unconstrained functional for $\left(u\right)$ alone. This is what I'm trying to say in the new Section 4.3.2. I have also reworded the end part of the Introduction to express this.

August 27, 2024

- p 4 line 116. $S = S_{B1} \cup S_{B2}$?

Changed

  - p 4 line 121–123. See above re: notation.

  - eqs (8), (10, (11), (13)–(15). The superscripts on your unit normals seem to be redundant. There is only one (outward-pointing) unit normal to each of these parts of the boundary.

Corrected, see Point 1 above.

  - p 6 line 174, "$f_i$ is a specified frictional sliding force vector." — $f_i$ has the wrong units for a force. It's a traction, but why not just call it an interfacial shear stress? $\tau_i$ would probably be a more common symbol.

Changed throughout the paper except that I used the symbol to $\tau_i^S$ to avoid confusion with the stress tensor $\tau_{ij}$.

  - p6, line 170 "the simplest representation" — I beg to differ. There is a standard way of writing shear stress, as the traction $\Sigma_i = \sigma_{ij} n_j$ minus the normal component of traction $n_i \Sigma_j n_j$, or

$$(\delta_{ij} - n_i n_j)\sigma_{jk} n_k.$$

That is equation (71) in appendix A, and I think you'd be well advised to just use that (standard) form. (The point about weak solutions is surely that you never need to actually evaluate the shear stress itself from the stress tensor and the normal to the surface; you just need to know the constitutive relation, which here is just

$$(\delta_{ij} - n_i n_j)\sigma_{jk} n_k. = \beta u_i.$$

I think what the paper says isn't wrong, but manipulations don't seem they will help the unwitting reader, who is just wrapping their head around the basic model formulation. Especially at this point, where you haven't motivated your use of the bracketed indices at all yet, in terms of the mathematics you're doing.

  - page 13, equation (32). As above, that is pretty ugly. I can't tell if there is a less confusing form, but might be worth trying.

You are right! The form I was using is computationally easier because it eliminates the complicated quantity $\tau_n = n_i \tau_{ij} n_j$, the normal component of the stress force. But if the discretization is based on a variational principle (or on a weak formulation) this is irrelevant because this quantity never needs to be explicitly calculated. One might as well eliminate Appendix A (shortens the paper!) and obtain the tangential frictional shear stress as in DPL (2010) or your expression above, except that I prefer to use the

August 27, 2024

deviatoric stress tensor as in the rest of the paper. Equations (11) and (32) have been updated.

- p 7 line 180 "...can easily be added" → "...can esaily be added to equation (10)"?

Done

- p 8 line 218 " Here we use $z_b$ as a shorthand notation for $z_b(x, y)'$ — this is surely redundant? Same on line 259.

Removed in both places.

- p 7 "(see DPL, 2010, for a fuller description of the variational principle applied to ice sheet modeling)" — you should refer to Chen, Q, M. Gunzburger and M. Perego. 2013. Well-posedness results for a nonlinear Stokes problem arising in glaciology. *SIAM J. Math. Anal.*, 45(5), 2710-2733. This is probably the most comprehensive analysis of Stokes flows with sliding in glaciology (and you'll find the relevant functional there, too, naturally).

Reference added

- p 7 line 200 "...(see Schoof, 2010, in connection with the Blatter-Pattyn model)" — I feel a little unfairly singled out here. There are earlier papers by Coling and Rappaz (MSAN, 1999), Glowinski and Rappaz (2003), Chow et al 92004) Rappaz and Reist (M3AS, 2005) that deserve an equal mention as having contributed to the analysis of the Herterich-Blatter-Pattyn model

I've inserted "as well as earlier references contained therein"

- p8 line 208 " As in DPL (2010), arguments enclosed in square brackets, here $u_i$ , $P$, $\lambda_i$ ,$\Lambda$ , indicate those variables that are used in the variation of the functional." — this is an odd way of putting it. A functional is simply a mapping (or function) from some vector space (or perhaps an affine space) into the real numbers. So why not just say that you're enclosing the arguments of the functional (which is a function and therefore takes arguemnts like any other function![5]) in square brackets.
* * *
[5]Unless you insist that a function have to take arguments in $\mathbb{R}^n$ for some finite $n$.

I have rephrased it as "As in DPL (2010), arguments enclosed in square brackets, here $u_i, P, \lambda_i, \Lambda$ , indicates those functions that are subject to variation as arguments of the functional ". Hope this helps.

- page 9 line 254"...as is done in the Blatter-Pattyn approximation (see DPL, 2010)."— is the apropriate reference not one of Herterich (1987), Blatter (1995) or Pattyn (2003(?

Yes, replaced by Pattyn (2003). I couldn't actually find it in Blatter (1995).

- page 12 line 296. It would be a good idea to explain the role of the flag parameters $\xi$ and $\hat{\xi}$ earlier. I spent a page half guessing what they were.

August 27, 2024

The dummy variables are now explained directly below where they are introduced, i.e., right after equations (23)-(28).

- p 8 line 210 onwards. I don't think this correct, or at least, it seems misleading. In my understanding, there are two ways of imposing Dirichlet conditions: by Lagrange multiplier, or by restricting the space of admissible functions. The latter is not the same as the kind of explicit substitution you're doing here. If I take the functional defined in equation (15) and I *don't* separately impose the constraint $\boldsymbol{u} \cdot \boldsymbol{n} = 0$ at $\S_b$ on the arguments $u_i$, then I don;t see that taking the first variation of the functional will recover that constraint. If you impose the constraint $\boldsymbol{u} \cdot \boldsymbol{n} = 0$, then you do not need to make the substitution in equation (14) in the friction term of the Stokes Lagrangian
  The substitution here is again quite nonstandard, with little advance warning or real motivation. I think you're really trying to lay the groundwork for the modified Stokes variational principle later (by eliminating $w$ where that's going to be necessary) but here is an awkward place to do it, unless you explain why you're eliminating $w$.

Again an oversight on my part! I should have pointed out that direct substitution is possible only in the discrete formulation of the functional where boundary variables are directly accessible (except of course in the surface integral terns where surface values are accessible even in the continuous formulation). I have tried to clarify this in the text.

- page 9 line 232 "...the specified values of velocity are then obtainable a posteriori from (9) or (14)" — this is a strange way of putting it. What is a specified value? One that is prescribed? I don't think that's what you mean. I think you mean that the taking the first variation does not recover the boundary condition $\boldsymbol{u} \cdot \boldsymbol{n} = 0$, see previous point. However, you can't impose that "a posteriori" since you cannot solve the Euler-Lagrange equations for the functional in (15) without using the Dirichlet condition on velocity. The standard way of putting this (I believe) is to restrict the functional to the vector space of suitable smooth velocities (in $[W^{1,1+1/n}(\Omega)]^3$) *that satisfy the Dirichlet condition.*

Yes, I do mean the prescribed values. I have changed the wording in the text to hopefully make this clearer.

- page 10, line 265 "The standard Stokes pressure $P$ is some three orders of magnitude larger than the transformed pressure $\tilde{P}$" — asymptotic analysis of the problem will show that, more generally, the "Stokes correction" $\tilde{P}$ scales as $O(\varepsilon)$ when there is signfiicant sliding, where $\varepsilon$ is aspect ratio, and of $O(\varepsilon^{1+1/n})$ in the absence of sliding (see the Schoof and Hindmarsh 2010 reference).

My main objective here was to highlight the absence of the large hydrostatic pressure component in the "Stokes correction".

- page 15 line 376 Remark # 1: seems like splitting hairs, especially as you wouldn't bother with this if you wrote down the weak form.

August 27, 2024

Maybe, but I think it's useful to point out that the presented BP equations differ, if only slightly, from the original.

- p 13 line 353 "The standard (or traditional) Blatter-Pattyn approximation (originally introduced by Blatter, 1995; Pattyn, 2003; later by DPL, 2010; Schoof and Hewitt, 2013)" — as above, Herterich (1987) was the person who introduced this. I don't think the Schoof and Hewitt review paper did much more than describe the theory, as oppposed to contributing to it. If I did anything here, then Schoof and Hindmarsh (2010) provided the first self-consistent asymptotic analysis of the model. If you want to list DPL for its description of the variatoinal formulation, you probably ought to cite the numerical analysis papers I have listed above (Coling and Rappaz, Glowinski and Rappaz, Chow et al, Rappaz and Reist; full citations in my 2010 paper that is in the reference list), who previously dealt with the same variational formulation.

Herterich is not pertinent since he did not introduce BP, just BP+. I've added "and references therein" to Schoof and Hewitt, 2013. Hope this is adequate. I'm reluctant to add a lot of extra references.

- page 15 line 419 "In summary, the extended Blatter-Pattyn model, (40)-(42), is equivalent to the standard Blatter-Pattyn model, (36), for the horizontal velocities, u,v , except that it also includes two additional equations that determine the pressure P ! and the vertical velocity w , which are usually ignored in the standard Blatter-Pattyn approximation when only the horizontal velocity is of interest. Because of this, we distinguish between the Blatter- Pattyn model that solves for just the two horizontal velocities (i.e., the standard Blatter- Pattyn approximation, (36)), and the Blatter-Pattyn system that solves for all the variables (i.e., the extended Blatter-Pattyn approximation, (40)-(42))." — again, this seems like splitting hairs. Anyone who needs to solve for temperature in an ice sheet (which is a standard part of any ice sheet simulation code) has to solve for the vertical velocity component, so I don't think insisting on a difference between the Blatter-Pattyn model and system is helpful. Solving for $\tilde{P}$ is not particularly relevant as we know $\tilde{P} = 0$ for the Blatter-Pattyn model. For the sake of not going down rabbit holes, I would omit this.

- p 17 line 437 "The use of the continuity equation to solve for the vertical velocity w is a novel feature of the Blatter-Pattyn approximation since the continuity equation is not normally used for this purpose." — as per my previous comment, I don't think this is true or novel. The vertical velocity component is a pretty important quantity to be able to compute in ice sheet codes.

- p 17 lie 439 onwards. The use of a depth-integrated mass conservation equation is a boilerplate approach for thin film models, so a lengthy discussion without citation seems unnecessary, especially as this doesn't really lead anywhere in developing the novel material in the paper.

I have rewritten the relevant sections and shortened them considerably. Hope this resolves the issue.

August 27, 2024

- p 17 line 482 "Additional details about the grid and the associated discretization..." — I flagged this above (grid versus mesh), but again: I think it's worth pointing out somewhere that some of the basis functions developed here really do require a regular mesh (grid?) rather than an unstructured triangulation. For the P1-E0 basis functions, I have to be able to match every triangle with precisely one neighbour, not leaving any neighbourless triangles. I don't think that's possible with every fully unstructured mesh. (?)

I think this is already done in Appendix B and especially in Fig. B2, which shows that the P1-E0 element is quite flexible and can be used with quite general triangulations. (Note: Appendix A has been eliminated so all remaining appendices have been renumbered.)

- page 19 line 504. Italicize the variables $u$ and $w$.

Done

- page 21 line 526, "...or else they are "saddle point" problems since the Hessian matrix $M(u, w)$ is symmetric but indefinite, with both positive and negative eigenvalues..." — you can be more definitive about this I think: the "saddle point" terminology refers to the fact that there really is a saddle point, where you mimiize the quadratuc form associated with the matrix with respect to the velocity variables and maximize with respect to pressure.

I have added a sentence to this effect in the text.

- p 22 line 565 Please define the acronym "LBB" before using it.

Done on p. 22 in Section 4.3.1

- p 23 line 601 Taylor-Hood elements have gone from being an illustration of an element that satisfies the inf-sup condition (line 569) to becoming the standard reference for stable elements. Is it true in general that all elements that satisfy the LBB condition for the Stokes problem will leave $M_{WP}^{T}$ non-invertible on dimensional grounds alone? If so, that is important to point out here.

Pointed out in Section 4.3.1

- Sections 5.1–5.3 You're focusing on velocity solutions here. If there are stability issues with your choice of basis functions, I'd expect these to show up in the pressure solution. Can we see resultss for some of those? Obviously not relevant for Herterich-Blatter-Pattyn, but for the transformed and original Stokes flow problems. Especially pressure along the bed would be useful, but also convergence or lack thereof.

I have added the following at the end of Section 5: "Pressure results are not shown because pressure, particularly in the transformed case, has little or no physical significance. However, pressure calculated on the P1-E0 grid is particularly smooth and well behaved."

August 27, 2024

- Figure 7: this is fine, but more convincing would be a grounding line, where you have to worry about a normal stress constraint determining the grounding line location (and that normal stress is in general not cryostatic as assumed by Herterich-Blatter-Pattyn)

That's true, but it's too late to make such a major change

**Appendix - Identification of the Herterich model**

From §3.4.1 of the present paper the 2D Blatter-Pattyn model is given by

$$\frac{\partial}{\partial x}\left(4\mu_{BP}\frac{\partial u}{\partial x}\right)+\frac{\partial}{\partial z}\left(\mu_{BP}\frac{\partial u}{\partial z}\right)-\rho g\frac{\partial z_s}{\partial x}=0, \tag{1}$$

where the effective viscosity is

$$\mu_{BP}=\eta_0\left(\dot{\varepsilon}_{BP}^2\right)^{(1-n)/2n}, \tag{2}$$

and the second invariant is

$$\dot{\varepsilon}_{BP}^2=\left(\frac{\partial u}{\partial x}\right)^2+\frac{1}{4}\frac{\partial u}{\partial z}^2. \tag{3}$$

After allowing for minor changes in notation, this is the same as found in Pattyn (2003), Schoof (2010), and DPL (2010).

The Herterich (1987) system, corresponding to (1)-(3), consists of his equations (2.11) and (2.12) which, using the present notation (i.e., x, u in the horizontal and z, w in the vertical), may be written as follows

$$\frac{\partial}{\partial x}\left(4\mu_H\frac{\partial u}{\partial x}\right)+\frac{\partial}{\partial z}\left(\mu_H\left(\frac{\partial u}{\partial z}+\frac{\partial w}{\partial x}\right)\right)-\rho g\frac{\partial z_s}{\partial x}=0,$$

$$\frac{\partial u}{\partial x}+\frac{\partial w}{\partial z}=0, \tag{4}$$

where

$$\mu_H=\frac{f}{2A^{1/n}}=\frac{1}{2A^{1/n}}\left(\dot{\varepsilon}_H^2\right)^{(1-n)/2n}, \tag{5}$$

and

$$\dot{\varepsilon}_H^2=\left(\frac{\partial u}{\partial x}\right)^2+\frac{1}{4}\left(\frac{\partial u}{\partial z}+\frac{\partial w}{\partial x}\right)^2. \tag{6}$$

Comparing (5) to (2), we can identify $\eta_0$ with $1/2A^{1/n}$. Thus, comparing Herterich's model, (4) and (5), to the Blatter-Pattyn model, (1)-(3), we see that the two models are

August 27, 2024

NOT the same because of the presence of the $\partial w/\partial x$ terms in the Herterich model.

(Note: There is a typo in Eq. (2.10) of the Herterich paper, i.e., $\partial v/\partial z$ should be $\partial v/\partial x$.)

Now, consider the BP+ approximation as given by the two functionals (62)-(64) from §6.2.1 in the paper. Ignoring the boundary condition terms since they don't matter for the present purpose, the functionals may be written as

[revised manuscript text omitted]

approximation in the adaptive switching method. The second approximation simply approximates the vertical velocity by discretizing the continuity equation on a coarser grid than the rest of the model.

The second innovation involves the introduction and use of finite element discretizations that feature a decoupled invertible continuity equation permitting the

August 30, 2024

numerical solution for the vertical velocity in terms of the horizontal velocity components, i.e., $w = w(u,v)$. Some examples of such grids for use in 2D and 3D are given in Appendix B. An important example is the P1-E0 grid that is used in most of the test problems in this paper. However, one can alternatively obtain $w = w(u,v)$ by other means, as for example by discretizing (43). For example, this is done in MALI (Hoffman et al., 2018), a code based on the Blatter-Pattyn approximation, to obtain the vertical velocity $w$ for the advection of ice temperature (Mauro Perego, private communication).

Finally, no cost comparisons have been presented because the present calculations are only proof of concept, 
[revised manuscript text omitted]

---

## Referee Report (RR1)

A Novel Transformation of the Ice Sheet Stokes Equations and Some of its Properties and Applications

by J. Dukowicz

Re-review by Christian Schoof, University of British Columbia

**Higher level comments**

This paper has improved in readability from the original submission. There do remain a number of significant issues to be addressed. I have to say that this is the first time I've found myself reviewing the "response to reviewers" as much as the revised paper, so don't take the points I make lightly.

Before I get started, let me say this: decisions on manuscripts are obviously up to the editor, but there are two issues without which I won't be endorsing publication of this paper.

The first is adequate citation of prior literature. My general comment about the manuscript being ungenerous to the prior literature resulted in the following line in the response to reviewers

*I am happy to add additional or more appropriate references. Any sugggestions?*

but several pages later, when I actually list the relevant references, you say

*I'm reluctant to add a lot of extra references*

Most scientists find out sooner or later that it's easier in the review process to follow the path of least resistance when being asked to cite additional papers. If those are the reviewer's papers, you have a point in questioning their motives, but if they are third party papers and you try to avoid referencing them, you risk appearing as though you'd rather not acknowledge them. In the present manuscript, that unwillingness seems to focus on relevant literature on variational formulations that predates the early 2010s. That is, coincidentally, when Dukowicz, Price and Lipscomb ("DPL 2010") was first published, which you are using as the standard reference for variational formulations.

The relevant citations are (and this is simply the most important subset!) J. Colinge and J. Rappaz, A strongly nonlinear problem arising in glaciology, Math. Model. Numer. Anal. 33 (1999) 395–406.

R. Glowinski and J. Rappaz, Approximation of a nonlinear elliptic problem arising in a non-Newtonian fluid flow model in glaciology, Math. Model. Numer. Anal. 37 (2003) 175–186.

J. Rappaz and A. Reist, Mathematical and numerical analysis of a three-dimensional fluid flow model in glaciology, Math. Model. Mech. Appl. Sci. 15 (2005) 37–52.

The obvious place for these is line 334 just before eq (34). This is first and foremost a matter of respect. If that is too many papers, leave out Glowinski and Rappaz.

The second is the naming of the "BP+ model". As you point out, Herterich's (1987)

model differs from Blatter-Pattyn. ("Needless to say, I will not be changing the name of the Blatter- Pattyn model.") But the Herterich model *is* what you now want to call "BP+", even though Herterich developed it nearly forty years ago, the best part of a decade before Blatter. It was *not* developed from the BP model, which is precisely what calling it the "BP+" model would suggest to future generations. The person who actually came up with it deserves a little more respect.

I have two main scientific points, some of which are repeated in greater detail in the line-by-line comments (which I wrote prior to this summary)

1. The inf-sup condition: Your response to reviewers states
   *I have eliminated Appendix D and rewritten Section 4 to bypass the difficult issue of the inf-sup condition. The inf-sup condition is relevant only so as to point out that the standard and transformed Stokes models are subject to it and that one must use one of the many inf-sup-stable elements available in the literature in the discretization.*
   *I hope I have made it clear that the inf-sup condition does not apply to problems using elements satisfying the "solvability condition" because they are no longer constrained problems since incompressibility is built-in when using $w = (u, v)$*
   This is deeply problematic as a rationale for not addressing the inf-sup condiiton. For one thing, you *do not* state ambiguously that you regard the ability to formulae the problem as an unconstrained minimzation problem as a reason for not addressing whether your novel finite elements satisfy the inf-sup condition. But worse is that the argument given above is incorrect.
   First of all, Stokes flow problems of the type discussed here are *always* equivalent to an unconstrained minimzation problems. See Chen et al 2013 for the continuum case. This reamins true for the discretized version so long as the discrete "divergence" operator given by $B^T$ in equiation (55) has a non-trivial right nullspace, meaning there are vectors $\mathbf{u} \neq \mathbf{0}$ such that $B^T \mathbf{u} = \mathbf{0}$. Being able to find such vectors does not require the solvability condition (57) to hold, or even the usual inf-sup condition to hold (more on this shortly), but the existence of such a nullspace is assured for any set of basis functions satisfying the inf-sup condition. When the nullspace exists, the unconstrained minimization problem is over that nullspace. I have written out additional detail on this in thespecific comments on lines 611 and 617.).
   Second, there is no reason why the equivalence with an unconstrained minimization problem should get around having to address whether your new basis functions satisfy the inf-sup condition. First, let me state a subtely to the inf-sup condiiton, which I touched on in my previous review: I can impose the inf-sup condition at a particular discretization level, sot hat the inf-sup bound (usually "$\beta$) exists for that disretization level, and hence my discretized problem is equivalent to a finite-dimensional unconstrained minimization, but that doesn't mean that I will get correct solutions.

For the partial differential equations, I have to make sure that the inf-sup (usually "$\beta$) does not depend on the discretization level, usually expressed in terms of the maximum element size $h$. This ensures that as you take finer and finer meshes, the discretized solution converges to a solution of the continuum problem, while you have no such assurance if $\beta = \beta_h$ depends on $h$ and $\beta_h \to 0$ as $h \to 0$ (so long as your unconstrained finite element basis can adequately approximate all functions of interest in the limit $h \to 0$, see below).

If you instead treat your discretization as defining a divergence free basis spanning a finite-dimensional subspace (the right nullspace of $B^T$) over which you do unconstrained optimization, then nature of the problem remains the same: you have to show that solutions converge to the continuum solution as $h \to 0$. It is only the way that you cast the problem that changes: instead of showing your basis functions satisfy the inf-sup condition with a $\beta$ indepndent of $h$, you now need to determine whether you new, divergence-free basis functions (which are a subset of standard piecewise linear or quadratic bases) can still approximate all functions in the admissible space

$$V = \{v \in [W^{1+1/n}(\Omega)]^3 : \mathrm{div}\, v = 0 \,\text{and}\, v = 0 \,\text{on}\, \partial\Omega_{b_1},\, v \cdot n = 0 \,\text{on}\, \partial\Omega_{b_2}\},$$

arbitrarily well in the limit $h \to 0$. That is *not* trivial; if it was, it's unclear anyone would ever have bothered with the inf-sup condition because it's much easier to find finite element functions for which there is an $h$-dependent $\beta_h$ for which you have equivalence to a finite-dimensional unconstrained minimizatoin problem but no guarantee of convergence as $h \to 0$, than it is to find basis functions for which $\beta$ is independent of mesh size. See the line-by-line comment for line 617.

This brings me back to: you talk about the inf-sup condition, but you still do not explicitly address the question of whether your novel finite element basis functions satisfy the inf-sup condition, and if they don't, exactly why that should not matter. See above re the unconstrained minization idea being a red herring. There are furhter specific notes related to this in the comments on line 544–617. I also note that your off-hand comment on line 1006 suggests that at least some of the finite element bases that you are constructing don't satisfy the inf-sup condition, so this isn't a flippant point to make, and you should at least point out there that the P2-E1 space is therefore likely not to satisfy the inf-sup condition, but your numerics give you hope that perhaps the solution in general does converge for the velocity field, if not the pressure field. (By "in general", I mean )

2. The Herterich ("BP+") model. I flagged in my original review that this is an *ad hoc* model for which there is no theoretical justification in terms of asymptotic error estimates; it just turns out to work well for the test cases you have run. Your response to reviewers says *I doubt that it's necessary to have a full scale analysis when introducing a new approximation. For example, the Blatter- Pattyn model did*

*not have a scale analysis for 15 years until Schoof and Hindmarsh (2010).*
I read that to say it is unreasonable and unnecessary to expect a scaling analysis, and that this would somehow be difficult, and that you do not wish to discuss the issue. Let me do the scaling analysis for you, in that case; you will need to discuss this when introducing the model 6.2.1 (where the section heading claims Herterich to be an improved Blatter-Pattyn model).

The Herterich model as you state it in equations (67)–(68) can be derived from the following form of the Stokes model, with terms selectively removed:

$$\frac{\partial \tau_{xz}}{\partial z} + \frac{\partial \tau_{xx}}{\partial x} - \frac{\partial p}{\partial x} = 0$$

$$\frac{\partial \tau_{zz}}{\partial z} - \frac{\partial p}{\partial z} = -\rho g$$

$$\frac{\partial u}{\partial x} + \frac{\partial w}{\partial z} = 0$$

$$\tau_{xz} = \eta \left( \frac{\partial u}{\partial z} + \frac{\partial w}{\partial x} \right) \quad \tau_{xx} = 2\eta \frac{\partial u}{\partial x}$$

$$\tau_{zz} = 2\eta \frac{\partial w}{\partial z}$$

subject to

$$\tau_{xz} - \frac{\partial z_s}{\partial x}(\tau_{xx} - p) = 0$$

$$\tau_{zz} - p = 0$$

at $z = z_s$, and appropriate basal boundary conditions that I won't write out in detail.

Assume for simplicity that eta is constant; the shear-thinning power law fluid case of Glen's law is only superficially different. If I take the full Stokes model instead and apply a standard "shallow ice" scaling, I obtain (see e.g. section 3.6 of Schoof and Hindmarsh (2010), although the original references would be Fowler and Larson

(1978), Morland and Johnson (1980) and Hutter (1983))

$$\frac{\partial \tau_{xz}^*}{\partial z^*} + \epsilon^2 \frac{\partial \tau_{xx}^*}{\partial x^*} - \frac{\partial p^*}{\partial x^*} = 0$$

$$\epsilon^2 \frac{\partial \tau_{xz}^*}{\partial z^*} + \epsilon^2 \frac{\partial \tau_{zz}^*}{\partial z^*} - \frac{\partial p^*}{z^*} = -1$$

$$\frac{\partial u^*}{\partial x^*} + \frac{\partial w^*}{\partial z^*} = 0$$

$$\tau_{xz}^* = \frac{\partial u^*}{\partial z^*} + \epsilon^2 \frac{\partial w^*}{\partial x^*}$$

$$\tau_{xx}^* = 2\frac{\partial u^*}{\partial x^*}$$

$$\tau_{zz}^* = 2\frac{\partial w^*}{\partial z^*}$$

subject to

$$\tau_{xz}^* - \frac{\partial z_s^*}{\partial x^*}(\epsilon^2 \tau_{xx}^* - p^*) = 0$$

$$\epsilon^2 \tau_{zz}^* - p^* - \epsilon^2 \frac{\partial z_s^*}{\partial x^*}\tau_{xz}^* = 0$$

where the asterisks denote scaled variables, and red colour marks the terms that are not retained in the Herterich model. It should be apparent that the Herterich model retains *some* of the $O(\epsilon^2)$ (the $\partial w^*/\partial x^*$ term in the definition of $\tau_{xz}^*$ and the full slope term in the shear stress boundary condition. It retains by no means all $O(\epsilon^2)$ terms, so there is no reason why the Herterich model should be any better in this parametric limit (with respect to sliding) than the BP model, which also selectiely omits $O(\epsilon^2)$ terms and therefore has an $O(\epsilon^2)$ error.

We can conversely turn to the limit of fast sliding. Using the appropriate scaling for that case (e.g. Schoof and Hindmarsh (2010) section 3.4, although the original

source for this goes back to MacAyeal (1989) and beyond)

$$\frac{\partial \tau_{xz}^*}{\partial z^*} + \frac{\partial \tau_{xx}^{**}}{\partial x^*} - \frac{\partial p^*}{\partial x^*} = 0$$

$$\epsilon^2 \frac{\partial \tau_{xz}^*}{\partial z^*} + \frac{\partial \tau_{zz}^{**}}{\partial z^*} - \frac{\partial p^*}{\partial z^*} = -1$$

$$\frac{\partial u^{**}}{\partial x^*} + \frac{\partial w^{**}}{\partial z^*} = 0$$

$$\epsilon^2 \tau_{xz}^* = \frac{\partial u^{**}}{\partial z^*} + \epsilon^2 \frac{\partial w^{**}}{\partial x^*}$$

$$\tau_{xx}^{**} = 2\frac{\partial u^{**}}{\partial x^*}$$

$$\tau_{zz}^{**} = 2\frac{\partial w^{**}}{\partial z^*}$$

subject to

$$\tau_{xz}^* - \frac{\partial z_s^*}{\partial x^*}(\tau_{xx}^{**} - p^*) = 0$$

$$\tau_{zz}^{**} - p^* - \epsilon^2 \frac{\partial z_s^*}{\partial x^*}\tau_{xz}^* = 0$$

where the variables with double asterisks denotes dimensionless variables that have been rescaled from the shallow ice case. Once more, the Herterich model retains some of the $O(\epsilon^2)$ terms, but not all, and will again only be accurate to an $O(\epsilon^2)$ error, the same as the Blatter-Pattyn model.

We can also look at the intermediate sliding regime (Schoof and Hindmarsh (2010), section 2.1, formally with lambda = 1), in which case

$$\frac{\partial \tau_{xz}^*}{\partial z^*} + \epsilon\frac{\partial \tau_{xx}^{***}}{\partial x^*} - \frac{\partial p^*}{\partial x^*} = 0$$

$$\epsilon^2 \frac{\partial \tau_{xz}^*}{\partial z^*} + \epsilon\frac{\partial \tau_{zz}^{***}}{\partial z^*} - \frac{\partial p^*}{\partial z^*} = -1$$

$$\frac{\partial u^{***}}{\partial x^*} + \frac{\partial w^{***}}{\partial z^*} = 0$$

$$\epsilon \tau_{xz}^* = \frac{\partial u^{***}}{\partial z^*} + \epsilon^2 \frac{\partial w^{***}}{\partial x^*}$$

$$\tau_{xx}^{***} = 2\frac{\partial u^{***}}{\partial x^*}$$

$$\tau_{zz}^{***} = 2\frac{\partial w^{***}}{\partial z^*}$$

on $\Omega$, subject to

$$\tau_{xz}^* - \frac{\partial z_s^*}{\partial x^*}(\epsilon\tau_{xx}^{***} - p^*) = 0$$

$$\epsilon^*\tau_{zz}^{***} - p^* - \epsilon^2\frac{\partial z_s^*}{dx^*}\tau_{xz}^* = 0$$

Same story. The Herterich model selectively retains $O(\epsilon^2)$ terms and offers no improvement in terms of asymptotic error over the Blatter-Pattyn model.

That should settle the case — if you want a single model with a better asymptotic error (that is, size of error in the limit of $\epsilon \to 0$ than Blatter-Pattyn, you really have to retain all $O(\epsilon^2)$ terms. In order to do that regardless of sliding regime, it should be clear you actually have to retain all terms of the full Stokes model (since the smallest term contained in all the scaled versions of the Stokes flow model above is $O(\epsilon^2)$, there is really nothing that you can omit if you want a model accurate to $O(\epsilon^2)$).

The fact that the Herterich model performs better than BP in specific tests you have computed is really not much of a guide to anything in general: if you had a general result predicting a smaller error, then something like figure 9 would be a welcome illustration. In the absence of a much more general investigation of model error, the smaller error here might as well be fortuitious and specific to some aspect of test B. (I would also note that figure 5 shows resutls for a fairly large epsilon, so we are not looking an approximation based on small aspect ratio: an aspect ratio of 1:5 really cannot be treated as small in general, as finite depth effects typically appear for aspect ratios around $1 : (2\pi)$)

**Line-by-line comments**

- eq (11): there is still no need for the brackets around the (i) subscripts. In fact, by putting them into equation, you actually need to add the statement that $\tau_i^S n_i = 0$ since (11) is mute about the value of its $z$-component. If you write instead that

$$\tau_i^S = \tau_{ij}n_j - \tau_n n_i$$

you automatically ensure that $\tau_i^S n_i = \tau_{ij}n_i n_j - \tau_n n_i n_i = \tau_n - \tau_n = 0$. That also tidies up your notation.

- line 188 "references contained therein in connection with the Blatter-Pattyn model" — No, not enough. See start of this re-review

- line 203: "However, this can only be done in the discrete formulation of the functional since only then are boundary values of velocity accessible (although they are always accessible in the surface integral terms)." This either not true or misleading,

depending on what you mean by "accessible". Infinite-dimensional variational formulations of pde problems impose homogenous Dirichlet conditions by restricting the function space on which the problem is posed, rather than using Lagrange mutlipliers. In the present case, no need to get technical about that function space, but simply delete this statement and state that the minimization problem needs to be restricted first arguments of the functional A that satisfy the Dirichlet conditions

- eq (15) The (b2) superscript on the second line is undefined and doesn't want to be there. Also, this is needlessly complicated: the form of the surface integral is exactly the same as

$$\int_{S_{B_2}} \frac{1}{2} \beta u_i u_i \mathrm{d}S,$$

but more complicated. And it's not like you've replaced $u_3$ in the rest of the formula yet, so why make it more complicated in the boundary term. (This is different from (33), where the substitution makes more sense.)

- eq (17) seems redundant. If you cannot see follows from (16) and (18), you're going to struggle mightily with the rest of this paper.

- eq (26) Again, write in standard subscript form?

- line 287 "remains positive" $\rightarrow$ "remains non-negative"

- line 289 "The dummy variables..." — I don't think this is the usual meaning of "dummy variables" (like the variable with respect to which you integrate in a definite integral). They are indicator terms, taking values of 1 or 0. Same in line 344.

- eq (29) — this is a bit misleading; your earlier text (line 289, "The dummy variables $\xi, \hat{}$ in (23)-(25) and (26)-(29) are used to identify terms that are neglected in the two types of the Blatter-Pattyn approximation discussed in §3.4") indicates that all I need to do in order to obtain Blatter-Pattyn is to set $\xi = \hat{\xi} = 0$.

- Remark #1 — this isn't about "computational savings", is it? The small slope approximation is what makes Blatter-Pattyn valid in the first place, and the error in replacing the normal by its small slope approximation is the same order as the error in the Blatter-Pattyn model in the first place. Besides — and you might want to spell this out clearly — the difference between the two amounts to replacing $\beta(x)$ by $\beta(x)\sqrt{1 + (\partial b/\partial x)^2}$.

- line 405 and elsewhere throughout the paper. Since you're focused on finite elements, I really think that "mesh" rather than "grid" would be the right choice here, since "grid" is usually taken to imply a greater degree of regularity than a mesh. Plus, "mesh" would seem to be standard parlance in finite elements. (Your

response to reviewers suggested at one point that you had replaced "grid", but that certainly doesn't seem to have been done consistently)

- line 425 "We follow this practice except that here the discretization originates from a variational principle. This has a number of advantages (see §2.3 and DPL, 2010)." — I expect I complained about this in the original submission, but in any case, this is likely a mischaracterization. Any finite element discretization that is based on the standard weak form of the Stokes equations will have the properties listed in section 2.3: boundary equations formulated in terms of surface integrals (rather than inaccurate one-sided expressions), and a symmetric stiffness matrix. That remains true regardless of whether I am alert enough to recognize (or bother to say out loud, for that matter) that the weak form is the Frechet derivative of the functional being minimized or not. If my discretization does something different, then I am likely not using standard finite elements. I don't see that there is much of a grey zone on this — if you know of an example of a paper using finite elements in a way that does not lead to a discretization with the stated properties, be sure to explicitly identify that paper, and where in that paper I might be able to find evidence of a non-standard finite element formulation (I'd be interested!). Otherwise omit this passage, unless you wish to reiterate the statement about the "extended Blatter-Pattyn model" solving for w and the dynamically rather redundant reduced pressure $\tilde{P}$.

- line 455 "in matrix form": misleading as $\mathcal{M}$ hasn't been cast in matrix form

- line 459 "$\mathcal{M}(u, w)$ is a nonlinear positive-definite function of the velocity" — this is an odd thing to say as I don't think I can have a nontrivial *linear* positive-definite function, unless it's defined on some bounded set of values $(u, w)$. To wit, if $\mathcal{M}(u, w) > 0$ and $\mathcal{M}$ was linear, then it would follow that $\mathcal{M}(-u, -w) < 0$ by linearity, no? Do you mean "convex"?

- line 466 "Discrete variation of the functional corresponds to partial differentiation with respect to each of the discrete variables in $V$." Partial differentiation of ... the discretized functional? It's unclear who the audience is, but probably best to be specific.

- equation (48) It's probably a bad idea to use the same letter $M$ for the dissipation potential part of the discretized functional $\mathcal{A}$, and for the Hermitian matrix of the discretized functional, at the same time, even if one $M$ is in calligraphic and the other is not. Also note that the Cryosphere has some rather inflexible (and unimaginative) rules around renderinng matrices and vectors in bold face, in in upright and the other in italicized font. Be prepared fo a bunch of fun at the copyediting stage.

- line 519 onewards "The form (55) is characteristic of Stokes-type problems, or more generally of constrained minimization problems using Lagrange multipliers. In finite element terminology these are called "mixed" or "saddle point" problems, meaning that velocity components and the pressure occupy different finite element spaces, and that the solution of (55) is actually at the saddle point with respect to the velocity and pressure variables of the quadratic form associated with (55). The matrix $M$ is symmetric but indefinite, with both positive and negative eigenvalues. As a result, the matrix inverse may not be bounded and may lack stability" There are a number of things that are problematic here. The first is the statement that the generic form of defines a saddle point problem. That is not true unless you add that $A$ has to be at least positive semi-definite. The next is the statement that "these are called mixed or saddle point problems, meaning that velocity components and the pressure occupy different finite element spaces." That is not the meaning of a saddle point problem. A saddle point problem corresponds to a problem of minimizing with respect to one variable (velocity) and maximizing with respect to another (pressure). You also can't say that the matrix inverse may not be bounded (if the inverse of a matrix exists, it is bounded!) — the point you're presumably getting at without spelling it out is this: implicit in the discretization is that you are looking at a family of discretizations of progressively of different resolutions resolution (parameterized by maximum element size, call it $h$, combined with a non-zero lower bound on internal element angles), and you want the corresponding family of saddle points to have a unique limit as $h \to 0$. The statement obout the bound on the inverse of $M$ (and this is what the inf-sup condition guarantees, in my understanding) is that $M^{-1}$ *remains bounded as $h \to 0$*. No?

- line 530 "In this case only the matrix $A$ exists, it is elliptic" — two things: i) you seem to be saying that $A$ only exists in this case and ii) "elliptic" is a term you haven't defined up to now, I think. Symmetric positive-definite?

- also line 530 "As a result, the standard Blatter-Pattyn model is a well-behaved and stable unconstrained minimization problem" — this is an unfortunate choice of words, since you've just told the reader that $B = B^T = 0$, in which case $M$ is manifestly not invertible; it seems like you have to state that you're not actually solving (55) but only the reduced version $A\mathbf{u} = \mathbf{f}$, for which the statement in question is true.

- line 544 "In fact, this is a problem for all inf-sup stable elements with $n_p \neq n_w$ , such as the Taylor-Hood element, for example" two things i) you have not defined what you mean by "inf-sup stable elements" and ii) the second half of the sentence about Taylor Hood elements is redundent. You could say "is a problem for all inf-sup stable elements with $n_p \neq n_w$". Comment on whether $n_p \neq n_w$ is a prerequisite for an element pair to be inf-sup stable (which I think is true); the last sentence grammatically leaves open the possibility that you could have $n_p = n_w$.

If that is the case, comment on the implications for your choice of basis functions, which do have $n_p = n_w$. Are they necessarily not inf-sup stable? This requires an explicit comment, not a response that says "I don't want to address this issue". See discussion re unconstrained minimization being a red herring, unless you can show that your divergence-free basis functions are dense in the underlying function space.

In terms of sequencing the three items (1) Blatter-Pattyn (2) extended Blatter-Pattyn and (3) Stokes / transformed Stokes, it would actually be preferable to reverse the current order since that would mean introducing inf-sup conditions before you talk about them for extended Blatter-Pattyn

- Line 564 "As mentioned previously, this is possible to do in the continuum but not necessarily so in the discrete case" — I don't think you've explained that this is "not necessarily possible" to do in the discrete case, unless you mean the discussion around equation (56) that is about to follow. To avoid confusion and leave the reader scurrying for where this was explained previously, especially as you've just referenced section 3.4.1, and there is no mention of not being able to compute w there, and even the discussion starting on line 396 also doesn't actually say that you cannot compute w in the discrete case. In fact, I expect that those running Blatter-Pattyn ice sheet models would beg to differ with the statement that you cannot compute w from the continuity equation, since they will naturally choose a scheme that finds w by simple quadrature. This seems more of an issue with the *extended* Blatter-Pattyn model of section 3.4.2, where you are forcing yourself to use the same mesh as for $(u, v)$, no?

- line 552 "Both the standard and transformed Stokes models are subject to this problem an in general must use inf-sup stable finite elements. Testing for stability is not trivial. However, collections of inf-sup stable elements for the Stokes equations may be found in many papers and books on mixed methods, e.g., Boffi et al. (2008)." and later line 594 "However, this model does work with a variant of the Taylor-Hood grid, the P2-E1 grid, illustrated in Fig. 13A, which does satisfy the solvability condition and this therefore allows for a successful calculation of the vertical velocity." — I think I raised this point in my original review, but I still don't see it being addressed here: *are* the new finite element basis functions like P2-E1 and P1-E0 inf-sup stable, or are you arguing (see comments immediately below) that this is somehow no longer relevant when you the solvability condiiton for the incompressibility condition is satisfied? Otherwise the discussion of the inf sup condition is still left dangling in thin air here. If you're referring to the many papers and books *because you won't be addressing the issue here*, at least say so explicitly. See also comment on line 544.

- line 575 "$n_p + \lambda_z + \Lambda = n_w$" should presumably be "$n_p + n_{\lambda_z} + n_\Lambda = n_w$"?

- line 599 "Perhaps the main reason for the importance of the solvability condition is that it implies that the Stokes variational principle, (15) or (33), may be transformed into and therefore that it is equivalent to an optimization or minimization problem." — this is at best misleading, but most likely just wrong. Finding stationary points of (15) and (33), and in fact the solution of all Stokes flow problems subject to suitable boundary conditions, is *always* equivalent to a convex minimzation problem, but that minimization is over an awkward vector space of Sobolev space of divergence free functions. I should: awkward in terms of finding suitable basis functions when discretizing in practice, but not particularly awkward in terms of the abstract analysis of the problem.

  For the present set of boundary conditions and choice of rheology, the relevant function space is $V = \{v \in [W^{1+1/n}(\Omega)]^3 : \mathrm{div}\, v = 0 \text{ and } v = 0 \text{ on } \partial\Omega_{b_1}, v \cdot n = 0 \text{ on } \partial\Omega_{b_2}\}$, and the cited Chen et al (2013) paper would be an appropriate reference for this fact. For the transformed Stokes flow problem, the transformed minimization would also seem to be pretty straightforward, since, if I restrict myself to the same Sobolev space as Chen et al, I obtain a restriction of A to velocities only in the form

  $$J(u, v, w) = \int_V \frac{4n\eta_0}{n+1} \dot{\epsilon}^{1+1/n} + \rho g \nabla z_s \cdot (u, v, 0) \mathrm{d}V + \int_{S_{b_2}} \beta |(u, v, w,)|^2 \mathrm{d}\Gamma$$

  and $J$ is convex, so any stationary point of $J$ is automatically a minimizer. If you are actually trying to say that you're trying to determine whether the *discretized* variational problem still corresponds to a minimization problem, that is a different matter, but then you can't refer to (15) or (33), which are formulated for general velocity fields.

- line 611 "This result suggests that a conventional Stokes problem, when solved on a grid satisfying the solvability condition, is equivalent to an unconstrained minimization problem and therefore is well behaved." As per the above, this risks sounding like you are unfamiliar with standard results in the field, and I would not put this statement into a paper.

  Even if you don't mind giving that appearance, it would still be a bad idea to leave the statement as is, because it could be read to suggest that basis functions that don't satisfy (57) don't correspond to a minimization problem. That would be wrong.

  The conventional Stokes flow problem remains equivalent to an unconstrainted minimzation problem even after discretization. That unconstrained minimization is however over the left nullspace of the matrix $B$ in (55) — that is, over all vectors $\mathbf{u}$ such that $B^T \mathbf{u} = 0$. In general, you don't want to have to figure out what that nullspace is: as you later point out, the basis vectors for that nullspace are no longer sparse when expressed in terms of nodal values of the mesh, so your linear algebra ends up non-sparse and therefore intractable for large problems. Instead, you continue to impose the need for your solution to lie in that nullspace through

the Lagrange multipier $p$, but that doesn't negate the fact that the problem is equivalent to an unconstrained minimization problem..

For the discretized problem on a given, fixed mesh, the only condition for this statement to be true is that the left nullspace of $B$ must be non-trivial, and for that to be true, you generally require the matrix to be taller than wide ($n_u + n_v + n_w > n_p$). You definitely don't need $n_w = n_p$, though $n_w = n_p$ will do. (The solvability condition (57) conveniently ensures that the nullspace in question has dimension $n_u + n_v$ and can be written in the form given on line 620, but that is far from the only form of nullspace you could construct. In general, if $n_p < n_w$ for an inf-sup stable basis you have to use more than a basis for $(u, v)$ to construct a divergence-free basis for $(u, v, w)$, but that is not in principle a problem)

- line 617 "if a divergence free basis exists" is a misleading way to make this statement. For the original continuum problem, the divergence free basis exists without a doubt. (Again, Chen et al 2013 are the correct reference for the particular problem you have in mind). My suspicion is that the book you are referencing has in mind a divergence free basis that is dense in the underlying function space, in the sense that it can approximate any element of $V = \{v \in [W^{1+1/n}(\Omega)]^3 : \text{div} v = 0 \text{ and } v = 0 \text{ on } \partial\Omega_{b_1},\ v \cdot n = 0 \text{ on } \partial\Omega_{b_2}\}$ to arbitrary accuracy simply by imposing a suitably small maximum edge length.

  That is not something that you have demonstrated. You seem to be alluding to the equivalence to an unconstrained minimization problem as evidence that the problem you're solving "is well behaved", without saying what you mean by that, precisely. Based on the text in your response to reviewers (though not the text here!) I assume you arguing that you don't need to worry whether your basis functions actually satisfy the inf-sup condition because you can show that your discretized problem is equivalent to an unconstrained finite-dimensional minimzation problem.

  If so, that is not: all that the equivalence with an unconstrained minimization problem that you've discussed (which applies to the discretized problem) ensures case is that the discretized problem is solvable, for any given mesh. It does *not* guarantee that the solutions for a family of discretized problems will converge to the solution of the continuum problem in the limit of element size $h$ in the mesh going to zero.

  More formally, let the function space spanned by your basis function be $V_h$ for a given mesh with maximum element edge length $h$. The thing you need to prove is that your diverge free finite element basis functions can adequately approximate *any* possible solution in the limit of small $h$, that is

  $$\inf_{v_h \in V_h} ||v - v_h|| \leq C_h ||v||$$

  where $C_h \to 0$ as $h \to 0$. In the absence of the divergence constraint $B^T \mathbf{u} = 0$, that behaviour is well established for all sorts of polynomial basis functions, but

is no longer guaranteed to hold if you restrict yourself to a linear subspace of of such polynomial basis functions. How you might prove the same behaviour for your divergence free subspace is beyond my pay grade, but you can't just skip past this issue. That would make a mockery of a lot of work by some very smart people in numerical analysis.

- line 633 "These tests are described in Appendix A" — I checked the appendix as well as Pattyn et al (2008), and it's unclear to me how the "true" solution relative to which the error is computed in figure 3. And in the same vein, if there is a true solution, you should plot that in figure 4.

- line 702 "It would be computationally cheaper to use the standard Blatter-Pattyn approximation $(\xi, \hat{\xi} = 0$ ) instead, solving only for the horizontal variables and coupling to the Stokes model with $p = 0$ and $w = w(u, v)$ at the interface but this, however, implies much more complicated programming." — see main comment re response to reviewers. That response argues (correctly, in my view) that Blatter-Pattyn is cheaper than a Stokes flow model. Which is understandable given that it has fewer degrees of freedom to solve for. The extended Blatter-Pattyn model on the other hand is constructed to have exactly the same number of degrees of freedom as the Stokes flow model, and it is *not* obvious that it will be much cheaper to solve than the Stokes flow model (transformed or otherwise). Please provide evidence (ideally here!) that it remains sufficiently cheaper to solve than the Stokes flow model to warrant doing this. As per my main comment, using the extended Blatter-Pattyn model will clearly incur a potentially significant model error relative to a Stokes flow solver, so it is important to know whether the reduction in computational cost is worth the increased error.

- line 725 "Somewhat counterintuitively, the Stokes region occupies the upper part of the domain in Fig. 7 and includes the obstacle, while the Blatter-Pattyn region occupies much of the bottom part of the domain" — I am not sure this is counterintiuitive in view of the no-penetration boundary condition combined with the moderate angle: basically, your boundary condition ensures that dw/dx is small along the bed, while there is no reason why du/dz would be. I am however not convinced that this test goes far enough: really, you should also compute the Stokes solution across the domain to see

- line 757 "The first method, to be called the BP+ approximation" and later line 773 "An Improved Blatter-Pattyn or BP+ Approximation" line 812 "Remarkably, this same model, i.e., the BP+ approximation, was introduced by Herterich (1987)" line 1029 "Remarkably, the BP+ approximation is actually the same as a model originally proposed by Herterich (1987)." etc ... Given that Herterich predates either Blatter or Pattyn, and given that "BP+" suggests that this is somehow a development of Blatter and Pattyn, this really needs to be "Herterich's improved

Blatter-Pattyn Approximation" or better just the "Herterich model". Definitely not "BP+", thereby giving further credit to Blatter and Pattyn while further consigning the person who actually invented this to a footnote? Let's have some respect.

- line 839 "Both BP+ versions converge to the same solution" — this confuses "model" (a set of equations" with "solution algorithm". These are not two different "BP+ versions", they are different solution algorithms

- line 868 "logically rectangular" — what does that mean? Producing the same graph as a rectangular grid? Define before using . . .

- line 1007 "On the other hand, the pressure in the P2-E1 case is highly oscillatory but well behaved in the P2-P1 case. However, this is not at all concerning since the transformed pressure, a Lagrange multiplier, has no physical significance" — so I think you've just demonstrated that your P2-E1 element most likely does not satisfy the inf-sup condition, and you're effectively left hoping that the lack of convergence in the limit of small element size only affects the pressure, but not the velocity. Comment?

- line 1011 Summary and Discussion — reading this, all is fine and dandy? Nothing that you would flag as an open question or necessary areas of future research where things need to be developed further, followed up on, etc? Any weaknesses? No?

- Appendix B This is a follow up on a comment from my first review, the response to which was that you thought *the P1-E0 element is quite flexible and can be used with quite general triangulations.* The trianglulations you require are *not* "quite general". By construction, your mesh cannot be unstructured as you require elements to be stackable to make columns. You should comment on that as it negates one of the usual advantages of finite elements, which is to permit adaptive meshing. For instance, at grounding lines, ice stream shear margins etc, you may want to have high resolution *near the bed* around a transition in sliding behaviour, but not extend that resolution throughout a column. Moreover, if you are required to keep the interior angles of elements from becoming excessively small, your mesh also makes it somewhat difficult to have variable vertical triangle edge lengths — adjacent columns need to have similar triangle numbers and triangle edge lengths. This needs to be flagged somewhere.

---

## Author Response (AR2)

**Response #2 to Reviewer RC2 – Prof. Christian Schoof**

**Egusphere-2024-1052 Article**

**A PDF of the revised manuscript is attached**

Author Revisions and Responses are highlighted in Blue

I again thank Prof. Schoof for a detailed and insightful review!

**Higher level comments**

This paper has improved in readability from the original submission. There do remain a number of significant issues to be addressed. I have to say that this is the first time I've found myself reviewing the "response to reviewers" as much as the revised paper, so don't take the points I make lightly.

Before I get started, let me say this: decisions on manuscripts are obviously up to the editor, but there are two issues without which I won't be endorsing publication of this paper.

The first is adequate citation of prior literature. My general comment about the manuscript being ungenerous to the prior literature resulted in the following line in the response to reviewersI am happy to add additional or more appropriate references. Any sugggestions?

but several pages later, when I actually list the relevant references, you say

I'm reluctant to add a lot of extra references

Most scientists find out sooner or later that it's easier in the review process to follow the path of least resistance when being asked to cite additional papers. If those are the reviewer's papers, you have a point in questioning their motives, but if they are third party papers and you try to avoid referencing them, you risk appearing as though you'd rather not acknowledge them. In the present manuscript, that unwillingness seems to focus on relevant literature on variational formulations that predates the early 2010s. That is, coincidentally, when Dukowicz, Price and Lipscomb ("DPL 2010") was first published, which you are using as the standard reference for variational formulations. The relevant citations are (and this is simply the most important subset!) J. Colinge and J. Rappaz, A strongly nonlinear problem arising in glaciology, Math. Model. Numer. Anal. 33 (1999) 395–406.

R. Glowinski and J. Rappaz, Approximation of a nonlinear elliptic problem arising in a non-Newtonian fluid flow model in glaciology, Math. Model. Numer. Anal. 37 (2003) 175–186.J. Rappaz and A. Reist, Mathematical and numerical analysis of a three-dimensional fluid flow model in glaciology, Math. Model. Mech. Appl. Sci. 15 (2005) 37–52.

The obvious place for these is line 334 just before eq (34). This is first and foremost a matter of respect. If that is too many papers, leave out Glowinski and Rappaz.

I have added the Glowinski and Rappaz, and the Rappaz and Reist references.

The second is the naming of the "BP+ model". As you point out, Herterich's (1987) model differs from Blatter-Pattyn. ("Needless to say, I will not be changing the name of the Blatter- Pattyn model.") But the Herterich model is what you now want to call "BP+", even though Herterich developed it nearly forty years ago, the best part of a decade before Blatter. It was *not* developed from the BP model, which is precisely what calling it the "BP+" model would suggest to future generations. The person who actually came up with it deserves a little more respect.

May 17, 2025

You are right. I have changed BP+ to Herterich throughout.

I have two main scientific points, some of which are repeated in greater detail in the line-by-line comments (which I wrote prior to this summary)

1. The inf-sup condition: Your response to reviewers states
   *I have eliminated Appendix D and rewritten Section 4 to bypass the difficult issue of the inf-sup condition. The inf-sup condition is relevant only so as to point out that the standard and transformed Stokes models are subject to it and that one must use one of the many inf-sup-stable elements available in the literature in the dis- cretization.*
   *I hope I have made it clear that the inf-sup condition does not apply to problems using elements satisfying the "solvability condition" because they are no longer con- strained problems since incompressibility is built-in when using $w = (u, v)$*
   This is deeply problematic as a rationale for not addressing the inf-sup condiiton. For one thing, you *do not* state ambiguously that you regard the ability to formulae the problem as an unconstrained minimzation problem as a reason for not address-ing whether your novel finite elements satisfy the inf-sup condition. But worse is that the argument given above is incorrect.
   First of all, Stokes flow problems of the type discussed here are *always* equivalent to an unconstrained minimzation problems. See Chen et al 2013 for the continuum case. This reamins true for the discretized version so long as the discrete "diver-gence" operator given by $B^T$ in equiation (55) has a non-trivial right nullspace, meaning there are vectors $\mathbf{u} \neq \mathbf{0}$ such that $B^T \mathbf{u} = \mathbf{0}$. Being able to find such vec-tors does not require the solvability condition (57) to hold, or even the usual inf-sup condition to hold (more on this shortly), but the existence of such a nullspace is assured for any set of basis functions satisfying the inf-sup condition. When the nullspace exists, the unconstrained minimization problem is over that nullspace. I have written out additional detail on this in thespecific comments on lines 611 and 617.).
   Second, there is no reason why the equivalence with an unconstrained minimiza-tion problem should get around having to address whether your new basis functions satisfy the inf-sup condition. First, let me state a subtely to the inf-sup condition, which I touched on in my previous review: I can impose the inf-sup condition at a particular discretization level, sot hat the inf-sup bound (usually "$\beta$) exists for that disretization level, and hence my discretized problem is equivalent to a finite-dimensional unconstrained minimization, but that doesn't mean that I will get correct solution. For the partial differential equations, I have to make sure that the inf-sup (usually "$\beta$) does not depend on the discretization level, usually expressed in terms of the maximum element size $h$. This ensures that as you take finer and finer meshes, the discretized solution converges to a solution of the continuum problem, while you have no such assurance if $\beta = \beta_h$ depends on $h$ and $\beta_h \rightarrow 0$ as $h \rightarrow 0$ (so long as your unconstrained finite element basis can adequately approximate all functions of interest in the limit $h \rightarrow 0$, see below). If you instead treat your discretization as defining a divergence free basis spanning a finite-dimensional subspace (the right nullspace of $B^T$ ) over which you do un-constrained optimization, then nature of the problem remains the same: you have to show that solutions converge to the continuum solution as $h \rightarrow 0$. It is only

May 17, 2025

the way that you cast the problem that changes: instead of showing your basis functions satisfy the inf-sup condition with a $\beta$ indepndent of $h$, you now need to determine whether you new, divergence-free basis functions (which are a subset of standard piecewise linear or quadratic bases) can still approximate all functions in the admissible space

$$V = \{v \in [W^{1+1/n}(\Omega)]^3 : \mathrm{div}\, v = 0 \text{ and } v = 0 \text{ on } \partial\Omega_b, \, v \cdot n = 0 \text{ on } \partial\Omega_b \},$$

arbitrarily well in the limit $h \to 0$. That is *not* trivial; if it was, it's unclear anyone would ever have bothered with the inf-sup condition because it's much easier to find finite element functions for which there is an $h$-dependent $\beta_h$ for which you have equivalence to a finite-dimensional unconstrained minimizatoin problem but no guarantee of convergence as $h \to 0$, than it is to find basis functions for which $\beta$ is independent of mesh size. See the line-by-line comment for line 617.

This brings me back to: you talk about the inf-sup condition, but you still do not explicitly address the question of whether your novel finite element basis functions satisfy the inf-sup condition, and if they don't, exactly why that should not matter. See above re the unconstrained minization idea being a red herring. There are furhter specific notes related to this in the comments on line 544–617. I also note that your off-hand comment on line 1006 suggests that at least some of the finite element bases that you are constructing don't satisfy the inf-sup condition, so this isn't a flippant point to make, and you should at least point out there that the P2-E1 space is therefore likely not to satisfy the inf-sup condition, but your numerics give you hope that perhaps the solution in general does converge for the velocity field, if not the pressure field. (By "in general", I mean )

This is the main issue to be addressed. Many of your comments here and below pertain to the inf-sup condition and the solvability condition introduced here. It's a highly technical discussion; to use your expression, it's really "beyond my pay grade". Fortunately, however, I believe I've made progress as explained in the new Appendix B and summarized below.

Benzi et al. (2005), Gerbeau (2025) discuss an alternative method (the dual problem) for the solution of the Stokes saddle point problem. An equation for the pressure is solved first, followed by the solution for velocity. The equation for the pressure (the Schur complement with respect to the pressure $p$) is solvable provided the inf-sup condition is satisfied. The standard inf-sup condition is shown to be equivalent to $Ker\mathbf{B}^T = \{0\}$, i.e., the absence of a null space. This identifies the source of the difficulty leading to the inf-sup condition as the existence of a null space (i.e., spurious pressure modes in the $B^T$ matrix in equation (54). I changed $B$ to $B^T$ in this equation to conform to standard practice in the literature.)

On the other hand, if the solvability condition is satisfied then $w(u)$ may be used to convert the Stokes model into an unconstrained minimization problem for the horizontal velocity $u$. Having obtained $u$, an equation for the pressure is obtained that involves an invertible matrix $\mathbf{M}_{WP}$, which means that the pressure problem in now well posed (no spurious modes). Thus, the entire Stokes problem in now well posed and there is no need to satisfy an inf-sup condition. Note that both the Schur complement equation for the pressure and the unconstrained minimization problem for the velocity involve solving dense matrix problems whose solution is possible but impractical in practice.

The only problem is that the material in Gerbeau (2025) is only available online as lecture notes for a course at Stanford University. Some of the material is available in Benzi et al. (2005) but not all. I reached out to Prof. Gerbeau but he said that the material was prepared so long ago that he did not remember the original source. If you know a better source for this material I would be very grateful.

The Herterich ("BP+") model. I flagged in y original review that this is an *ad hoc* model for which there is no theoretical justification in terms of asymptotic error estimates; it just turns out to work well for the test cases you have run. Your response to reviewers says *I doubt that it's necessary to have a full scale analysis when introducing a new approximation. For example, the Blatter- Pattyn model did not have a scale analysis for 15 years until Schoof and Hindmarsh (2010).*
I read that to say it is unreasonable and unnecessary to expect a scaling analysis, and that this would somehow be difficult, and that you do not wish to discuss the issue. Let me do the scaling analysis for you, in that case; you will need to discuss this when introducing the model 6.2.1 (where the section heading claims Herterich to be an improved Blatter-Pattyn model).
The Herterich model as you state it in equations (67)–(68) can be derived from the following form of the Stokes model, with terms selectively removed:

$$\frac{\partial \tau_{xz}}{\partial z} + \frac{\partial \tau_{xx}}{\partial x} - \frac{\partial p}{\partial x} = 0$$
$$\frac{\partial \tau_{zz}}{\partial z} - \frac{\partial p}{\partial z} = -\rho g$$
$$\frac{\partial u}{\partial x} + \frac{\partial w}{\partial z} = 0$$
$$\tau_{xz} = \eta \left( \frac{\partial u}{\partial z} + \frac{\partial w}{\partial x} \right) \quad \tau_{xx} = 2\eta \frac{\partial u}{\partial x}$$
$$\tau_{zz} = 2\eta \frac{\partial w}{\partial z}$$

subject to

$$\tau_{xz} - \frac{\partial z_s}{\partial x}(\tau_{xx} - p) = 0$$
$$\tau_{zz} - p = 0$$

at $z = z_s$, and appropriate basal boundary conditions that I won't write out in detail.
Assume for simplicity that eta is constant; the shear-thinning power law fluid case of Glen's law is only superficially different. If I take the full Stokes model instead and apply a standard "shallow ice" scaling, I obtain (see e.g. section 3.6 of Schoof and

Hindmarsh (2010), although the original references would be Fowler and Larson (1978), Morland and Johnson (1980) and Hutter (1983))

$$\frac{\partial \tau_{xz}^*}{\partial z^*} + \epsilon^2 \frac{\partial \tau_{xx}^*}{\partial x^*} - \frac{\partial p^*}{\partial x^*} = 0$$

$$\epsilon^2 \frac{\partial \tau_{xz}^*}{\partial z^*} + \epsilon^2 \frac{\partial \tau_{zz}^*}{\partial z^*} - \frac{\partial p^*}{z^*} = -1$$

$$\frac{\partial u^*}{\partial x^*} + \frac{\partial w^*}{\partial z^*} = 0$$

$$\tau_{xz}^* = \frac{\partial u^*}{\partial z^*} + \epsilon^2 \frac{\partial w^*}{\partial x^*}$$

$$\tau_{xx}^* = 2\frac{\partial u^*}{\partial x^*}$$

$$\tau_{zz}^* = 2\frac{\partial w^*}{\partial z^*}$$

subject to

$$\tau_{xz}^* - \frac{\partial z_s^*}{\partial x^*}(\epsilon^2 \tau_{xx}^* - p^*) = 0$$

$$\epsilon^2 \tau_{zz}^* - p^* - \epsilon^2 \frac{\partial z_s^*}{\partial x^*}\tau_{xz}^* = 0$$

where the asterisks denote scaled variables, and red colour marks the terms that are not retained in the Herterich model. It should be apparent that the Herterich model retains *some* of the $O(\epsilon)$ (the $\partial w^*/\partial x^*$ term in the definition of $\tau_{xz}^*$ and the full slope term in the shear stress boundary condition. It retains by no means all $O(\epsilon)$ terms, so there is no reason why the Herterich model should be any better in this parametric limit (with respect to sliding) than the BP model, which also selectiely omits $O(\epsilon)$ terms and therefore has an $O(\epsilon)$ error.

We can conversely turn to the limit of fast sliding. Using the appropriate scaling for that case (e.g. Schoof and Hindmarsh (2010) section 3.4, although the original source for this goes back to MacAyeal (1989) and beyond)

May 17, 2025

$$\frac{\partial \tau_{xz}^*}{\partial z^*} + \frac{\partial \tau_{xx}^{**}}{\partial x^*} - \frac{\partial p^*}{\partial x^*} = 0$$

$$\epsilon^2 \frac{\partial \tau_{xz}^*}{\partial z^*} + \frac{\partial \tau_{zz}^{**}}{\partial z^*} - \frac{\partial p^*}{\partial z^*} = -1$$

$$\frac{\partial u^{**}}{\partial x^*} + \frac{\partial w^{**}}{\partial z^*} = 0$$

$$\epsilon^2 \tau_{xz}^* = \frac{\partial u^{**}}{\partial z^*} + \epsilon^2 \frac{\partial w^* *}{\partial x^*}$$

$$\tau_{xx}^{**} = 2 \frac{\partial u^{**}}{\partial x^*}$$

$$\tau_{zz}^{**} = 2 \frac{\partial w^{**}}{\partial z^*}$$

subject to

$$\tau_{xz}^* - \frac{\partial z_s^*}{\partial x^*}(\tau_{xx}^{**} - p^*) = 0$$

$$\tau_{zz}^{**} - p^* - \epsilon^2 \frac{\partial z_s^*}{\partial x^*}\tau_{xz}^* = 0$$

where the variables with double asterisks denotes dimensionless variables that have been rescaled from the shallow ice case. Once more, the Herterich model retains some of the $O(\epsilon)$ terms, but not all, and will again only be accurate to an $O(\epsilon)$ error, the same as the Blatter-Pattyn model.

We can also look at the intermediate sliding regime (Schoof and Hindmarsh (2010), section 2.1, formally with lambda = 1), in which case

$$\frac{\partial \tau_{xz}^*}{\partial z^*} + \epsilon \frac{\partial \tau_{xx}^{***}}{\partial x^*} - \frac{\partial p^*}{\partial x^*} = 0$$

$$\epsilon^2 \frac{\partial \tau_{xz}^*}{\partial z^*} + \epsilon \frac{\partial \tau_{zz}^{***}}{\partial z^*} - \frac{\partial p^*}{\partial z^*} = -1$$

$$\frac{\partial u^{***}}{\partial x^*} + \frac{\partial w^{***}}{\partial z^*} = 0$$

$$\epsilon \tau_{xz}^* = \frac{\partial u^{***}}{\partial z^*} + \epsilon^2 \frac{\partial w^{***}}{\partial x^*}$$

$$\tau_{xx}^{***} = 2 \frac{\partial u^{***}}{\partial x^*}$$

$$\tau_{zz}^{***} = 2 \frac{\partial w^{***}}{\partial z^*}$$

on $\Omega$, subject to

$$\tau_{xz}^* - \frac{\partial z_s^*}{\partial x^*}(\epsilon \tau_{xx}^{***} - p^*) = 0$$

$$\epsilon^* \tau_{zz}^{***} - p^* - \epsilon^2 \frac{\partial z_s^*}{dx^*}\tau_{xz}^* = 0$$

Same story. The Herterich model selectively retains $O(\epsilon)$ terms and offers no improvement in terms of asymptotic error over the Blatter-Pattyn model. That should settle the case — if you want a single model with a better asymptotic error (that is, size of error in the limit of $\epsilon \to 0$ than Blatter-Pattyn, you really have to retain all $O(\epsilon)$ terms. In order to do that regardless of sliding regime, it should be clear you actually have to retain all terms of the full Stokes model (since the smallest term contained in all the scaled versions of the Stokes flow model above is $O(\epsilon)$, there is really nothing that you can omit if you want a model accurate to $O(\epsilon)$).

The fact that the Herterich model performs better than BP in specific tests you have computed is really not much of a guide to anything in general: if you had a general result predicting a smaller error, then something like figure 9 would be a welcome illustration. In the absence of a much more general investigation of model error, the smaller error here might as well be fortuitious and specific to some aspect of test B. (I would also note that figure 5 shows resutls for a fairly large epsilon, so we are not looking an approximation based on small aspect ratio: an aspect ratio of 1:5 really cannot be treated as small in general, as finite depth effects typically appear for aspect ratios around $1 : (2\pi)$)

My sense of why the Herterich model is possibly an improvement over Blatter-Pattyn comes from the fact that the Herterich model restores a term dropped from the Stokes functional by an approximation of that term. This sense is probably invalid because it's the Euler-Lagrange equations derived from the functional and not the functional itself that carry the important information. I agree that a scale analysis of the equation could be helpful. However, I still feel that it would be well beyond the scope of the paper as is. What I have done instead is expand the comparison of the BP and Herterich approximations in Fig. 9 by adding results for $L = 2\,km$, i.e., an even higher problem aspect ratio, thereby stressing small aspect ratio approximations further. The results are surprising. While the Herterich approximation is clearly an improvement in Test B at $L = 5, 10\,km$, Blatter-Pattyn is better at $L = 2\,km$ in both Tests B and D*. As pointed out in the paper, these results make the Herterich approximation somewhat problematic and so more experience would be helpful.

**Line-by-line comments**

- eq (11): there is still no need for the brackets around the (i) subscripts. In fact, by putting them into equation, you actually need to add the statement that $\tau_i^S n_i = 0$ since (11) is mute about the value of its z-component. If you write instead that

$$\tau_i^S = \tau_{ij} n_j - \tau_n n_i$$

you automatically ensure that $\tau_i^S n_i = \tau_{ij} n_i n_j - \tau_n n_i n_i = \tau_n - \tau_n = 0.$ '

May 17, 2025

That also tidies up your notation.

Section involving equations (10)-(13) has been rewritten.

- line 188 "references contained therein in connection with the Blatter-Pattyn model" — No, not enough. See start of this re-review

References added, see above.

- line 203: "However, this can only be done in the discrete formulation of the functional since only then are boundary values of velocity accessible (although they are always accessible in the surface integral terms)." This either not true or misleading, depending on what you mean by "accessible". Infinite-dimensional variational formulations of pde problems impose homogenous Dirichlet conditions by restricting the function space on which the problem is posed, rather than using Lagrange mutlipliers. In the present case, no need to get technical about that function space, but simply delete this statement and state that the minimization problem needs to be restricted first arguments of the functional A that satisfy the Dirichlet conditions

Lines 197-210 have been rewritten. Hope this clarifies the situation.

- eq (15) The (b2) superscript on the second line is undefined and doesn't want to be there. Also, this is needlessly complicated: the form of the surface integral is exactly the same as

$$\int_{S_{B_2}} \frac{1}{2} \beta u_i u_i \mathrm{d}S,$$

but more complicated. And it's not like you've replaced $u_3$ in the rest of the formula yet, so why make it more complicated in the boundary term. (This is different from (33), where the substitution makes more sense.)

Subscript corrected. I believe that the need for the more complicated form has been addressed. It's needed because the surface integral is over the section where the frictional boundary condition applies and this boundary condition involves the replacement $w \rightarrow -u_{(i)} n_{(i)} / n_z$ as explained in the text.

- eq (17) seems redundant. If you cannot see follows from (16) and (18), you're going to struggle mightily with the rest of this paper.

Equation (17) has been removed.

- eq (26) Again, write in standard subscript form?

May 17, 2025

Equations (26), (27) replaced by a single equation for the stress tensor.

- line 287 "remains positive" → "remains non-negative"

Corrected.

- line 289 "The dummy variables..." — I don't think this is the usual meaning of "dummy variables" (like the variable with respect to which you integrate in a definite integral). They are indicator terms, taking values of 1 or 0. Same in line 344.

I think "dummy variables" is appropriate here.

- eq (29) — this is a bit misleading; your earlier text (line 289, "The dummy variables $\xi, \hat{\xi}$ in (23)-(25) and (26)-(29) are used to identify terms that are neglected in the two types of the Blatter-Pattyn approximation discussed in §3.4") indicates that all I need to do in order to obtain Blatter-Pattyn is to set $\xi = \hat{\xi} = 0$.

The paragraph regarding dummy variables has been shifted to after equation (31).

- Remark #1 — this isn't about "computational savings", is it? The small slope approximation is what makes Blatter-Pattyn valid in the first place, and the error in replacing the normal by its small slope approximation is the same order as the error in the Blatter-Pattyn model in the first place. Besides — and you might want to spell this out clearly — the difference between the two amounts to replacing

  $\beta(x)$ by $\beta(x)1 + (\partial b/\partial x)^2$.

Remark #1 has been eliminated.

- line 405 and elsewhere throughout the paper. Since you're focused on finite elements, I really think that "mesh" rather than "grid" would be the right choice here, since "grid" is usually taken to imply a greater degree of regularity than a mesh. Plus, "mesh" would seem to be standard parlance in finite elements. (Your response to reviewers suggested at one point that you had replaced "grid", but that certainly doesn't seem to have been done consistently)

"grid" has been replaced by "mesh" throughout.

- line 425 "We follow this practice except that here the discretization originates from a variational principle. This has a number of advantages (see §2.3 and DPL, 2010)." — I expect I complained about this in the original submission, but in any case, this is likely a mischaracterization. Any finite element discretization that is based on the standard weak form of the Stokes equations will have the properties listed in section 2.3: boundary equations formulated in terms of surface integrals (rather than inaccurate one-sided expressions), and a symmetric stiffness matrix. That remains true regardless of whether I am alert enough to recognize (or bother to say out loud, for that matter) that the weak form is the Frechet derivative of the functional being minimized or not. If my discretization does something different, then I am likely not using standard finite elements. I don't see that there is much

May 17, 2025

of a grey zone on this — if you know of an example of a paper using finite elements in a way that does not lead to a discretization with the stated properties, be sure to explicitly identify that paper, and where in that paper I might be able to find evidence of a non-standard finite element formulation (I'd be interested!). Otherwise omit this passage, unless you wish to reiterate the statement about the "extended Blatter-Pattyn model" solving for w and the dynamically rather redundant reduced pressure $\tilde{P}$ .

I have rewritten this section. It is my understanding that a finite element method based on the weak form is not equivalent to a variational principle unless the Galerkin method is used (trial functions the same as test functions). Finite element methods are used for many problems that, unlike the Stokes problem, do not have a variational formulation. See the Zienkiewicz and Taylor book, " The Finite Element Method – Volume 1: The Basis", particularly Section 3.8.2.

- line 455 "in matrix form": misleading as M hasn't been cast in matrix form

Word "matrix" changed to "discrete".

- line 459 "M(u, w) is a nonlinear positive-definite function of the velocity" — this is an odd thing to say as I don't think I can have a nontrivial linear positive- definite function, unless it's defined on some bounded set of values (u, w). To wit, if M(u, w) > 0 and M was linear, then it would follow that M(−u, −w) < 0 by linearity, no? Do you mean "convex"?

I wanted to emphasize that the function is both nonlinear and positive-definite. To make it unambiguous, I've changed "nonlinear" to "highly nonlinear".

- line 466 "Discrete variation of the functional corresponds to partial differentiation with respect to each of the discrete variables in V ." Partial differentiation of . . . the discretized functional? It's unclear who the audience is, but probably best to be specific.

I've referred back to equation (43) to be specific.

- equation (48) It's probably a bad idea to use the same letter M for the dissipation potential part of the discretized functional A, and for the Hermitian matrix of the discretized functional, at the same time, even if one M is in calligraphic and the other is not. Also note that the Cryosphere has some rather inflexible (and unimaginative) rules around renderinng matrices and vectors in bold face, in in upright and the other in italicized font. Be prepared fo a bunch of fun at the copyediting stage.

Good point! I've changed all matrices M to bold $\mathbf{M}$ to avoid confusion with script $\mathcal{M}$ for functionals. However, I'm sure there will be lots of problems with typesetting.

May 17, 2025

- line 519 onewards "The form (55) is characteristic of Stokes-type problems, or more generally of constrained minimization problems using Lagrange multipliers. In finite element terminology these are called "mixed" or "saddle point" problems, meaning that velocity components and the pressure occupy different finite element spaces, and that the solution of (55) is actually at the saddle point with respect to the velocity and pressure variables of the quadratic form associated with (55). The matrix M is symmetric but indefinite, with both positive and negative eigenvalues. As a result, the matrix inverse may not be bounded and may lack stability" There are a number of things that are problematic here. The first is the statement that the generic form of defines a saddle point problem. That is not true unless you add that A has to be at least positive semi-definite. The next is the statement that "these are called mixed or saddle point problems, meaning that velocity components and the pressure occupy different finite element spaces." That is not the meaning of a saddle point problem. A saddle point problem corresponds to a problem of minimizing with respect to one variable (velocity) and maximizing with respect to another (pressure). You also can't say that the matrix inverse may not be bounded (if the inverse of a matrix exists, it is bounded!) — the point you're presumably getting at without spelling it out is this: implicit in the discretization is that you are looking at a family of discretizations of progressively of different resolutions resolution (parameterized by maximum element size, call it h, combined with a non-zero lower bound on internal element angles), and you want the corresponding family of saddle points to have a unique limit as h → 0. The statement obout the bound on the inverse of M (and this is what the inf-sup condition guarantees, in my understanding) is that $M^{-1}$ remains bounded as h → 0. No?

The entire section, old lines 513-559, has been rewritten. This applies to

- line 530 "In this case only the matrix A exists, it is elliptic" — two things: i) you seem to be saying that A only exists in this case and ii) "elliptic" is a term you haven't defined up to now, I think. Symmetric positive-definite?

Eliminated

- also line 530 "As a result, the standard Blatter-Pattyn model is a well-behaved and stable unconstrained minimization problem" — this is an unfortunate choice of words, since you've just told the reader that $B = B^T = 0$, in which case M is manifestly not invertible; it seems like you have to state that you're not actually solving (55) but only the reduced version Au = f, for which the statement in question is true.

Corrected

- line 544 "In fact, this is a problem for all inf-sup stable elements with $n_p \neq n_w$, such as the Taylor-Hood element, for example" two things i) you have not de- fined what you mean by "inf-sup stable elements" and ii) the second half of the sentence about Taylor Hood elements is redundant. You could say "is a problem for all inf-sup stable elements with $n_p \neq n_w$". Comment on whether $n_p \neq n_w$ is a prerequisite for an element pair to be inf-sup stable (which I think is true); the last sentence grammatically leaves open the possibility that you could have $n_p = n_w$. If that is the case, comment on the implications for your choice of basis functions, which do have $n_p = n_w$. Are they necessarily not inf-sup stable? This requires an explicit comment, not a response that says "I don't want to address this issue". See discussion re unconstrained minimization being a red herring, unless you can show that your divergence-free basis functions are dense in the underlying function space.

In terms of sequencing the three items (1) Blatter-Pattyn (2) extended Blatter-Pattyn and (3) Stokes / transformed Stokes, it would actually be preferable to reverse the current order since that would mean introducing inf-sup conditions before you talk about them for extended Blatter-Pattyn

The inf-sup condition section has been moved earlier, shortly after equation (54). The discussion of $n_p = n_w$ refers only to the extended Blatter-Pattyn model, while the discussion of meshes satisfying the solvability condition is deferred to the next section.

- line 552 "Both the standard and transformed Stokes models are subject to this problem an in general must use inf-sup stable finite elements. Testing for stability is not trivial. However, collections of inf-sup stable elements for the Stokes equations may be found in many papers and books on mixed methods, e.g., Boffi et al. (2008)." and later line 594 "However, this model does work with a variant of the Taylor-Hood grid, the P2-E1 grid, illustrated in Fig. 13A, which does satisfy the solvability condition and this therefore allows for a successful calculation of the vertical velocity." — I think I raised this point in my original review, but I still don't see it being addressed here: are the new finite element basis functions like P2-E1 and P1-E0 inf-sup stable, or are you arguing (see comments immediately below) that this is somehow no longer relevant when you the solvability condiiton for the incompressibility condition is satisfied? Otherwise the discussion of the inf sup condition is still left dangling in thin air here. If you're referring to the many papers and books because you won't be addressing the issue here, at least say so explicitly. See also comment on line 544.

This issue is now addressed in Appendix B, as explained above. The elements, like P1-E0, that satisfy the solvability condition, may possibly also satisfy the inf-sup condition, but not necessarily. If they do, that would be fortuitous.

- Line 564 "As mentioned previously, this is possible to do in the continuum but not necessarily so in the discrete case" — I don't think you've explained that this is "not necessarily possible" to do in the discrete case, unless you mean the discussion around equation (56) that is about to follow. To avoid confusion and

May 17, 2025

leave the reader scurrying for where this was explained previously, especially as you've just referenced section 3.4.1, and there is no mention of not being able to compute w there, and even the discussion starting on line 396 also doesn't actually say that you cannot compute w in the discrete case. In fact, I expect that those running Blatter-Pattyn ice sheet models would beg to differ with the statement that you cannot compute w from the continuity equation, since they will naturally choose a scheme that finds w by simple quadrature. This seems more of an issue with the extended Blatter-Pattyn model of section 3.4.2, where you are forcing yourself to use the same mesh as for (u, v), no?

The discussion at the start of Section 4.3.2 has been modified to address this.

- line575 "$n_p + \lambda_z + \Lambda = n_w$" should presumably be "$n_p + n_{\lambda_z} + n_\Lambda = n_w$"?

Corrected.

• line 599 "Perhaps the main reason for the importance of the solvability condition is that it implies that the Stokes variational principle, (15) or (33), may be transformed into and therefore that it is equivalent to an optimization or minimization problem." — this is at best misleading, but most likely just wrong. Finding stationary points of (15) and (33), and in fact the solution of all Stokes flow problems subject to suitable boundary conditions, is always equivalent to a convex minimzation problem, but that minimization is over an awkward vector space of Sobolev space of divergence free functions. I should: awkward in terms of finding suitable basis functions when discretizing in practice, but not particularly awkward in terms of the abstract analysis of the problem.

For the present set of boundary conditions and choice of rheology, the relevant function space is $V = \{v \in [W^{1+1/n}(\Omega)]^3 : \mathrm{div}\,v = 0 \text{ and } v = 0 \text{ on } \partial\Omega_{b1}, v \cdot n = 0 \text{ on } \partial\Omega_{b2}\}$, and the cited Chen et al (2013) paper would be an appropriate reference for this fact. For the transformed Stokes flow problem, the transformed minimization would also seem to be pretty straightforward, since, if I restrict myself to the same Sobolev space as Chen et al, I obtain a restriction of A to velocities only in the form

$$J(u, v, w) = \int_V \frac{4n\eta_0}{n+1} \dot{\epsilon}^{1+1/n} + \rho g \nabla z_s \cdot (u, v, 0) \mathrm{d}V + \int_{S_{b_2}} \beta |(u, v, w,)|^2 \mathrm{d}\Gamma$$

and J is convex, so any stationary point of J is automatically a minimizer. If you are actually trying to say that you're trying to determine whether the *discretized* variational problem still corresponds to a minimization problem, that is a different matter, but then you can't refer to (15) or (33), which are formulated for general velocity fields.

Much of Section 4.3.2 has been rewritten. I think the revision stating that "The solvability condition can also be used to convert the Stokes variational principle, (15) or (32), from a constrained to an unconstrained minimization problem." takes care of the problem.

May 17, 2025

• line 611 "This result suggests that a conventional Stokes problem, when solved on a grid satisfying the solvability condition, is equivalent to an unconstrained minimization problem and therefore is well behaved." As per the above, this risks sounding like you are unfamiliar with standard results in the field, and I would not put this statement into a paper.

Even if you don't mind giving that appearance, it would still be a bad idea to leave the statement as is, because it could be read to suggest that basis functions that don't satisfy (57) don't correspond to a minimization problem. That would be wrong.

The conventional Stokes flow problem remains equivalent to an unconstrainted minimzation problem even after discretization. That unconstrained minimization is however over the left nullspace of the matrix B in (55) — that is, over all vectors u such that $B^T u = 0$. In general, you don't want to have to figure out what that nullspace is: as you later point out, the basis vectors for that nullspace are no longer sparse when expressed in terms of nodal values of the mesh, so your linear algebra ends up non-sparse and therefore intractable for large problems. Instead, you continue to impose the need for your solution to lie in that nullspace through the Lagrange multipier p, but that doesn't negate the fact that the problem is equivalent to an unconstrained minimization problem..For the discretized problem on a given, fixed mesh, the only condition for this statement to be true is that the left nullspace of B must be non-trivial, and for that to be true, you generally require the matrix to be taller than wide ($n_u + n_v + n_w > n_p$). You definitely don't need $n_w = n_p$, though $n_w = n_p$ will do. (The solvability condition (57) conveniently ensures that the nullspace in question has dimension $n_u + n_v$ and can be written in the form given on line 620, but that is far from the only form of nullspace you could construct. In general, if $n_p < n_w$ for an inf-sup stable basis you have to use more than a basis for (u,v) to construct a divergence-free basis for (u, v, w), but that is not in principle a problem)

Again, this section has been rewritten to take into account the new Appendix B. I hope this resolves the issue.

• line 617 "if a divergence free basis exists" is a misleading way to make this statement. For the original continuum problem, the divergence free basis exists without a doubt. (Again, Chen et al 2013 are the correct reference for the particular problem you have in mind). My suspicion is that the book you are referencing has in mind a divergence free basis that is dense in the underlying function space, in the sense that it can approximate any element of $V = \{v \in [W^{1+1/n}(\Omega)]^3 : div v = 0 \, and \, v = 0 \, on \, \partial\Omega_{b1}, v \cdot n = 0 \, on \, \partial\Omega_{b2}\}$ to arbitrary accuracy simply by imposing a suitably small maximum edge length.

That is not something that you have demonstrated. You seem to be alluding to the equivalence to an unconstrained minimization problem as evidence that the problem you're solving "is well behaved", without saying what you mean by that, precisely. Based on the text in your response to reviewers (though not the text here!) I assume you arguing that you don't need to worry whether your basis functions actually satisfy the inf-sup

May 17, 2025

condition because you can show that your discretized problem is equivalent to an unconstrained finite-dimensional minimzation problem.

If so, that is not: all that the equivalence with an unconstrained minimization problem that you've discussed (which applies to the discretized problem) ensures case is that the discretized problem is solvable, for any given mesh. It does not guarantee that the solutions for a family of discretized problems will converge to the solution of the continuum problem in the limit of element size h in the mesh going to zero.

More formally, let the function space spanned by your basis function be $V_h$ for a given mesh with maximum element edge length h. The thing you need to prove is that your diverge free finite element basis functions can adequately approximate any possible solution in the limit of small h, that is inf $\|v-v_h\| \leq C_h \|v\|$  $v_h \in V_h$

where $C_h \to 0$ as $h \to 0$. In the absence of the divergence constraint $B^T u = 0$, that behaviour is well established for all sorts of polynomial basis functions, but is no longer guaranteed to hold if you restrict yourself to a linear subspace of of such polynomial basis functions. How you might prove the same behaviour for your divergence free subspace is beyond my pay grade, but you can't just skip past this issue. That would make a mockery of a lot of work by some very smart people in numerical analysis.

The original section, lines 617-622, has been eliminated.

- line 633 "These tests are described in Appendix A" — I checked the appendix as well as Pattyn et al (2008), and it's unclear to me how the "true" solution relative to which the error is computed in figure 3. And in the same vein, if there is a true solution, you should plot that in figure 4.

Inserted "converge to the same transport value, obtained by Richardson extrapolation".

- line 702 "It would be computationally cheaper to use the standard Blatter-Pattyn approximation $(\xi, \hat{\xi} = 0)$ instead, solving only for the horizontal variables and coupling to the Stokes model with $p = 0$ and $w = w(u, v)$ at the interface but this, however, implies much more complicated programming." — see main comment re response to reviewers. That response argues (correctly, in my view) that Blatter- Pattyn is cheaper than a Stokes flow model. Which is understandable given that it has fewer degrees of freedom to solve for. The extended Blatter-Pattyn model on the other hand is constructed to have exactly the same number of degrees of freedom as the Stokes flow model, and it is *not* obvious that it will be much cheaper to solve than the Stokes flow model (transformed or otherwise). Please provide evidence (ideally here!) that it remains sufficiently cheaper to solve than the Stokes flow model to warrant doing this. As per my main comment, using the extended Blatter-Pattyn model will clearly incur a potentially significant model er- ror relative to a Stokes flow solver, so it is important to know whether the reduction in computational cost is worth the increased error.

May 17, 2025

Included the statement that "For example, the extended BP calculation of Test B at a 40x40 resolution, as in Figs. 5 and 6, is some 4.3 times cheaper than the corresponding transformed Stokes calculation". Hope this addresses the issue.

- line 725 "Somewhat counterintuitively, the Stokes region occupies the upper part of the domain in Fig. 7 and includes the obstacle, while the Blatter-Pattyn region occupies much of the bottom part of the domain" — I am not sure this is counterintiuitive in view of the no-penetration boundary condition combined with the moderate angle: basically, your boundary condition ensures that dw/dx is small along the bed, while there is no reason why du/dz would be. I am however not convinced that this test goes far enough: really, you should also compute the Stokes solution across the domain to see

Removed the word "counterintuitively".

- line 757 "The first method, to be called the BP+ approximation" and later line 773 "An Improved Blatter-Pattyn or BP+ Approximation" line 812 "Remarkably, this same model, i.e., the BP+ approximation, was introduced by Herterich (1987)" line 1029 "Remarkably, the BP+ approximation is actually the same as a model originally proposed by Herterich (1987)." etc ... Given that Herterich predates either Blatter or Pattyn, and given that "BP+" suggests that this is somehow a development of Blatter and Pattyn, this really needs to be "Herterich's improved Blatter-Pattyn Approximation" or better just the "Herterich model". Definitely not "BP+", thereby giving further credit to Blatter and Pattyn while further consigning the person who actually invented this to a footnote? Let's have some respect.

As noted earlier, replaced "BT+" with "Herterich" everywhere.

- line 839 "Both BP+ versions converge to the same solution" — this confuses "model" (a set of equations" with "solution algorithm". These are not two dif- ferent "BP+ versions", they are different solution algorithms

Changed "versions" to "algorithms".

- line 868 "logically rectangular" — what does that mean? Producing the same graph as a rectangular grid? Define before using . . .

Replaced this with "As described in Appendix C, our test problem meshes are logically rectangular, i.e., divided into $n$ cells horizontally and $m$ cells vertically". Hope this is sufficient.

- line 1007 "On the other hand, the pressure in the P2-E1 case is highly oscillatory but well behaved in the P2-P1 case. However, this is not at all concerning since the transformed pressure, a Lagrange multiplier, has no physical significance" — so I think you've just demonstrated that your P2-E1 element most likely does not satisfy the inf-sup condition, and you're effectively left hoping that the lack of convergence in the limit of small element size only affects the pressure, but not the velocity. Comment?

This has been replaced with "This is consistent with the discussion in Appendix B since the P2-P1 mesh is chosen to satisfy the inf-sup condition, which is concerned with regularizing the pressure solution (i.e., a condition on the $\mathbf{B}$ matrix), while the P2-E1 mesh has no such requirement". Hope this is sufficient to clarify the issue.

- line 1011 Summary and Discussion — reading this, all is fine and dandy? Nothing that you would flag as an open question or necessary areas of future research where things need to be developed further, followed up on, etc? Any weaknesses? No?

A minor change inserted at the end concerning Appendix B but no other additions.

- Appendix B This is a follow up on a comment from my first review, the response to which was that you thought the P1-E0 element is quite flexible and can be used with quite general triangulations. The trianglulations you require are not "quite general". By construction, your mesh cannot be unstructured as you require elements to be stackable to make columns. You should comment on that as it negates one of the usual advantages of finite elements, which is to permit adaptive meshing. For instance, at grounding lines, ice stream shear margins etc, you may want to have high resolution *near the bed* around a transition in sliding behaviour, but not extend that resolution throughout a column. Moreover, if you are required to keep the interior angles of elements from becoming excessively small, your mesh also makes it somewhat difficult to have variable vertical triangle edge lengths — adjacent columns need to have similar triangle numbers and triangle edge lengths. This needs to be flagged somewhere.

[revised manuscript text omitted]

May 17, 2025

---

## Author Response (AR3)

**Response to Editor – Prof. Josefin Ahlkrona**

**Egusphere-2024-1052 Article**

June 21, 2025

**Editor Request**

Thank you for your second revised version of the manuscript, which takes into consideration the reviewers suggestion on model names, as well as an updated discussion on the inf-sup condition. The inf-sup condition is a topic which can always be discussed at great lengths, but I think for the scope of this paper the current arguments are sufficient, especially with the support of the well designed numerical experiments. I think this is an interesting contribution and am happy to accept the paper for publication after one very minor edit: I am happy to see that you included two of the references mentioned by the review, but I think perhaps you missed Glowinski & Rappaz, please add this final reference as suggested by the reviewer.

**Author Response**

The Glowinski and Rappaz reference has been added. The added reference is indicated in blue in the manuscript.